# Clipped Gradient Methods for Nonsmooth Convex Optimization under Heavy-Tailed Noise: A Refined Analysis[*]

**Zijian Liu**
Stern School of Business, New York University
zl3067@stern.nyu.edu

## Abstract

Optimization under heavy-tailed noise has become popular recently, since it better fits many modern machine learning tasks, as captured by empirical observations. Concretely, instead of a finite second moment on gradient noise, a bounded $\mathfrak{p}$-th moment where $\mathfrak{p} \in (1, 2]$ has been recognized to be more realistic (say being upper bounded by $\sigma_{\mathfrak{l}}^{\mathfrak{p}}$ for some $\sigma_{\mathfrak{l}} \geq 0$). A simple yet effective operation, gradient clipping, is known to handle this new challenge successfully. Specifically, Clipped Stochastic Gradient Descent (Clipped SGD) guarantees a high-probability rate $\mathcal{O}(\sigma_{\mathfrak{l}} \ln(1/\delta) T^{\frac{1}{\mathfrak{p}}-1})$ (resp. $\mathcal{O}(\sigma_{\mathfrak{l}}^2 \ln^2(1/\delta) T^{\frac{2}{\mathfrak{p}}-2})$) for nonsmooth convex (resp. strongly convex) problems, where $\delta \in (0, 1]$ is the failure probability and $T \in \mathbb{N}$ is the time horizon. In this work, we provide a refined analysis for Clipped SGD and offer two rates, $\mathcal{O}(\sigma_{\mathfrak{l}} d_{\text{eff}}^{-\frac{1}{2\mathfrak{p}}} \ln^{1-\frac{1}{\mathfrak{p}}}(1/\delta) T^{\frac{1}{\mathfrak{p}}-1})$ and $\mathcal{O}(\sigma_{\mathfrak{l}}^2 d_{\text{eff}}^{-\frac{1}{\mathfrak{p}}} \ln^{2-\frac{2}{\mathfrak{p}}}(1/\delta) T^{\frac{2}{\mathfrak{p}}-2})$, faster than the aforementioned best results, where $d_{\text{eff}} \geq 1$ is a quantity we call the *generalized effective dimension*. Our analysis improves upon the existing approach in two respects: better utilization of Freedman's inequality and finer bounds for clipping error under heavy-tailed noise. In addition, we extend the refined analysis to convergence in expectation and obtain new rates that break the known lower bounds. Lastly, to complement the study, we establish new lower bounds for both high-probability and in-expectation convergence. Notably, the in-expectation lower bounds match our new upper bounds, indicating the optimality of our refined analysis for convergence in expectation.

## 1 Introduction

In first-order methods for stochastic optimization, one can only query an unbiased though noisy gradient and then implement a gradient descent step, which is known as Stochastic Gradient Descent (SGD) (Robbins & Monro, 1951). Under the widely assumed finite variance condition, i.e., the gradient noise[1] has a finite second moment, the in-expectation convergence of SGD has been substantially studied (Bottou et al., 2018; Lan, 2020).

However, many recent empirical observations suggest that the finite variance assumption might be too strong and could be violated in different tasks (Simsekli et al., 2019; Zhang et al., 2020; Zhou et al., 2020; Garg et al., 2021; Gurbuzbalaban et al., 2021; Hodgkinson & Mahoney, 2021; Battash et al., 2024). Instead, a bounded $\mathfrak{p}$-th moment condition where $\mathfrak{p} \in (1, 2]$ (say with an upper bound $\sigma_{\mathfrak{l}}^{\mathfrak{p}}$ for some $\sigma_{\mathfrak{l}} \geq 0$) better fits modern machine learning, which is named heavy-tailed noise. Facing this new challenge, SGD has been proved to exhibit undesirable behaviors (Zhang et al., 2020; Sadiev et al., 2023). Therefore, an algorithmic change is necessary. A simple yet effective operation, gradient clipping, is known to handle this harder situation successfully with both favorable practical performance and provable theoretical guarantees (see, e.g., Pascanu et al. (2013); Zhang et al. (2020)). The clipping mechanism replaces the stochastic gradient $\mathbf{g}_t$ in every iterate of SGD

---

[*]This is a self-contained conference version of the full paper available at https://arxiv.org/abs/2512.23178. Compared to the full version, we omit the formal statements of lower bounds and their proofs.

[1]This refers to the difference between the stochastic estimate and the true gradient.

with its truncated counterpart $\text{clip}_{\tau_t}(\mathbf{g}_t)$, resulting in a method known as Clipped SGD, where $\tau_t$ is called the clipping threshold and $\text{clip}_\tau(\mathbf{g}) \triangleq \min\{1, \tau/\|\mathbf{g}\|\}\mathbf{g}$ is the clipping function.

Specifically, for nonsmooth convex (resp. strongly convex) optimization, Clipped SGD achieves a high-probability rate $\mathcal{O}(\sigma_{\mathfrak{l}}\ln(1/\delta)T^{\frac{1}{\mathfrak{p}}-1})^2$ (resp. $\mathcal{O}(\sigma_{\mathfrak{l}}^2\ln^2(1/\delta)T^{\frac{2}{\mathfrak{p}}-2})$) (Liu & Zhou, 2023), where $\delta \in (0,1]$ is the failure probability and $T \in \mathbb{N}$ is the time horizon. These two results seem to be optimal as they match the existing in-expectation lower bounds (Nemirovski & Yudin, 1983; Vural et al., 2022; Zhang et al., 2020), if viewing the $\text{poly}(\ln(1/\delta))$ term as a constant. However, a recent advance (Das et al., 2024) established a better rate $\mathcal{O}(\sigma_{\mathfrak{l}}d_{\text{eff}}^{-\frac{1}{4}}\sqrt{\ln(\ln(T)/\delta)/T})$ for general convex problems when $\mathfrak{p} = 2$, where $1 \le d_{\text{eff}} \le d$ is known as the *effective dimension* (also named *intrinsic dimension* (Tropp, 2015)) and $d$ is the true dimension. This reveals that the in-expectation lower bound does not necessarily apply to the term containing $\text{poly}(\ln(1/\delta))$. More importantly, such a result hints that a general improvement may exist for all $\mathfrak{p} \in (1,2]$.

This work confirms that a general improvement does exist by providing a refined analysis for Clipped SGD. Concretely, we offer two faster rates, $\mathcal{O}(\sigma_{\mathfrak{l}}d_{\text{eff}}^{-\frac{1}{2\mathfrak{p}}}\ln^{1-\frac{1}{\mathfrak{p}}}(1/\delta)T^{\frac{1}{\mathfrak{p}}-1})$ for general convex problems with a known $T$ and $\mathcal{O}(\sigma_{\mathfrak{l}}^2 d_{\text{eff}}^{-\frac{1}{\mathfrak{p}}}\ln^{2-\frac{2}{\mathfrak{p}}}(1/\delta)T^{\frac{2}{\mathfrak{p}}-2})$ for strongly convex problems with an unknown $T$, improved upon the aforementioned best results, where $1 \le d_{\text{eff}} \le \mathcal{O}(d)$ is a quantity that we call the *generalized effective dimension*[3]. Moreover, we devise an algorithmic variant of Clipped SGD named Stabilized Clipped SGD that achieves the same rate[4] for convex objectives listed above in an anytime fashion, i.e., no extra $\text{poly}(\ln T)$ factor even without $T$.

We highlight that our analysis improves upon the existing approach in two respects: 1. We observe a better way to apply Freedman's inequality when analyzing Clipped SGD, which leads to a provably tighter concentration. Remarkably, our approach is fairly simple in contrast to the previous complex iterative refinement strategy (Das et al., 2024). 2. We establish finer bounds for clipping error under heavy-tailed noise, which is another essential ingredient in the analysis for Clipped SGD when the noise has a heavy tail. We believe both of these new insights could be of independent interest and potentially useful for future research.

Furthermore, equipped with the new finer bounds for clipping error, we extend the analysis to in-expectation convergence and obtain two new rates, $\mathcal{O}(\sigma_{\mathfrak{l}}d_{\text{eff}}^{-\frac{2-\mathfrak{p}}{2\mathfrak{p}}}T^{\frac{1}{\mathfrak{p}}-1})$ for general convex objectives and $\mathcal{O}(\sigma_{\mathfrak{l}}^2 d_{\text{eff}}^{-\frac{2-\mathfrak{p}}{\mathfrak{p}}}T^{\frac{2}{\mathfrak{p}}-2})$ for strongly convex problems. Notably, once $\mathfrak{p} < 2$, these two rates are both faster by a $\text{poly}(1/d_{\text{eff}})$ factor than the known optimal lower bounds $\Omega(\sigma_{\mathfrak{l}}T^{\frac{1}{\mathfrak{p}}-1})$ and $\Omega(\sigma_{\mathfrak{l}}^2 T^{\frac{2}{\mathfrak{p}}-2})$ in the corresponding setting (Nemirovski & Yudin, 1983; Vural et al., 2022; Zhang et al., 2020).

Lastly, to complement the study, we establish new lower bounds for both high-probability and in-expectation convergence. Notably, the in-expectation lower bounds match our new upper bounds, indicating the optimality of our refined analysis for convergence in expectation.

## 1.1 RELATED WORK

We review the literature that studies nonsmooth (strongly) convex optimization under heavy-tailed noise. For other different settings, e.g., smooth (strongly) convex or smooth/nonsmooth nonconvex problems under heavy-tailed noise, the interested reader could refer to, for example, Nazin et al. (2019); Davis & Drusvyatskiy (2020); Gorbunov et al. (2020); Mai & Johansson (2021); Cutkosky & Mehta (2021); Wang et al. (2021); Tsai et al. (2022); Holland (2022); Jakovetić et al. (2023); Sadiev et al. (2023); Liu et al. (2023); Nguyen et al. (2023); Puchkin et al. (2024); Gorbunov et al. (2024b); Liu et al. (2024); Armacki et al. (2025); Hübler et al. (2025); Liu & Zhou (2025); Sun et al. (2025), for recent progress.

**High-probability rates.** If $\mathfrak{p} = 2$, Gorbunov et al. (2024a) proves the first $\mathcal{O}(\sigma_{\mathfrak{l}}\sqrt{\ln(T/\delta)/T})$ (resp. $\mathcal{O}(\sigma_{\mathfrak{l}}^2\ln(T/\delta)/T)$) high-probability rate for nonsmooth convex (resp. strongly convex) prob-

---

[2]When stating rates in this section, we only keep the dominant term when $T \to \infty$ and $\delta \to 0$ for simplicity.

[3]We use the same notation to denote the effective dimension and the generalized version proposed by us, since our new quantity can recover the previous one when $\mathfrak{p} = 2$. See discussion after (1) for details.

[4]To clarify, "the same rate" refers to the same lower-order term. The full bound is slightly different.

lems under standard assumptions. If additionally assuming a bounded domain, an improved rate $\mathcal{O}(\sigma_{\mathfrak{l}}\sqrt{\ln(1/\delta)/T})$ for convex objectives is obtained by Parletta et al. (2024). Still for convex problems, Das et al. (2024) recently gives the first refined bound $\mathcal{O}(\sigma_{\mathfrak{l}}d_{\text{eff}}^{-\frac{1}{4}}\sqrt{\ln(\ln(T)/\delta)/T})$ but additionally requiring $T \geq \Omega(\ln(\ln d))$, where $d_{\text{eff}}$ (resp. $d$) is the effective (resp. true) dimension, satisfying $1 \leq d_{\text{eff}} \leq d$. For general $\mathfrak{p} \in (1, 2]$, Zhang & Cutkosky (2022) studies the harder online convex optimization, whose result implies a rate $\mathcal{O}(\sigma_{\mathfrak{l}}\text{poly}(\ln(T/\delta))T^{\frac{1}{\mathfrak{p}}-1})$ for heavy-tailed convex optimization. Later on, Liu & Zhou (2023) establishes two bounds, $\mathcal{O}(\sigma_{\mathfrak{l}}\ln(1/\delta)T^{\frac{1}{\mathfrak{p}}-1})$ and $\mathcal{O}(\sigma_{\mathfrak{l}}^2\ln^2(1/\delta)T^{\frac{2}{\mathfrak{p}}-2})$, for convex and strongly convex problems, respectively. These two rates are the best-known results for general $\mathfrak{p} \in (1, 2]$ and have been recognized as optimal since they match the in-expectation lower bounds (see below), if viewing the $\text{poly}(\ln(1/\delta))$ term as a constant.

**In-expectation rates.** Note that the in-expectation rates for $\mathfrak{p} = 2$ are not worth much attention as they are standard results (Bottou et al., 2018; Lan, 2020). As for general $\mathfrak{p} \in (1, 2]$, many existing works prove the rates $\mathcal{O}(\sigma_{\mathfrak{l}}T^{\frac{1}{\mathfrak{p}}-1})$ and $\mathcal{O}(\sigma_{\mathfrak{l}}^2T^{\frac{2}{\mathfrak{p}}-2})$ (Zhang et al., 2020; Vural et al., 2022; Liu & Zhou, 2023; 2024; Parletta et al., 2025; Fatkhullin et al., 2025; Liu, 2025).

**Lower bounds.** The high-probability lower bounds are not fully explored in the literature. To the best of our knowledge, there are only few results for the general convex case and no lower bounds for the strongly convex case. Therefore, the following discussion is only for convex problems. For $\mathfrak{p} = 2$, Carmon & Hinder (2024) shows a lower bound $\Omega(\sigma_{\mathfrak{l}}\sqrt{\ln(1/\delta)/T})$. However, it is only proved for $d = 1$ (or at most $d = 4$). As such, it cannot reveal useful information for the case that $d$ should also be viewed as a parameter (if more accurately, $d_{\text{eff}}$). In other words, it does not contradict our new refined upper bound. For general $\mathfrak{p} \in (1, 2]$, Raginsky & Rakhlin (2009) is the only work that we are aware of. However, as far as we can check, only the time horizon $T$ is in the right order of $\Omega(T^{\frac{1}{\mathfrak{p}}-1})$. For other parameters, they are either hidden or not tight.

Next, we summarize the in-expectation lower bounds. For convex problems, it is known that any first-order method cannot do better than $\Omega(\sigma_{\mathfrak{l}}T^{\frac{1}{\mathfrak{p}}-1})$ (Nemirovski & Yudin, 1983; Vural et al., 2022). If strong convexity additionally holds, Zhang et al. (2020) establishes the lower bound $\Omega(\sigma_{\mathfrak{l}}^2T^{\frac{2}{\mathfrak{p}}-2})$.

## 2 PRELIMINARY

**Notation.** $\mathbb{N}$ is the set of natural numbers (excluding 0). We denote by $[T] \triangleq \{1, \cdots, T\}, \forall T \in \mathbb{N}$. $\langle \cdot, \cdot \rangle$ represents the standard Euclidean inner product. $\|\mathbf{x}\|$ is the Euclidean norm of the vector $\mathbf{x}$ and $\|\mathbf{X}\|$ is the operator norm of the matrix $\mathbf{X}$. $\text{Tr}(\mathbf{X})$ is the trace of a square matrix $\mathbf{X}$. $\mathbb{S}^{d-1}$ stands for the unit sphere in $\mathbb{R}^d$. Given a convex function $h : \mathbb{R}^d \to \mathbb{R}$, $\nabla h(\mathbf{x})$ denotes an arbitrary element in $\partial h(\mathbf{x})$ where $\partial h(\mathbf{x})$ is the subgradient set of $h$ at $\mathbf{x}$. $\text{sgn}(x)$ is the sign function with $\text{sgn}(0) = 0$.

We study the composite optimization problem in the form of

$$\inf_{\mathbf{x} \in \mathbb{X}} F(\mathbf{x}) \triangleq f(\mathbf{x}) + r(\mathbf{x}),$$

where $\mathbb{X} \subseteq \mathbb{R}^d$ is a nonempty closed convex set. Our analysis relies on the following assumptions.

**Assumption 1.** *There exists $\mathbf{x}_\star \in \mathbb{X}$ such that $F_\star \triangleq F(\mathbf{x}_\star) = \inf_{\mathbf{x} \in \mathbb{X}} F(\mathbf{x})$.*

**Assumption 2.** *Both $f : \mathbb{R}^d \to \mathbb{R}$ and $r : \mathbb{R}^d \to \mathbb{R}$ are convex. In addition, $r$ is $\mu$-strongly convex on $\mathbb{X}$ for some $\mu \geq 0$, i.e., $r(\mathbf{x}) \geq r(\mathbf{y}) + \langle \nabla r(\mathbf{y}), \mathbf{x} - \mathbf{y} \rangle + \frac{\mu}{2} \|\mathbf{x} - \mathbf{y}\|^2, \forall \mathbf{x}, \mathbf{y} \in \mathbb{X}$.*

**Assumption 3.** *$f$ is $G$-Lipschitz on $\mathbb{X}$, i.e., $\|\nabla f(\mathbf{x})\| \leq G, \forall \mathbf{x} \in \mathbb{X}$.*

The above assumptions are standard in the literature (Bottou et al., 2018; Nesterov et al., 2018; Lan, 2020). Next, we consider a fine-grained heavy-tailed noise assumption, the key to obtaining refined convergence for Clipped SGD.

**Assumption 4.** *There exists a function $\mathbf{g} : \mathbb{X} \times \Xi \to \mathbb{R}^d$ and a probability distribution $\mathbb{D}$ on $\Xi$ such that $\mathbb{E}_{\xi \sim \mathbb{D}}[\mathbf{g}(\mathbf{x}, \xi)] = \nabla f(\mathbf{x}), \forall \mathbf{x} \in \mathbb{X}$. In addition, for some $\mathfrak{p} \in (1, 2]$, we have*

$$\mathbb{E}_{\xi \sim \mathbb{D}}\left[|\langle \mathbf{e}, \mathbf{g}(\mathbf{x}, \xi) - \nabla f(\mathbf{x})\rangle|^{\mathfrak{p}}\right] \leq \sigma_{\mathfrak{s}}^{\mathfrak{p}}, \quad \mathbb{E}_{\xi \sim \mathbb{D}}\left[\|\mathbf{g}(\mathbf{x}, \xi) - \nabla f(\mathbf{x})\|^{\mathfrak{p}}\right] \leq \sigma_{\mathfrak{l}}^{\mathfrak{p}}, \quad \forall \mathbf{x} \in \mathbb{X}, \mathbf{e} \in \mathbb{S}^{d-1},$$

*where $\sigma_{\mathfrak{s}}$ and $\sigma_{\mathfrak{l}}$ are two constants satisfying $0 \leq \sigma_{\mathfrak{s}} \leq \sigma_{\mathfrak{l}} \leq \sqrt{\pi d/2}\sigma_{\mathfrak{s}}$.*

*Remark* 1. In the remaining paper, if the context is clear, we drop the subscript $\xi \sim \mathbb{D}$ in $\mathbb{E}_{\xi \sim \mathbb{D}}$ to ease the notation. Moreover, $\mathbf{d}(\mathbf{x}, \xi) \triangleq \mathbf{g}(\mathbf{x}, \xi) - \nabla f(\mathbf{x})$ denotes the error in estimating the gradient.

*Remark* 2. It is noteworthy that Assumption 4 actually implicitly exists in prior works for heavy-tailed stochastic optimization, since Cauchy-Schwarz inequality gives us

$$\mathbb{E}\left[|\langle \mathbf{e}, \mathbf{d}(\mathbf{x}, \xi)\rangle|^{\mathfrak{p}}\right] \leq \mathbb{E}\left[\|\mathbf{e}\|^{\mathfrak{p}} \|\mathbf{d}(\mathbf{x}, \xi)\|^{\mathfrak{p}}\right] = \mathbb{E}\left[\|\mathbf{d}(\mathbf{x}, \xi)\|^{\mathfrak{p}}\right], \forall \mathbf{x} \in \mathbb{X}, \mathbf{e} \in \mathbb{S}^{d-1}.$$

In other words, once the condition $\mathbb{E}\left[\|\mathbf{d}(\mathbf{x}, \xi)\|^{\mathfrak{p}}\right] \leq \sigma_{\mathfrak{l}}^{\mathfrak{p}}, \forall \mathbf{x} \in \mathbb{X}$ is assumed like in prior works, there must exist a real number $0 \leq \sigma_{\mathfrak{s}} \leq \sigma_{\mathfrak{l}}$ such that $\mathbb{E}\left[|\langle \mathbf{e}, \mathbf{d}(\mathbf{x}, \xi)\rangle|^{\mathfrak{p}}\right] \leq \sigma_{\mathfrak{s}}^{\mathfrak{p}}, \forall \mathbf{x} \in \mathbb{X}, \mathbf{e} \in \mathbb{S}^{d-1}$.

*Remark* 3. The reason we can assume $\sigma_{\mathfrak{l}} \leq \sqrt{\pi d/2}\sigma_{\mathfrak{s}}$ is that $\mathbb{E}\left[\|\mathbf{d}(\mathbf{x}, \xi)\|^{\mathfrak{p}}\right] \leq (\pi d/2)^{\frac{\mathfrak{p}}{2}}\sigma_{\mathfrak{s}}^{\mathfrak{p}}$ holds provided $\mathbb{E}\left[|\langle \mathbf{e}, \mathbf{d}(\mathbf{x}, \xi)\rangle|^{\mathfrak{p}}\right] \leq \sigma_{\mathfrak{s}}^{\mathfrak{p}}, \forall \mathbf{e} \in \mathbb{S}^{d-1}$, due to Lemma 4.1 in Cherapanamjeri et al. (2022).

Now we define the following quantity named *generalized effective dimension* (where we use the convention $0 = 0/0$),

$$d_{\text{eff}} \triangleq \sigma_{\mathfrak{l}}^2/\sigma_{\mathfrak{s}}^2 \in \{0\} \cup [1, \pi d/2] = \mathcal{O}(d), \tag{1}$$

in which $d_{\text{eff}} = 0$ if and only if $\sigma_{\mathfrak{l}} = \sigma_{\mathfrak{s}} = 0$, i.e., the noiseless case. As discussed later, this definition recovers the effective dimension used in Das et al. (2024) when $\mathfrak{p} = 2$.

To better understand Assumption 4, we first take $\mathfrak{p} = 2$. Note that a finite second moment of $\mathbf{d}(\mathbf{x}, \xi)$ implies the covariance matrix $\Sigma(\mathbf{x}) \triangleq \mathbb{E}\left[\mathbf{d}(\mathbf{x}, \xi)\mathbf{d}^{\top}(\mathbf{x}, \xi)\right] \in \mathbb{R}^{d \times d}$ is well defined. As such, we can interpret $\sigma_{\mathfrak{l}}$ and $\sigma_{\mathfrak{s}}$ as $\sigma_{\mathfrak{l}}^2 = \sup_{\mathbf{x} \in \mathbb{X}} \text{Tr}(\Sigma(\mathbf{x}))$ and $\sigma_{\mathfrak{s}}^2 = \sup_{\mathbf{x} \in \mathbb{X}} \|\Sigma(\mathbf{x})\|$. In particular, if $\Sigma(\mathbf{x}) \preceq \Sigma, \forall \mathbf{x} \in \mathbb{X}$ holds for some positive semidefinite $\Sigma$ as assumed in Das et al. (2024), then one can directly take $\sigma_{\mathfrak{l}}^2 = \text{Tr}(\Sigma)$ and $\sigma_{\mathfrak{s}}^2 = \|\Sigma\|$, which also recovers the effective dimension defined as $\text{Tr}(\Sigma)/\|\Sigma\|$ in Das et al. (2024).

For general $\mathfrak{p} \in (1, 2]$, as discussed in Remark 2, one can view Assumption 4 as a finer version of the classical heavy-tailed noise condition, the latter omits the existence of $\sigma_{\mathfrak{s}}$. Therefore, Assumption 4 describes the behavior of noise more precisely. Such refinement was only introduced to the classical mean estimation problem (Cherapanamjeri et al., 2022) as far as we know, and hence is new to the optimization literature. In Appendix A, we provide more discussions on how large $d_{\text{eff}}$ can be across different settings.

## 3 CLIPPED STOCHASTIC GRADIENT DESCENT

---
**Algorithm 1** Clipped Stochastic Gradient Descent (Clipped SGD)

---
**Input:** initial point $\mathbf{x}_1 \in \mathbb{X}$, stepsize $\eta_t > 0$, clipping threshold $\tau_t > 0$
**for** $t = 1$ **to** $T$ **do**
    $\mathbf{g}_t^c = \text{clip}_{\tau_t}(\mathbf{g}_t)$ where $\mathbf{g}_t = \mathbf{g}(\mathbf{x}_t, \xi_t)$ and $\xi_t \sim \mathbb{D}$ is sampled independently from the history
    $\mathbf{x}_{t+1} = \text{argmin}_{\mathbf{x} \in \mathbb{X}} r(\mathbf{x}) + \langle \mathbf{g}_t^c, \mathbf{x}\rangle + \frac{\|\mathbf{x} - \mathbf{x}_t\|^2}{2\eta_t}$
**end for**

---

We present the main method studied in this work, Clipped Stochastic Gradient Descent (Clipped SGD), in Algorithm 1. Strictly speaking, the algorithm should be called Proximal Clipped SGD as it contains a proximal update step. However, we drop the word "Proximal" for simplicity. We remark that Clipped SGD with a proximal step has not been fully studied yet and is different from the Prox-Clipped-SGD-Shift method introduced in Gorbunov et al. (2024b), the only work considering composite optimization under heavy-tailed noise that we are aware of.

In comparison to the classical Proximal SGD, Algorithm 1 only contains an extra clipping operation on the stochastic gradient. As pointed out in prior works (e.g., Sadiev et al. (2023)), the additional clipping step is the key to proving the high-probability convergence.

## 4 REFINED HIGH-PROBABILITY RATES

In this section, we will establish refined high-probability convergence results for Clipped SGD. To simplify the notation in the upcoming theorems, we denote by $D \triangleq \|\mathbf{x}_{\star} - \mathbf{x}_1\|$ the distance between

the optimal solution and the initial point. Moreover, given $\delta \in (0, 1]$, we introduce the quantity

$$\tau_\star \triangleq \left( \min \left\{ \frac{\sigma_\mathfrak{s} \sigma_\mathfrak{l}^{\mathfrak{p}-1}}{\ln \frac{3}{\delta}}, \frac{\sigma_\mathfrak{s}^2}{\sigma_\mathfrak{l}^{2-\mathfrak{p}} \mathbb{1} \left[ \mathfrak{p} < 2 \right]} \right\} \right)^{\frac{1}{\mathfrak{p}}}, \tag{2}$$

which is an important value used in the clipping threshold. Recall that $d_{\text{eff}} = \sigma_\mathfrak{l}^2/\sigma_\mathfrak{s}^2$, then $\tau_\star$ can be equivalently written into

$$\tau_\star = \sigma_\mathfrak{l}/\varphi_\star^{1/\mathfrak{p}} \quad \text{where} \quad \varphi_\star \triangleq \max \left\{ \sqrt{d_{\text{eff}}} \ln \frac{3}{\delta}, d_{\text{eff}} \mathbb{1} \left[ \mathfrak{p} < 2 \right] \right\}. \tag{3}$$

### 4.1 GENERAL CONVEX CASE

We start from the general convex case (i.e., $\mu = 0$ in Assumption 2). $\bar{\mathbf{x}}_{T+1}^{\text{cvx}} \triangleq \frac{1}{T} \sum_{t=1}^{T} \mathbf{x}_{t+1}$ in the following denotes the average iterate after $T$ steps. To clarify, $T$ is assumed to be known in advance in this subsection. Though Clipped SGD can provably handle an unknown time horizon $T$, it is well-known to incur extra $\text{poly}(\ln T)$ factors (Liu & Zhou, 2023). To deal with this issue, we propose a variant of Clipped SGD named Stabilized Clipped SGD in Appendix C, which incorporates the stabilization trick introduced by Fang et al. (2022). As an example, Theorem 11 in Appendix E shows that Stabilized Clipped SGD converges at an almost identical rate to Theorem 1 below, but in an anytime fashion without incurring any $\text{poly}(\ln T)$ factor.

**Theorem 1.** *Under Assumptions 1, 2 (with $\mu = 0$), 3 and 4, for any $T \in \mathbb{N}$ and $\delta \in (0, 1]$, setting $\eta_t = \eta_\star, \tau_t = \max \left\{ 2G, \tau_\star T^{\frac{1}{\mathfrak{p}}} \right\}, \forall t \in [T]$ where $\eta_\star$ is a properly picked stepsize (explicated in Theorem 10), then Clipped SGD (Algorithm 1) guarantees that with probability at least $1 - \delta$, $F(\bar{\mathbf{x}}_{T+1}^{\text{cvx}}) - F_\star$ converges at the rate of*

$$\mathcal{O} \left( \frac{(\varphi + \ln \frac{3}{\delta}) GD}{T} + \frac{GD}{\sqrt{T}} + \frac{(\sigma_\mathfrak{s}^{\frac{2}{\mathfrak{p}}-1} \sigma_\mathfrak{l}^{2-\frac{2}{\mathfrak{p}}} + \sigma_\mathfrak{s}^{\frac{1}{\mathfrak{p}}} \sigma_\mathfrak{l}^{1-\frac{1}{\mathfrak{p}}} \ln^{1-\frac{1}{\mathfrak{p}}} \frac{3}{\delta}) D}{T^{1-\frac{1}{\mathfrak{p}}}} \right),$$

*where $\varphi \leq \varphi_\star$ is a constant (explicated in Theorem 10) and equals $\varphi_\star$ when $T = \Omega \left( \frac{G^\mathfrak{p}}{\sigma_\mathfrak{l}^\mathfrak{p}} \varphi_\star \right)$.*

To better understand Theorem 1, we first consider a special case of $\mathfrak{p} = 2$ (i.e., the classical finite variance condition) and obtain a rate at most $\mathcal{O} \left( \frac{(\sqrt{d_{\text{eff}}}+1) \ln(\frac{1}{\delta}) GD}{T} + \frac{(G+\sigma_\mathfrak{l}+\sqrt{\sigma_\mathfrak{s}\sigma_\mathfrak{l} \ln(\frac{1}{\delta})})D}{\sqrt{T}} \right)$. In comparison, the previous best high-probability bound in the finite variance setting proved by Das et al. (2024) is $\mathcal{O} \left( C_T + \frac{(\sqrt{d_{\text{eff}}}+\frac{G}{\sigma_\mathfrak{s}}) \ln(\frac{\ln T}{\delta}) GD}{T} + \frac{(G+\sigma_\mathfrak{l}+\sqrt{\sigma_\mathfrak{s}(\sigma_\mathfrak{l}+G) \ln(\frac{\ln T}{\delta})})D}{\sqrt{T}} \right)$, but under an extra requirement $T \geq \Omega(\ln(\ln d))$, where $C_T$ is a term in the order of $\mathcal{O}(T^{-\frac{3}{2}})$ but will blow up to $+\infty$ when the variance approaches 0. As one can see, even in this special case, our result immediately improves upon Das et al. (2024) in the following three folds: 1. Our theory works for any time horizon $T \in \mathbb{N}$. 2. Our bound is strictly better than theirs by shaving off many redundant terms. Especially, the dependence on $\delta$ is only $\ln(1/\delta)$ in contrast to their $\ln((\ln T)/\delta)$. 3. Our rate will not blow up when $\sigma_\mathfrak{l} \to 0$ (equivalently, $\sigma_\mathfrak{s} \to 0$) and instead recover the standard $\mathcal{O}(GD/\sqrt{T})$ result for deterministic nonsmooth convex optimization (Nesterov et al., 2018).

Next, the prior best result for $\mathfrak{p} \in (1, 2]$ is $\mathcal{O} \left( \frac{GD \ln \frac{1}{\delta}}{\sqrt{T}} + \frac{\sigma_\mathfrak{l} D \ln \frac{1}{\delta}}{T^{1-\frac{1}{\mathfrak{p}}}} \right)$ (Liu & Zhou, 2023), whose dominant term is $\mathcal{O}(\sigma_\mathfrak{l} D \ln(1/\delta) T^{\frac{1}{\mathfrak{p}}-1})$ as $T$ becomes larger. In comparison, using $d_{\text{eff}} = \sigma_\mathfrak{l}^2/\sigma_\mathfrak{s}^2$, the lower-order term in Theorem 1 can be written as $\mathcal{O}(\sigma_\mathfrak{l} D(d_{\text{eff}}^{\frac{1}{2}-\frac{1}{\mathfrak{p}}} + d_{\text{eff}}^{-\frac{1}{2\mathfrak{p}}} \ln^{1-\frac{1}{\mathfrak{p}}}(1/\delta)) T^{\frac{1}{\mathfrak{p}}-1})$. Therefore, Theorem 1 improves upon Liu & Zhou (2023) for large $T$ by a factor of

$$\rho \triangleq \Theta \left( \frac{d_{\text{eff}}^{\frac{1}{2}-\frac{1}{\mathfrak{p}}} + d_{\text{eff}}^{-\frac{1}{2\mathfrak{p}}} \ln^{1-\frac{1}{\mathfrak{p}}} \frac{1}{\delta}}{\ln \frac{1}{\delta}} \right) = \Theta \left( \frac{1}{d_{\text{eff}}^{\frac{2-\mathfrak{p}}{2\mathfrak{p}}} \ln \frac{1}{\delta}} + \frac{1}{d_{\text{eff}}^{\frac{1}{2\mathfrak{p}}} \ln^{\frac{1}{\mathfrak{p}}} \frac{1}{\delta}} \right). \tag{4}$$

*Remark* 4. Especially, when $d_{\text{eff}} = \Omega(d)$, $\rho$ could be in the order of $\Theta(\text{poly}(1/d, 1/\ln(1/\delta)))$. We provide an example in Appendix A showing that $d_{\text{eff}} = \Omega(d)$ is attainable.

For general $T \in \mathbb{N}$, note that $\mathcal{O}(GD \ln(1/\delta)/T + \text{last two terms})$ in Theorem 1 are always smaller than the rate of Liu & Zhou (2023) due to $\sigma_{\mathfrak{s}} \leq \sigma_{\mathfrak{l}}$. Therefore, we only need to pay attention to the redundant term $\mathcal{O}(\varphi GD/T)$. Observe that a critical time could be $T_\star = \Theta(\varphi_\star^2) = \Theta(d_{\text{eff}} \ln^2(1/\delta) + d_{\text{eff}}^2 \mathbb{1}\,[\mathfrak{p} < 2])^5$. Once $T \geq T_\star$, we can ignore $\mathcal{O}(\varphi GD/T)$ as it is at most $\mathcal{O}(GD/\sqrt{T})$ now. It is currently unknown whether the term $\mathcal{O}(\varphi GD/T)$ is inevitable or can be removed to obtain a better bound than Liu & Zhou (2023) for any $T \in \mathbb{N}$. We remark that similar additional terms also appear in the refined rate for $\mathfrak{p} = 2$ by Das et al. (2024) as discussed before.

## 4.2 STRONGLY CONVEX CASE

We now move to the strongly convex case (i.e., $\mu > 0$ in Assumption 2). $\bar{\mathbf{x}}_{T+1}^{\text{str}} \triangleq \frac{\sum_{t=1}^{T}(t+4)(t+5)\mathbf{x}_{t+1}}{\sum_{t=1}^{T}(t+4)(t+5)}$ in the following denotes the weighted average iterate after $T$ steps. Unlike the general convex case, we do not need to know $T$ in advance to remove the extra $\text{poly}(\ln T)$ factor.

**Theorem 2.** *Under Assumptions 1, 2 (with $\mu > 0$), 3 and 4, for any $T \in \mathbb{N}$ and $\delta \in (0, 1]$, setting $\eta_t = \frac{6}{\mu t}, \tau_t = \max\left\{2G, \tau_\star t^{\frac{1}{\mathfrak{p}}}\right\}, \forall t \in [T]$, then Clipped SGD (Algorithm 1) guarantees that with probability at least $1 - \delta$, both $F(\bar{\mathbf{x}}_{T+1}^{\text{str}}) - F_\star$ and $\mu \|\mathbf{x}_{T+1} - \mathbf{x}_\star\|^2$ converge at the rate of*

$$\mathcal{O}\left(\frac{\mu D^2}{T^3} + \frac{(\varphi^2 + \ln^2 \frac{3}{\delta})G^2}{\mu T^2} + \frac{G^2}{\mu T} + \frac{\sigma_{\mathfrak{s}}^{\frac{4}{\mathfrak{p}}-2}\sigma_{\mathfrak{l}}^{4-\frac{4}{\mathfrak{p}}} + \sigma_{\mathfrak{s}}^{\frac{2}{\mathfrak{p}}}\sigma_{\mathfrak{l}}^{2-\frac{2}{\mathfrak{p}}} \ln^{2-\frac{2}{\mathfrak{p}}} \frac{3}{\delta}}{\mu T^{2-\frac{2}{\mathfrak{p}}}}\right),$$

*where $\varphi \leq \varphi_\star$ is the same constant as in Theorem 1 and equals $\varphi_\star$ when $T = \Omega\left(\frac{G^{\mathfrak{p}}}{\sigma_{\mathfrak{l}}^{\mathfrak{p}}}\varphi_\star\right)$.*

*Remark 5.* The problem studied in prior works (e.g., Liu & Zhou (2023); Gorbunov et al. (2024a)) considers strongly convex and Lipschitz $f$ with $r = 0$, which seems different from our assumption of strongly convex $r$. However, a simple reduction can convert their instance to fit our setting. Moreover, the first term $\mathcal{O}(\mu D^2/T^3)$ in Theorem 2 can also be omitted in that case (as we will do so in the following discussion). We refer the interested reader to Appendix B for the reduction and why the term $\mathcal{O}(\mu D^2/T^3)$ can be ignored.

To save space, we only compare with the rate $\mathcal{O}\left(\frac{G^2 \ln^2 \frac{1}{\delta}}{\mu T} + \frac{(\sigma_{\mathfrak{l}}^2 + \sigma_{\mathfrak{l}}^{\mathfrak{p}}G^{2-\mathfrak{p}}) \ln^2 \frac{1}{\delta}}{\mu T^{2-\frac{2}{\mathfrak{p}}}}\right)$ (Liu & Zhou, 2023) for general $\mathfrak{p} \in (1, 2]$. For the special case $\mathfrak{p} = 2$, the rate of Liu & Zhou (2023) is almost identical to the bound of Gorbunov et al. (2024a); moreover, as far as we know, no improved result like Das et al. (2024) has been obtained to give a better bound for the term containing $\text{poly}(\ln(1/\delta))$. Similar to the discussion after Theorem 1, one can find that for large $T$, the improvement over Liu & Zhou (2023) is at least by a factor of

$$\rho^2 \overset{(4)}{=} \Theta\left(\frac{1}{d_{\text{eff}}^{\frac{2-\mathfrak{p}}{\mathfrak{p}}} \ln^2 \frac{1}{\delta}} + \frac{1}{d_{\text{eff}}^{\frac{1}{\mathfrak{p}}} \ln^{\frac{2}{\mathfrak{p}}} \frac{1}{\delta}}\right) = \Theta\left(\text{poly}\left(\frac{1}{d_{\text{eff}}}, \frac{1}{\ln \frac{1}{\delta}}\right)\right).$$

For general $T \in \mathbb{N}$, every term in Theorem 2 is still better except for $\mathcal{O}(\varphi^2 G^2/(\mu T^2))$. However, this extra term has no effect once $T \geq T_\star = \Theta(\varphi_\star^2) = \Theta(d_{\text{eff}} \ln^2(1/\delta) + d_{\text{eff}}^2 \mathbb{1}\,[\mathfrak{p} < 2])$, the same critical time for Theorem 1 (a similar discussion to Footnote 5 also applies here), since it is at most $\mathcal{O}(G^2/(\mu T))$ now, being dominated by other terms. Same as before, it is unclear whether this redundant term $\mathcal{O}(\varphi^2 G^2/(\mu T^2))$ can be shaved off to conclude a faster rate for any $T \in \mathbb{N}$ or not. We leave it as future work and look forward to it being addressed.

## 5 PROOF SKETCH AND NEW INSIGHTS

In this section, we sketch the proof of Theorem 1 as an example and introduce our new insights in the analysis. To start with, given $T \in \mathbb{N}$ and suppose $\eta_t = \eta, \tau_t = \tau, \forall t \in [T]$ for simplicity, we

---

[5]Actually, any $T_\star$ that makes $\mathcal{O}(\varphi GD/T)$ in Theorem 1 smaller than the sum of the terms left is enough. Hence, it is possible to find a smaller critical time. We keep this one here due to its clear expression.

have the following inequality for Clipped SGD (see Lemma 4 in Appendix F), which holds almost surely without any restriction on $\tau$,

$$F(\bar{\mathbf{x}}_{T+1}^{\mathrm{cvx}}) - F_\star \leq \frac{D^2}{\eta T} + \frac{2I_T^{\mathrm{cvx}}}{T}, \quad \text{where } I_T^{\mathrm{cvx}} \text{ is a residual term in the order of}$$

$$I_T^{\mathrm{cvx}} = \mathcal{O}\left(\eta\left(\underbrace{\max_{t \in [T]}\left(\sum_{s=1}^{t}\langle \mathbf{d}_s^{\mathrm{u}}, \mathbf{y}_s\rangle\right)^2}_{\mathrm{I}} + \underbrace{\sum_{t=1}^{T}\|\mathbf{d}_t^{\mathrm{u}}\|^2}_{\mathrm{II}} + \underbrace{\left(\sum_{t=1}^{T}\|\mathbf{d}_t^{\mathrm{b}}\|\right)^2}_{\mathrm{III}} + G^2 T\right)\right), \tag{5}$$

in which $\mathbf{d}_t^{\mathrm{u}} \triangleq \mathbf{g}_t^{\mathrm{c}} - \mathbb{E}_{t-1}[\mathbf{g}_t^{\mathrm{c}}]$ and $\mathbf{d}_t^{\mathrm{b}} \triangleq \mathbb{E}_{t-1}[\mathbf{g}_t^{\mathrm{c}}] - \nabla f(\mathbf{x}_t)$ respectively denote the unbiased and biased part in the clipping error, where $\mathbb{E}_t[\cdot] \triangleq \mathbb{E}[\cdot \mid \mathcal{F}_t]$ for $\mathcal{F}_t \triangleq \sigma(\xi_1, \cdots, \xi_t)$ being the natural filtration, and $\mathbf{y}_t$ is some predictable vector (i.e., $\mathbf{y}_t \in \mathcal{F}_{t-1}$) satisfying $\|\mathbf{y}_t\| \leq 1$ almost surely.

The term $\eta G^2 T$ in $I_T^{\mathrm{cvx}}$ is standard. Hence, the left task is to bound terms I, II and, III in high probability. In particular, for I and III, we will move beyond the existing approach via a refined analysis. To formalize the difference, we borrow the following bounds for clipping error commonly used in the literature (see, e.g., Sadiev et al. (2023); Liu & Zhou (2023); Nguyen et al. (2023)):

$$\|\mathbf{d}_t^{\mathrm{u}}\| \leq \mathcal{O}(\tau), \quad \mathbb{E}_{t-1}\left[\|\mathbf{d}_t^{\mathrm{u}}\|^2\right] \overset{\text{if } \tau \geq 2G}{\leq} \mathcal{O}(\sigma_{\mathfrak{l}}^{\mathfrak{p}}\tau^{2-\mathfrak{p}}), \quad \|\mathbf{d}_t^{\mathrm{b}}\| \overset{\text{if } \tau \geq 2G}{\leq} \mathcal{O}(\sigma_{\mathfrak{l}}^{\mathfrak{p}}\tau^{1-\mathfrak{p}}). \tag{6}$$

**Term** I. Note that $X_t \triangleq \langle \mathbf{d}_t^{\mathrm{u}}, \mathbf{y}_t\rangle$ is a martingale difference sequence (MDS), then Freedman's inequality (Lemma 10 in Appendix F) implies with probability at least $1 - \delta$, $\sqrt{\mathrm{I}} \leq \mathcal{O}(\max_{t \in [T]}|X_t|\ln(1/\delta) + \sqrt{\sum_{t=1}^{T}\mathbb{E}_{t-1}[X_t^2]\ln(1/\delta)})$ (this inequality is for illustration, not entirely rigorous in math). To the best of our knowledge, prior works studying Clipped SGD under heavy-tailed noise always bound similar terms in the following manner

$$|X_t| \overset{\|\mathbf{y}_t\| \leq 1}{\leq} \|\mathbf{d}_t^{\mathrm{u}}\| \overset{(6)}{\leq} \mathcal{O}(\tau) \quad \text{and} \quad \mathbb{E}_{t-1}[X_t^2] \overset{\|\mathbf{y}_t\| \leq 1}{\leq} \mathbb{E}_{t-1}\left[\|\mathbf{d}_t^{\mathrm{u}}\|^2\right] \overset{(6)}{\leq} \mathcal{O}(\sigma_{\mathfrak{l}}^{\mathfrak{p}}\tau^{2-\mathfrak{p}}).$$

However, a critical observation is that the above-described widely adopted way is very likely to be loose, as the conditional variance can be better controlled by

$$\mathbb{E}_{t-1}[X_t^2] = \mathbf{y}_t^{\top}\mathbb{E}_{t-1}\left[\mathbf{d}_t^{\mathrm{u}}(\mathbf{d}_t^{\mathrm{u}})^{\top}\right]\mathbf{y}_t \overset{\|\mathbf{y}_t\| \leq 1}{\leq} \left\|\mathbb{E}_{t-1}\left[\mathbf{d}_t^{\mathrm{u}}(\mathbf{d}_t^{\mathrm{u}})^{\top}\right]\right\|.$$

Note that $\left\|\mathbb{E}_{t-1}\left[\mathbf{d}_t^{\mathrm{u}}(\mathbf{d}_t^{\mathrm{u}})^{\top}\right]\right\|$ is at most $\mathbb{E}_{t-1}\left[\|\mathbf{d}_t^{\mathrm{u}}\|^2\right]$ but could be much smaller. Inspired by this, we develop a new bound for $\left\|\mathbb{E}_{t-1}\left[\mathbf{d}_t^{\mathrm{u}}(\mathbf{d}_t^{\mathrm{u}})^{\top}\right]\right\|$ in Lemma 1. Consequently, this better utilization of Freedman's inequality concludes a tighter high-probability bound for term I.

Actually, this simple but effective idea has been implicitly used in Das et al. (2024) when $\mathfrak{p} = 2$. However, their proof eventually falls complex due to an argument they call the iterative refinement strategy, which not only imposes extra undesired factors like $\ln((\ln T)/\delta)$ in their final bound but also leads to an additional requirement $T \geq \Omega(\ln(\ln d))$ in their theory. Our analysis indicates that such a complication is unnecessary, instead, one can keep it simple.

**Term** II. For this term, we follow the same way employed in many previous works (e.g., Cutkosky & Mehta (2021); Zhang & Cutkosky (2022)), i.e., let $X_t \triangleq \|\mathbf{d}_t^{\mathrm{u}}\|^2 - \mathbb{E}_{t-1}\left[\|\mathbf{d}_t^{\mathrm{u}}\|^2\right]$ and decompose $\sum_{t=1}^{T}\|\mathbf{d}_t^{\mathrm{u}}\|^2 \overset{(6)}{\leq} \mathcal{O}(\sum_{t=1}^{T}X_t + \sigma_{\mathfrak{l}}^{\mathfrak{p}}\tau^{2-\mathfrak{p}}T)$ then use Freedman's inequality to bound $\sum_{t=1}^{T}X_t$.

*Remark* 6. Although the above analysis follows the literature, we still obtain a refined inequality for $\mathbb{E}_{t-1}\left[\|\mathbf{d}_t^{\mathrm{u}}\|^2\right]$ in Lemma 1, in the sense of dropping the condition $\tau \geq 2G$ required in (6).

**Term** III. Estimating the clipping error $\|\mathbf{d}_t^{\mathrm{b}}\|$ is another key ingredient when analyzing Clipped SGD. As far as we know, all existing works apply the inequality $\|\mathbf{d}_t^{\mathrm{b}}\| \leq \mathcal{O}(\sigma_{\mathfrak{l}}^{\mathfrak{p}}\tau^{1-\mathfrak{p}})$ in (6). However, we show that this important inequality still has room for improvement. In other words, it is in fact not tight, as revealed by our finer bounds in Lemma 1. Thus, our result is more refined.

From the above discussion, in addition to better utilization of Freedman's inequality, the improvement heavily relies on finer bounds for clipping error under heavy-tailed noise, which we give in the following Lemma 1.

**Lemma 1.** *Under Assumptions 3 and 4, and assuming $\tau_t = \tau > 0$, there are:*

$$\|\mathbf{d}_t^{\mathrm{u}}\| \le \mathcal{O}(\tau), \quad \left\|\mathbb{E}_{t-1}\left[\mathbf{d}_t^{\mathrm{u}}(\mathbf{d}_t^{\mathrm{u}})^\top\right]\right\| \overset{if\,\tau\ge 2G}{\le} \mathcal{O}(\sigma_{\mathfrak{s}}^{\mathfrak{p}}\tau^{2-\mathfrak{p}} + \sigma_{\mathfrak{l}}^{\mathfrak{p}}G^2\tau^{-\mathfrak{p}}),$$

$$\mathbb{E}_{t-1}\left[\|\mathbf{d}_t^{\mathrm{u}}\|^2\right] \le \mathcal{O}(\sigma_{\mathfrak{l}}^{\mathfrak{p}}\tau^{2-\mathfrak{p}}), \quad \left\|\mathbf{d}_t^{\mathrm{b}}\right\| \overset{if\,\tau\ge 2G}{\le} \mathcal{O}(\sigma_{\mathfrak{s}}\sigma_{\mathfrak{l}}^{\mathfrak{p}-1}\tau^{1-\mathfrak{p}} + \sigma_{\mathfrak{l}}^{\mathfrak{p}}G\tau^{-\mathfrak{p}}).$$

*Remark* 7. We highlight that Theorem 9 in Appendix D provides a further generalization of clipping error bounds under heavy-tailed noise not limited to clipped gradient methods (even without the requirement in the form of $\tau \ge 2G$), which could be potentially useful for future research.

Except for the standard bound $\|\mathbf{d}_t^{\mathrm{u}}\| \le \mathcal{O}(\tau)$, the other three inequalities in Lemma 1 are either new or improve over the existing results. 1. The bound on $\left\|\mathbb{E}_{t-1}\left[\mathbf{d}_t^{\mathrm{u}}(\mathbf{d}_t^{\mathrm{u}})^\top\right]\right\|$ is new in the heavy-tailed setting. Importantly, observe that $\mathcal{O}(\sigma_{\mathfrak{s}}^{\mathfrak{p}}\tau^{2-\mathfrak{p}} + \sigma_{\mathfrak{l}}^{\mathfrak{p}}G^2\tau^{-\mathfrak{p}}) \le \mathcal{O}(\sigma_{\mathfrak{l}}^{\mathfrak{p}}\tau^{2-\mathfrak{p}})$ due to $\sigma_{\mathfrak{s}} \le \sigma_{\mathfrak{l}}$ and $\tau \ge 2G$, which thereby leads to a tighter high-probability bound for term I in combination with our better application of Freedman's inequality (see the paragraph before starting with **Term** I.). 2. For term $\mathbb{E}_{t-1}\left[\|\mathbf{d}_t^{\mathrm{u}}\|^2\right]$, in contrast to (6), Lemma 1 removes the condition $\tau \ge 2G$. Moreover, the hidden constant in our lemma is actually slightly better. 3. As mentioned above (see the paragraph before starting with **Term** III.), the bound of $\left\|\mathbf{d}_t^{\mathrm{b}}\right\|$ is another key to obtaining a refined result. Precisely, we note that the new bound $\mathcal{O}(\sigma_{\mathfrak{s}}\sigma_{\mathfrak{l}}^{\mathfrak{p}-1}\tau^{1-\mathfrak{p}} + \sigma_{\mathfrak{l}}^{\mathfrak{p}}G\tau^{-\mathfrak{p}})$ improves upon $\mathcal{O}(\sigma_{\mathfrak{l}}^{\mathfrak{p}}\tau^{1-\mathfrak{p}})$ in (6) because of $\sigma_{\mathfrak{s}} \le \sigma_{\mathfrak{l}}$ and $\tau \ge 2G$. Therefore, Lemma 1 guarantees a better control for term III.

Combining all the new insights mentioned, we can finally prove Theorem 1. As one can imagine, the analysis sketched above is essentially more refined than previous works, since we apply tighter bounds for the two central parts in analyzing Clipped SGD, i.e., concentration inequalities and estimation of clipping error. To confirm this claim, we discuss how to recover the existing rate through our finer analysis, the details of which are deferred to Appendix E.

Lastly, we mention that Theorem 2 for strongly convex problems is also inspired by the above two new insights. The full proofs of both Theorems 1 and 2 can be found in Appendix E.

## 6 EXTENSION TO FASTER IN-EXPECTATION CONVERGENCE

In this section, we show that Lemma 1 presented before can also lead to faster in-expectation convergence for Clipped SGD, further highlighting the value of refined clipping error bounds. Proofs of both theorems given below can be found in Appendix E.

This time, we consider a new quantity $\widetilde{\tau}_\star \triangleq \sigma_{\mathfrak{s}}^{\frac{2}{\mathfrak{p}}} / (\sigma_{\mathfrak{l}}^{\frac{2}{\mathfrak{p}}-1}\mathbb{1}\,[\mathfrak{p} < 2])$ for the clipping threshold. Recall that $d_{\mathrm{eff}} = \sigma_{\mathfrak{l}}^2/\sigma_{\mathfrak{s}}^2$, then $\widetilde{\tau}_\star$ can be equivalently written into

$$\widetilde{\tau}_\star = \sigma_{\mathfrak{l}}/\widetilde{\varphi}_\star^{1/\mathfrak{p}} \quad \text{where} \quad \widetilde{\varphi}_\star \triangleq d_{\mathrm{eff}}\mathbb{1}\,[\mathfrak{p} < 2]. \tag{7}$$

*Remark* 8. When $\mathfrak{p} = 2$, $\widetilde{\varphi}_\star = 0 \Rightarrow \widetilde{\tau}_\star = +\infty$, i.e., no clipping operation is required. This matches the well-known fact that SGD provably converges in expectation under the finite variance condition.

### 6.1 GENERAL CONVEX CASE

**Theorem 3.** *Under Assumptions 1, 2 (with $\mu = 0$), 3 and 4, for any $T \in \mathbb{N}$, setting $\eta_t = \eta_\star, \tau_t = \max\left\{2G, \widetilde{\tau}_\star T^{\frac{1}{\mathfrak{p}}}\right\}, \forall t \in [T]$ where $\eta_\star$ is a properly picked stepsize (explicated in Theorem 12), then Clipped SGD (Algorithm 1) guarantees that $\mathbb{E}\left[F(\bar{\mathbf{x}}_{T+1}^{\mathrm{cvx}}) - F_\star\right]$ converges at the rate of*

$$\mathcal{O}\left(\frac{\widetilde{\varphi}GD}{T} + \frac{GD}{\sqrt{T}} + \frac{\sigma_{\mathfrak{s}}^{\frac{2}{\mathfrak{p}}-1}\sigma_{\mathfrak{l}}^{2-\frac{2}{\mathfrak{p}}}D}{T^{1-\frac{1}{\mathfrak{p}}}}\right),$$

*where $\widetilde{\varphi} \le \widetilde{\varphi}_\star$ is a constant (explicated in Theorem 12) and equals $\widetilde{\varphi}_\star$ when $T = \Omega\left(\frac{G^{\mathfrak{p}}}{\sigma_{\mathfrak{l}}^{\mathfrak{p}}}\widetilde{\varphi}_\star\right)$.*

Theorem 3 gives a better lower-order term $\mathcal{O}(\sigma_{\mathfrak{l}} d_{\text{eff}}^{\frac{1}{2} - \frac{1}{\mathfrak{p}}} D T^{\frac{1}{\mathfrak{p}} - 1})$ (recall $d_{\text{eff}} = \sigma_{\mathfrak{l}}^2 / \sigma_{\mathfrak{s}}^2$) than the existing lower bound $\Omega(\sigma_{\mathfrak{l}} D T^{\frac{1}{\mathfrak{p}} - 1})$ (Nemirovski & Yudin, 1983; Vural et al., 2022) by a factor of $\Theta(1/d_{\text{eff}}^{\frac{2-\mathfrak{p}}{2\mathfrak{p}}})$, a strict improvement being polynomial in $1/d_{\text{eff}}$, if $\mathfrak{p} \in (1, 2)$. For the case of an unknown $T$, the interested reader could refer to Theorem 13 in Appendix E.

## 6.2 STRONGLY CONVEX CASE

**Theorem 4.** *Under Assumptions 1, 2 (with $\mu > 0$), 3 and 4, for any $T \in \mathbb{N}$, setting $\eta_t = \frac{6}{\mu t}, \tau_t = \max\left\{2G, \widetilde{\tau}_\star t^{\frac{1}{\mathfrak{p}}}\right\}, \forall t \in [T]$, then Clipped SGD (Algorithm 1) guarantees that both $\mathbb{E}\left[F(\bar{\mathbf{x}}_{T+1}^{\text{str}}) - F_\star\right]$ and $\mu\mathbb{E}\left[\|\mathbf{x}_{T+1} - \mathbf{x}_\star\|^2\right]$ converge at the rate of*

$$\mathcal{O}\left(\frac{\mu D^2}{T^3} + \frac{\widetilde{\varphi}^2 G^2}{\mu T^2} + \frac{G^2}{\mu T} + \frac{\sigma_{\mathfrak{s}}^{\frac{4}{\mathfrak{p}} - 2} \sigma_{\mathfrak{l}}^{4 - \frac{4}{\mathfrak{p}}}}{\mu T^{2 - \frac{2}{\mathfrak{p}}}}\right),$$

*where $\widetilde{\varphi} \leq \widetilde{\varphi}_\star$ is the same constant as in Theorem 3 and equals $\widetilde{\varphi}_\star$ when $T = \Omega\left(\frac{G^{\mathfrak{p}}}{\sigma_{\mathfrak{l}}^{\mathfrak{p}}} \widetilde{\varphi}_\star\right)$.*

Theorem 4 provides a faster rate $\mathcal{O}(\sigma_{\mathfrak{l}}^2 d_{\text{eff}}^{1 - \frac{2}{\mathfrak{p}}} T^{\frac{2}{\mathfrak{p}} - 2})$ than the known lower bound $\Omega(\sigma_{\mathfrak{l}}^2 T^{\frac{2}{\mathfrak{p}} - 2})$ (Zhang et al., 2020) by a factor of $\Theta(1/d_{\text{eff}}^{\frac{2-\mathfrak{p}}{\mathfrak{p}}})$. This is again a strict improvement once $\mathfrak{p} < 2$, and could be in the order of $\text{poly}(1/d)$ if $d_{\text{eff}} = \Omega(d)$.

## 7 LOWER BOUNDS

To complement the study, we provide new high-probability and in-expectation lower bounds for both $\mu = 0$ and $\mu > 0$. We employ information-theoretic methods to establish these new lower bounds, following the existing literature (Raginsky & Rakhlin, 2009; Agarwal et al., 2012; Duchi et al., 2013; Vural et al., 2022; Carmon & Hinder, 2024; Ma et al., 2024). For their formal statements and corresponding proofs, the interested reader could refer to Appendix G in the full version of this work available at https://arxiv.org/abs/2512.23178.

*Remark* 9. One may wonder why our upper bounds can beat the existing lower bounds, and also where the difference between our new lower bounds and the prior ones lies. The key is our fine-grained Assumption 4. Roughly speaking, the existing lower bounds are proved for the following oracle class (we slightly abuse the notation by still using $\mathbf{g}$ to denote the stochastic gradient oracle),

$$\mathfrak{G}_{\sigma_{\mathfrak{l}}}^{\mathfrak{p}} = \left\{\mathbf{g} : \mathbb{R}^d \times \mathfrak{f} \to \mathbb{R}^d : \begin{smallmatrix} \mathbb{E}[\mathbf{g}(\mathbf{x}, f) | \mathbf{x}, f] = \nabla f(\mathbf{x}) \in \partial f(\mathbf{x}) \\ \mathbb{E}[\|\mathbf{g}(\mathbf{x}, f) - \nabla f(\mathbf{x})\|^{\mathfrak{p}} | \mathbf{x}, f] \leq \sigma_{\mathfrak{l}}^{\mathfrak{p}} \end{smallmatrix}, \forall \mathbf{x} \in \mathbb{R}^d, f \in \mathfrak{f}\right\},$$

where $\mathfrak{p} \in (1, 2]$ and $\sigma_{\mathfrak{l}} \geq 0$ are two parameters and $\mathfrak{f}$ is the function class that we are interested in (e.g., the family of $G$-Lipschitz convex functions). In contrast, the oracle class we study is parameterized by one more parameter $\sigma_{\mathfrak{s}} \in \left[\sigma_{\mathfrak{l}}/\sqrt{\pi d/2}, \sigma_{\mathfrak{l}}\right]$ as follows,

$$\mathfrak{G}_{\sigma_{\mathfrak{s}}, \sigma_{\mathfrak{l}}}^{\mathfrak{p}} \triangleq \left\{\mathbf{g} : \mathbb{R}^d \times \mathfrak{f} \to \mathbb{R}^d : \begin{smallmatrix} \mathbb{E}[\mathbf{g}(\mathbf{x}, f) | \mathbf{x}, f] = \nabla f(\mathbf{x}) \in \partial f(\mathbf{x}) \\ \mathbb{E}[|\langle \mathbf{e}, \mathbf{g}(\mathbf{x}, f) - \nabla f(\mathbf{x})\rangle|^{\mathfrak{p}} | \mathbf{x}, f] \leq \sigma_{\mathfrak{s}}^{\mathfrak{p}}, \forall \mathbf{e} \in \mathbb{S}^{d-1} \\ \mathbb{E}[\|\mathbf{g}(\mathbf{x}, f) - \nabla f(\mathbf{x})\|^{\mathfrak{p}} | \mathbf{x}, f] \leq \sigma_{\mathfrak{l}}^{\mathfrak{p}} \end{smallmatrix}, \forall \mathbf{x} \in \mathbb{R}^d, f \in \mathfrak{f}\right\}.$$

Note that there is $\mathfrak{G}_{\sigma_{\mathfrak{s}}, \sigma_{\mathfrak{l}}}^{\mathfrak{p}} \subseteq \mathfrak{G}_{\sigma_{\mathfrak{l}}}^{\mathfrak{p}}$, implying the lower bound proved for $\mathfrak{G}_{\sigma_{\mathfrak{l}}}^{\mathfrak{p}}$ could be loose for $\mathfrak{G}_{\sigma_{\mathfrak{s}}, \sigma_{\mathfrak{l}}}^{\mathfrak{p}}$. Therefore, our upper bounds can surpass the existing lower bounds, and our new lower bounds are established for the fine-grained oracle class $\mathfrak{G}_{\sigma_{\mathfrak{s}}, \sigma_{\mathfrak{l}}}^{\mathfrak{p}}$.

## 7.1 HIGH-PROBABILITY LOWER BOUNDS

**Theorem 5** (Informal). *Under Assumptions 1, 2 (with $\mu = 0$), 3 and 4, assuming $d \geq d_{\text{eff}} \geq 1$ and $\delta \in \left(0, \frac{1}{10}\right)$, any algorithm converges at least at the rate of $\Omega\left(\frac{(\sigma_{\mathfrak{s}}^{\frac{2}{\mathfrak{p}} - 1} \sigma_{\mathfrak{l}}^{2 - \frac{2}{\mathfrak{p}}} + \sigma_{\mathfrak{s}} \ln^{1 - \frac{1}{\mathfrak{p}}} \frac{1}{\delta}) D}{T^{1 - \frac{1}{\mathfrak{p}}}}\right)$ with probability at least $\delta$ when $T$ is large enough.*

**Theorem 6** (Informal). *Under Assumptions 1, 2 (with $\mu > 0$), 3 and 4, assuming $d \geq d_{\text{eff}} \geq 1$ and $\delta \in \left(0, \frac{1}{10}\right)$, any algorithm converges at least at the rate of $\Omega \left( \frac{\sigma_s^{\frac{4}{p}-2} \sigma_t^{4-\frac{4}{p}} + \sigma_s^2 \ln^{2-\frac{2}{p}} \frac{1}{\delta}}{\mu T^{2-\frac{2}{p}}} \right)$ with probability at least $\delta$ when $T$ is large enough.*

Compared to our upper bounds in high probability, i.e., Theorems 1 ($\mu = 0$) and 2 ($\mu > 0$), there are still differences between the terms that contain the $\text{poly}(\ln(1/\delta))$ factor. Closing this important gap is an interesting task, which we leave for future work.

### 7.2 IN-EXPECTATION LOWER BOUNDS

**Theorem 7** (Informal). *Under Assumptions 1, 2 (with $\mu = 0$), 3 and 4, assuming $d \geq d_{\text{eff}} \geq 1$, any algorithm converges at least at the rate of $\Omega \left( \frac{\sigma_s^{\frac{2}{p}-1} \sigma_t^{2-\frac{2}{p}} D}{T^{1-\frac{1}{p}}} \right)$ in expectation when $T$ is large enough.*

**Theorem 8** (Informal). *Under Assumptions 1, 2 (with $\mu > 0$), 3 and 4, assuming $d \geq d_{\text{eff}} \geq 1$, any algorithm converges at least at the rate of $\Omega \left( \frac{\sigma_s^{\frac{4}{p}-2} \sigma_t^{4-\frac{4}{p}}}{\mu T^{2-\frac{2}{p}}} \right)$ in expectation when $T$ is large enough.*

For in-expectation convergence, the above lower bounds match our new upper bounds, i.e., Theorems 3 ($\mu = 0$) and 4 ($\mu > 0$), indicating the optimality of our refined analysis for convergence in expectation.

## 8 CONCLUSION AND FUTURE WORK

In this work, we provide a refined analysis of Clipped SGD and obtain faster high-probability rates than the previously best-known bounds. The improvement is achieved by better utilization of Freedman's inequality and finer bounds for clipping error under heavy-tailed noise. Moreover, we extend the analysis to in-expectation convergence and show new rates that break the existing lower bounds. To complement the study, we establish new lower bounds for both high-probability and in-expectation convergence. Notably, the in-expectation upper and lower bounds match each other, indicating the optimality of our refined analysis for convergence in expectation.

There are still some directions worth exploring in the future, which we list below:

**The extra term.** Each of our refined rates has a higher-order term related to $d_{\text{eff}}$ (e.g., $\mathcal{O}(\varphi G D/T)$ in Theorem 1 and $\mathcal{O}(\varphi^2 G^2/(\mu T^2))$ in Theorem 2). Although it is negligible when $T$ is large, proving/disproving it can be removed for any $T \in \mathbb{N}$ could be an interesting task.

**Gaps in high-probability bounds.** As discussed in Section 7, there are still gaps between high-probability upper and lower bounds for both convex and strongly convex cases. Closing them is an important direction for the future.

**Other optimization problems.** We remark that our two new insights are not limited to nonsmooth convex problems. Instead, they are general concepts/results. Therefore, we believe that it is possible to apply them to other optimization problems under heavy-tailed noise (e.g., smooth (strongly) convex/nonconvex problems) and obtain improved upper bounds faster than existing ones.

### ACKNOWLEDGMENTS

The author thanks anonymous reviewers for their helpful comments and valuable feedback.

### REPRODUCIBILITY STATEMENT

We include the full proofs of all theorems in the appendix.

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

## A  LOWER BOUNDS ON $d_{\text{eff}}$

This section provides lower bounds on $d_{\text{eff}}$ for the additive noise model, i.e., $\mathbf{g}(\mathbf{x}, \xi) = \nabla f(\mathbf{x}) + \xi$. Given $i \in [d]$, $\xi_i$ denotes the $i$-th coordinate of $\xi$, and $\sigma_i \triangleq \left( \mathbb{E}\left[ |\xi_i|^{\mathfrak{p}} \right] \right)^{\frac{1}{\mathfrak{p}}}$ is the $\mathfrak{p}$-th moment of $\xi_i$. Additionally, $\iota : [d] \to [d]$ is the permutation that makes $\sigma_i$ in a nonincreasing order, i.e., $\sigma_{\iota_1} \geq \sigma_{\iota_2} \geq \cdots \geq \sigma_{\iota_{d-1}} \geq \sigma_{\iota_d}$.

### A.1  INDEPENDENT COORDINATES

In this subsection, we assume that all of $\xi_i$ are mutually independent.

- For any $j \in [d]$, we can lower bound

$$\|\xi\|^{\mathfrak{p}} = \left( \sum_{i=1}^{d} \xi_i^2 \right)^{\frac{\mathfrak{p}}{2}} \geq \left( \sum_{i=1}^{j} \xi_{\iota_i}^2 \right)^{\frac{\mathfrak{p}}{2}} \geq j^{\frac{\mathfrak{p}}{2}-1} \sum_{i=1}^{j} |\xi_{\iota_i}|^{\mathfrak{p}},$$

where the last step is by the concavity of $x^{\frac{\mathfrak{p}}{2}}$, which implies $\sigma_{\mathfrak{l}}^{\mathfrak{p}} = \mathbb{E}\left[ \|\xi\|^{\mathfrak{p}} \right] \geq j^{\frac{\mathfrak{p}}{2}-1} \sum_{i=1}^{j} \sigma_{\iota_i}^{\mathfrak{p}}$. Therefore, we can find

$$\sigma_{\mathfrak{l}}^{\mathfrak{p}} \geq \max_{j \in [d]} j^{\frac{\mathfrak{p}}{2}-1} \sum_{i=1}^{j} \sigma_{\iota_i}^{\mathfrak{p}}. \tag{8}$$

- For any $\mathbf{e} \in \mathbb{S}^{d-1}$, we write $\mathbf{e} = \sum_{i=1}^{d} \lambda_i \mathbf{e}_i$ where $\sum_{i=1}^{d} \lambda_i^2 = 1$ and $\mathbf{e}_i$ denotes the all-zero vector except for the $i$-th coordinate, which is one. Therefore, we have

$$\mathbb{E}\left[ |\langle \mathbf{e}, \xi \rangle|^{\mathfrak{p}} \right] = \mathbb{E}\left[ \left| \sum_{i=1}^{d} \lambda_i \xi_i \right|^{\mathfrak{p}} \right] \overset{(a)}{\leq} 2^{2-\mathfrak{p}} \sum_{i=1}^{d} \mathbb{E}\left[ |\lambda_i \xi_i|^{\mathfrak{p}} \right]$$

$$= 2^{2-\mathfrak{p}} \sum_{i=1}^{d} |\lambda_i|^{\mathfrak{p}} \sigma_i^{\mathfrak{p}} \overset{(b)}{\leq} 2^{2-\mathfrak{p}} \left( \sum_{i=1}^{d} \sigma_i^{\frac{2\mathfrak{p}}{2-\mathfrak{p}}} \right)^{1-\frac{\mathfrak{p}}{2}},$$

where $(a)$ holds by $|a + b|^{\mathfrak{p}} \leq |a|^{\mathfrak{p}} + \mathfrak{p} |a|^{\mathfrak{p}-1} \operatorname{sgn}(a) b + 2^{2-\mathfrak{p}} |b|^{\mathfrak{p}}$ (see Proposition 18 of Vural et al. (2022)) and the mutual independence of $\xi_i$, and $(b)$ is due to

$$\sum_{i=1}^{d} |\lambda_i|^{\mathfrak{p}} \sigma_i^{\mathfrak{p}} \leq \left( \sum_{i=1}^{d} \lambda_i^2 \right)^{\frac{\mathfrak{p}}{2}} \left( \sum_{i=1}^{d} \sigma_i^{\frac{2\mathfrak{p}}{2-\mathfrak{p}}} \right)^{1-\frac{\mathfrak{p}}{2}} = \left( \sum_{i=1}^{d} \sigma_i^{\frac{2\mathfrak{p}}{2-\mathfrak{p}}} \right)^{1-\frac{\mathfrak{p}}{2}}.$$

Hence, we know

$$\sigma_{\mathfrak{s}}^{\mathfrak{p}} = \sup_{\mathbf{e} \in \mathbb{S}^{d-1}} \mathbb{E}\left[ |\langle \mathbf{e}, \xi \rangle|^{\mathfrak{p}} \right] \leq 2^{2-\mathfrak{p}} \left( \sum_{i=1}^{d} \sigma_i^{\frac{2\mathfrak{p}}{2-\mathfrak{p}}} \right)^{1-\frac{\mathfrak{p}}{2}}. \tag{9}$$

As such, we can lower bound

$$d_{\text{eff}} = \frac{\sigma_{\mathfrak{l}}^2}{\sigma_{\mathfrak{s}}^2} \overset{(8),(9)}{\geq} \frac{\max_{j \in [d]} j^{1-\frac{2}{\mathfrak{p}}} \left( \sum_{i=1}^{j} \sigma_{\iota_i}^{\mathfrak{p}} \right)^{\frac{2}{\mathfrak{p}}}}{2^{\frac{4}{\mathfrak{p}}-2} \left( \sum_{i=1}^{d} \sigma_i^{\frac{2\mathfrak{p}}{2-\mathfrak{p}}} \right)^{\frac{2}{\mathfrak{p}}-1}}. \tag{10}$$

Though (10) does not directly give a lower bound for $d_{\text{eff}}$ expressed in terms of $d$, it has already provided some useful information. For example, when $\sigma_i$ are all in the same order, (10) implies that $d_{\text{eff}} = \Omega\left( d^{2-\frac{2}{\mathfrak{p}}} \right)$.

### A.2  I.I.D. COORDINATES

In this subsection, we further assume that all $\xi_i$ are i.i.d. and then lower bound $d_{\text{eff}}$ by $d$. Since all coordinates are identically distributed now, we write $\sigma_i = \sigma, \forall i \in [d]$ for some $\sigma \geq 0$ in the following.

### A.2.1 A GENERAL $\Omega(d^{2-\frac{2}{\mathfrak{p}}})$ BOUND

We invoke (10) and plug in $\sigma_i = \sigma$ to obtain

$$d_{\text{eff}} \overset{(10)}{\geq} \frac{\max_{j\in[d]} j\sigma^2}{2^{\frac{4}{\mathfrak{p}}-2}d^{\frac{2}{\mathfrak{p}}-1}\sigma^2} = \frac{d^{2-\frac{2}{\mathfrak{p}}}}{2^{\frac{4}{\mathfrak{p}}-2}}. \tag{11}$$

When $\mathfrak{p} = 2$, the above bound recovers the fact that $d_{\text{eff}} = d$ for $\xi$ with i.i.d. coordinates.

### A.2.2 A SPECIAL $\Omega(d)$ BOUND

Now we consider a special kind of noise. Suppose $\mathfrak{p} \in (1,2)$ and all $\xi_i$ have the characteristic function

$$\mathbb{E}\left[\exp\left(it\xi_i\right)\right] = \exp\left(-\gamma^\alpha |t|^\alpha \left(1 - i\beta \tan\left(\frac{\pi\alpha}{2}\right)\text{sgn}(t)\right)\right), \forall t \in \mathbb{R},$$

where $\alpha = \mathfrak{p} + \epsilon$ for $\epsilon \in (0, 2-\mathfrak{p}]$, $\beta \in [-1,1]$, and $\gamma \geq 0$. Such a distribution is known as $\alpha$-stable distribution satisfying that $\mathbb{E}[\xi_i] = 0$, $\sigma < \infty$, and $\sum_{i=1}^d \xi_i$ equals to $d^{\frac{1}{\alpha}}\xi_1$ in distribution (Zolotarev, 1986; Samorodnitsky & Taqqu, 1994; Nolan, 2020). This suggests that we can lower bound $\sigma_{\mathfrak{l}}^{\mathfrak{p}}$ in another way,

$$\sigma_{\mathfrak{l}}^{\mathfrak{p}} = \mathbb{E}\left[\|\xi\|^{\mathfrak{p}}\right] = \mathbb{E}\left[\left(\sum_{i=1}^d \xi_i^2\right)^{\frac{\mathfrak{p}}{2}}\right] \geq \frac{\mathbb{E}\left[\left|\sum_{i=1}^d \xi_i\right|^{\mathfrak{p}}\right]}{18^{\mathfrak{p}}\mathfrak{p}^{\mathfrak{p}}\left(\frac{\mathfrak{p}}{\mathfrak{p}-1}\right)^{\frac{\mathfrak{p}}{2}}} = \frac{\mathbb{E}\left[\left|d^{\frac{1}{\alpha}}\xi_1\right|^{\mathfrak{p}}\right]}{18^{\mathfrak{p}}\mathfrak{p}^{\mathfrak{p}}\left(\frac{\mathfrak{p}}{\mathfrak{p}-1}\right)^{\frac{\mathfrak{p}}{2}}} = \frac{d^{\frac{\mathfrak{p}}{\mathfrak{p}+\epsilon}}\sigma^{\mathfrak{p}}}{18^{\mathfrak{p}}\mathfrak{p}^{\mathfrak{p}}\left(\frac{\mathfrak{p}}{\mathfrak{p}-1}\right)^{\frac{\mathfrak{p}}{2}}}, \tag{12}$$

where the inequality is due to Burkholder (1973). Therefore, in this special case, we have

$$d_{\text{eff}} = \frac{\sigma_{\mathfrak{l}}^2}{\sigma_{\mathfrak{s}}^2} \overset{(9),(12)}{\geq} \frac{d^{\frac{2}{\mathfrak{p}+\epsilon}}\sigma^2 / 18^2 \frac{\mathfrak{p}^3}{\mathfrak{p}-1}}{2^{\frac{4}{\mathfrak{p}}-2}d^{\frac{2}{\mathfrak{p}}-1}\sigma^2} = \frac{(\mathfrak{p}-1)d^{1-\frac{2\epsilon}{\mathfrak{p}(\mathfrak{p}+\epsilon)}}}{\mathfrak{p}^3 3^4 2^{\frac{4}{\mathfrak{p}}}}. \tag{13}$$

In particular, for any $0 < \epsilon \leq \min\left\{\frac{\mathfrak{p}}{2\ln d - 1}, 2-\mathfrak{p}\right\}$ (assume $d \geq 2$ here, since the case $d = 1$ is trivial),

$$\frac{2\epsilon}{\mathfrak{p}(\mathfrak{p}+\epsilon)} \leq \frac{1}{\mathfrak{p}\ln d} \Rightarrow d^{\frac{2\epsilon}{\mathfrak{p}(\mathfrak{p}+\epsilon)}} \leq e^{\frac{1}{\mathfrak{p}}} \Rightarrow d_{\text{eff}} \overset{(13)}{\geq} \frac{(\mathfrak{p}-1)d}{\mathfrak{p}^3 3^4 2^{\frac{4}{\mathfrak{p}}}e^{\frac{1}{\mathfrak{p}}}} = \Omega(d). \tag{14}$$

## B REDUCTION FOR STRONGLY CONVEX PROBLEMS

We provide the reduction mentioned in Remark 5. Recall that existing works assume $f$ being $\mu$-strongly convex and $G$-Lipschitz with a minimizer $\mathbf{x}_\star$ on $\mathbb{X}$. Now we consider the following problem instance to fit into our problem structure

$$F(\mathbf{x}) = \underbrace{f(\mathbf{x}) - \frac{\mu}{2}\|\mathbf{x}-\mathbf{y}\|^2}_{\triangleq \bar{f}(\mathbf{x})} + \underbrace{\frac{\mu}{2}\|\mathbf{x}-\mathbf{y}\|^2}_{\triangleq r(\mathbf{x})} = f(\mathbf{x}),$$

where $\mathbf{y}$ can be any known point in $\mathbb{X}$. For example, one can set $\mathbf{y} = \mathbf{x}_1$ to be the initial point. Next, we show that $F$ fulfills all assumptions in Section 2.

- $F$ on $\mathbb{X}$ has the same optimal solution $\mathbf{x}_\star$ as $f$ and hence satisfies Assumption 1.
- Note that $\bar{f}$ is convex (since $f$ is $\mu$-strongly convex) and $r$ is $\mu$-strongly convex, which fits Assumption 2.
- Moreover, because $f$ is $\mu$-strongly convex and $G$-Lipschitz with a minimizer $\mathbf{x}_\star \in \mathbb{X}$, a well-known fact is that $\mathbb{X}$ has to be bounded, since for any $\mathbf{x} \in \mathbb{X}$,

$$\frac{\mu}{2}\|\mathbf{x}-\mathbf{x}_\star\|^2 \leq f(\mathbf{x}) - f(\mathbf{x}_\star) - \langle\nabla f(\mathbf{x}_\star), \mathbf{x}-\mathbf{x}_\star\rangle \leq f(\mathbf{x}) - f(\mathbf{x}_\star)$$

$$\leq \langle\nabla f(\mathbf{x}), \mathbf{x}-\mathbf{x}_\star\rangle \leq \|\nabla f(\mathbf{x})\|\|\mathbf{x}-\mathbf{x}_\star\| \leq G\|\mathbf{x}-\mathbf{x}_\star\|$$

$$\Rightarrow \|\mathbf{x}-\mathbf{x}_\star\| \leq \frac{2G}{\mu}. \tag{15}$$

Then we can calculate $\nabla \bar{f}(\mathbf{x}) = \nabla f(\mathbf{x}) - \mu(\mathbf{x} - \mathbf{y}), \forall \mathbf{x} \in \mathbb{X}$ and find $\left\|\nabla \bar{f}(\mathbf{x})\right\| \leq$

$\left\|\nabla f(\mathbf{x})\right\| + \mu \left\|\mathbf{x} - \mathbf{x}_\star\right\| + \mu \left\|\mathbf{y} - \mathbf{x}_\star\right\| \overset{(15)}{\leq} 5G, \forall \mathbf{x} \in \mathbb{X}$, meaning that Assumption 3 holds under the parameter $5G$.

- In addition, suppose we have a first-order oracle $\mathbf{g}(\mathbf{x}, \xi)$ for $\nabla f$ satisfying Assumption 4. Then $\bar{\mathbf{g}}(\mathbf{x}, \xi) \triangleq \mathbf{g}(\mathbf{x}, \xi) - \mu(\mathbf{x} - \mathbf{y})$ is a first-order oracle for $\bar{f}$ satisfying Assumption 4 with the same parameters $\mathfrak{p}, \sigma_{\mathfrak{s}}$ and $\sigma_{\mathfrak{l}}$.

Therefore, any instance in existing works can be transferred to fit our problem structure. Moreover, for such an instance, we have $D = \left\|\mathbf{x}_1 - \mathbf{x}_\star\right\| \overset{(15)}{\leq} \frac{2G}{\mu}$, implying that the first term $\mathcal{O}\left(\frac{\mu D^2}{T^3}\right)$ in Theorem 2 is at most $\mathcal{O}\left(\frac{G^2}{\mu T^3}\right)$, which can be further bounded by the third term $\mathcal{O}\left(\frac{G^2}{\mu T}\right)$. So $\mathcal{O}\left(\frac{\mu D^2}{T^3}\right)$ in Theorem 2 can be omitted if compared with prior works.

*Remark* 10. The above reduction does not hold in the reverse direction. This is because, as one can see, the domain $\mathbb{X}$ in prior works has to be bounded (due to (15)), which is however not necessary under our problem structure. For example, $\mathbb{X}$ in our problem can take $\mathbb{R}^d$, which cannot be true for previous works in contrast. In other words, the problem studied in our paper is strictly more general.

## C STABILIZED CLIPPED STOCHASTIC GRADIENT DESCENT

---

**Algorithm 2** Stabilized Clipped Stochastic Gradient Descent (Stabilized Clipped SGD)

---

**Input:** initial point $\mathbf{x}_1 \in \mathbb{X}$, stepsize $\eta_t > 0$, clipping threshold $\tau_t > 0$
**for** $t = 1$ **to** $T$ **do**
$\quad \mathbf{g}_t^c = \mathrm{clip}_{\tau_t}(\mathbf{g}_t)$ where $\mathbf{g}_t = \mathbf{g}(\mathbf{x}_t, \xi_t)$ and $\xi_t \sim \mathbb{D}$ is sampled independently from the history
$\quad \mathbf{x}_{t+1} = \mathrm{argmin}_{\mathbf{x} \in \mathbb{X}} r(\mathbf{x}) + \langle \mathbf{g}_t^c, \mathbf{x} \rangle + \frac{\left\|\mathbf{x} - \mathbf{x}_t\right\|^2}{2\eta_t} + \frac{(\eta_t/\eta_{t+1} - 1)\left\|\mathbf{x} - \mathbf{x}_1\right\|^2}{2\eta_t}$
**end for**

---

In this section, we propose Stabilized Clipped Stochastic Gradient Descent (Stabilized Clipped SGD) in Algorithm 2, an algorithmic variant of Clipped SGD to deal with the undesired $\mathrm{poly}(\ln T)$ factor appearing in the anytime convergence rate of Clipped SGD for general convex functions.

Compared to Clipped SGD, the only difference is an extra $\frac{(\eta_t/\eta_{t+1} - 1)\left\|\mathbf{x} - \mathbf{x}_1\right\|^2}{2\eta_t}$ term injected into the update rule, which is borrowed from the dual stabilization technique introduced by Fang et al. (2022). The stabilization trick was originally introduced to make Online Mirror Descent (Nemirovski & Yudin, 1983; Warmuth et al., 1997; Beck & Teboulle, 2003) achieve an anytime optimal $\mathcal{O}(\sqrt{T})$ regret on unbounded domains without knowing $T$. For how it works and the intuition behind this mechanism, we kindly refer the reader to Fang et al. (2022) for details. Inspired by its anytime optimality, we incorporate it with Clipped SGD here and will show that this stabilized modification also works well under heavy-tailed noise. Precisely, assuming all problem-dependent parameters are known but not $T$, we prove in Theorem 11 that Stabilized Clipped SGD converges at an anytime rate almost identical (though slightly different) to the bound for Clipped SGD given in Theorem 1 that requires a known $T$ in contrast.

Lastly, we remark that when the stepsize $\eta_t$ is constant, Stabilized Clipped SGD and Clipped SGD degenerate to the same algorithm. Therefore, Theorems 1 and 3 can directly apply to Stabilized Clipped SGD as well. For the same reason and also to save space, we will only analyze Stabilized Clipped SGD when studying general convex functions.

## D FINER BOUNDS FOR CLIPPING ERROR UNDER HEAVY-TAILED NOISE

In this section, we study the clipping error under heavy-tailed noise, whose finer bounds are critical in the analysis. Moreover, instead of limiting to clipped gradient methods, we will study a more general setting as in the following Theorem 9, which may benefit broader research. In Appendix F,

we apply this general result to prove clipping error bounds specialized for clipped gradient methods in Lemma 2, which is the full statement of Lemma 1.

**Theorem 9.** *Given a $\sigma$-algebra $\mathcal{F}$ and two random vectors $\mathbf{g}, \mathbf{f} \in \mathbb{R}^d$, suppose they satisfy $\mathbb{E}[\mathbf{g} \mid \mathcal{F}] = \mathbf{f}$ and, for some $\mathfrak{p} \in (1, 2]$ and two constants $\sigma_{\mathfrak{s}}, \sigma_{\mathfrak{l}} \geq 0$,*

$$\mathbb{E}\left[\|\mathbf{g} - \mathbf{f}\|^{\mathfrak{p}} \mid \mathcal{F}\right] \leq \sigma_{\mathfrak{l}}^{\mathfrak{p}}, \quad \mathbb{E}\left[|\langle \mathbf{e}, \mathbf{g} - \mathbf{f}\rangle|^{\mathfrak{p}} \mid \mathcal{F}\right] \leq \sigma_{\mathfrak{s}}^{\mathfrak{p}}, \quad \forall \mathbf{e} \in \mathbb{S}^{d-1}. \tag{16}$$

*Moreover, we assume there exists another random vector $\bar{\mathbf{g}} \in \mathbb{R}^d$ that is independent of $\mathbf{g}$ conditional on $\mathcal{F}$ and satisfies that $\bar{\mathbf{g}} \mid \mathcal{F}$ equals $\mathbf{g} \mid \mathcal{F}$ in distribution. For any $0 < \tau \in \mathcal{F}$, let $\mathbf{g}^{\mathrm{c}} \triangleq \mathrm{clip}_\tau(\mathbf{g}) = \min\left\{1, \frac{\tau}{\|\mathbf{g}\|}\right\}\mathbf{g}$, $\mathbf{d}^{\mathrm{u}} \triangleq \mathbf{g}^{\mathrm{c}} - \mathbb{E}[\mathbf{g}^{\mathrm{c}} \mid \mathcal{F}]$, $\mathbf{d}^{\mathrm{b}} \triangleq \mathbb{E}[\mathbf{g}^{\mathrm{c}} \mid \mathcal{F}] - \mathbf{f}$, and $\chi(\alpha) \triangleq \mathbb{1}\left[(1 - \alpha)\tau \geq \|\mathbf{f}\|\right], \forall \alpha \in [0, 1)$, then there are:*

1. $\|\mathbf{d}^{\mathrm{u}}\| \leq 2\tau$.

2. $\mathbb{E}\left[\|\mathbf{d}^{\mathrm{u}}\|^2 \mid \mathcal{F}\right] \leq 4\sigma_{\mathfrak{l}}^{\mathfrak{p}}\tau^{2-\mathfrak{p}}$.

3. $\left\|\mathbb{E}\left[\mathbf{d}^{\mathrm{u}}(\mathbf{d}^{\mathrm{u}})^\top \mid \mathcal{F}\right]\right\| \leq 4\sigma_{\mathfrak{s}}^{\mathfrak{p}}\tau^{2-\mathfrak{p}} + 4\|\mathbf{f}\|^2$.

4. $\left\|\mathbb{E}\left[\mathbf{d}^{\mathrm{u}}(\mathbf{d}^{\mathrm{u}})^\top \mid \mathcal{F}\right]\right\|\chi(\alpha) \leq 4\sigma_{\mathfrak{s}}^{\mathfrak{p}}\tau^{2-\mathfrak{p}} + 4\alpha^{1-\mathfrak{p}}\sigma_{\mathfrak{l}}^{\mathfrak{p}}\|\mathbf{f}\|^2\tau^{-\mathfrak{p}}$.

5. $\left\|\mathbf{d}^{\mathrm{b}}\right\| \leq \sqrt{2}\left(\sigma_{\mathfrak{l}}^{\mathfrak{p}-1} + \|\mathbf{f}\|^{\mathfrak{p}-1}\right)\sigma_{\mathfrak{s}}\tau^{1-\mathfrak{p}} + 2\left(\sigma_{\mathfrak{l}}^{\mathfrak{p}} + \|\mathbf{f}\|^{\mathfrak{p}}\right)\|\mathbf{f}\|\tau^{-\mathfrak{p}}$.

6. $\left\|\mathbf{d}^{\mathrm{b}}\right\|\chi(\alpha) \leq \sigma_{\mathfrak{s}}\sigma_{\mathfrak{l}}^{\mathfrak{p}-1}\tau^{1-\mathfrak{p}} + \alpha^{1-\mathfrak{p}}\sigma_{\mathfrak{l}}^{\mathfrak{p}}\|\mathbf{f}\|\tau^{-\mathfrak{p}}$.

Before proving Theorem 9, we discuss one point here. As one can see, we require the existence of a random vector $\bar{\mathbf{g}} \in \mathbb{R}^d$ satisfying a certain condition. This technical assumption is mild as it can hold automatically in many cases. For example, if $\mathcal{F}$ is the trivial sigma algebra, then we can set $\bar{\mathbf{g}}$ as an independent copy of $\mathbf{g}$. For clipped gradient methods under Assumption 4, suppose $\mathcal{F} = \mathcal{F}_{t-1}$, $\mathbf{g} = \mathbf{g}(\mathbf{x}_t, \xi_t)$ and $\mathbf{f} = \nabla f(\mathbf{x}_t)$, then we can set $\bar{\mathbf{g}} = \mathbf{g}(\mathbf{x}_t, \xi_{t+1})$, where we recall $\mathcal{F}_{t-1} = \sigma(\xi_1, \cdots, \xi_{t-1})$ and $\xi_1$ to $\xi_{t+1}$ are sampled from $\mathbb{D}$ independently.

*Proof.* Inspired by Das et al. (2024), we denote by $h \triangleq \min\left\{1, \frac{\tau}{\|\mathbf{g}\|}\right\} \in [0, 1]$. Under this notation, we have

$$\mathbf{g}^{\mathrm{c}} = \mathrm{clip}_\tau(\mathbf{g}) = h\mathbf{g}. \tag{17}$$

We first give two useful properties of $h$.

- For any $q \geq 0$, we have

$$1 - h \leq \frac{\|\mathbf{g}\|^q}{\|\mathbf{g}\|^q}\mathbb{1}\left[\|\mathbf{g}\| \geq \tau\right] \leq \frac{\|\mathbf{g}\|^q}{\tau^q}\mathbb{1}\left[\|\mathbf{g}\| \geq \tau\right] \leq \frac{\|\mathbf{g}\|^q}{\tau^q},$$

  which implies

$$1 - h \leq \inf_{q \geq 0}\frac{\|\mathbf{g}\|^q}{\tau^q}. \tag{18}$$

- We can also observe

$$1 - h = \frac{\|\mathbf{g}\| - \tau}{\|\mathbf{g}\|}\mathbb{1}\left[\|\mathbf{g}\| \geq \tau\right] \leq \frac{\|\mathbf{g}\| - \tau}{\tau}\mathbb{1}\left[\|\mathbf{g}\| \geq \tau\right]$$

$$\leq \frac{\|\mathbf{g} - \mathbf{f}\| + \|\mathbf{f}\| - \tau}{\tau}\mathbb{1}\left[\|\mathbf{g}\| \geq \tau\right],$$

  which implies

$$(1 - h)\chi(\alpha) \leq \frac{\|\mathbf{g} - \mathbf{f}\| + \|\mathbf{f}\| - \tau}{\tau}\mathbb{1}\left[\|\mathbf{g}\| \geq \tau \geq \frac{\|\mathbf{f}\|}{1 - \alpha}\right]$$

$$\leq \frac{\|\mathbf{g} - \mathbf{f}\|}{\tau}\mathbb{1}\left[\|\mathbf{g}\| \geq \tau \geq \frac{\|\mathbf{f}\|}{1 - \alpha}\right] \leq \inf_{q \geq 1}\frac{\|\mathbf{g} - \mathbf{f}\|^q}{\alpha^{q-1}\tau^q}\chi(\alpha), \tag{19}$$

where the last step is by noticing that the event $\left\{\|\mathbf{g}\| \geq \tau \geq \frac{\|\mathbf{f}\|}{1-\alpha}\right\}$ implies the event $\left\{\tau \geq \frac{\|\mathbf{f}\|}{1-\alpha}, \|\mathbf{g} - \mathbf{f}\| \geq \alpha\tau\right\}$, thereby leading to, for any $q \geq 1$,

$$
\begin{aligned}
\frac{\|\mathbf{g} - \mathbf{f}\|}{\tau} \mathbb{1}\left[\|\mathbf{g}\| \geq \tau \geq \frac{\|\mathbf{f}\|}{1-\alpha}\right] &\leq \frac{\|\mathbf{g} - \mathbf{f}\|}{\tau} \mathbb{1}\left[\tau \geq \frac{\|\mathbf{f}\|}{1-\alpha}, \|\mathbf{g} - \mathbf{f}\| \geq \alpha\tau\right] \\
&\leq \frac{\|\mathbf{g} - \mathbf{f}\|^q}{\alpha^{q-1}\tau^q} \mathbb{1}\left[\tau \geq \frac{\|\mathbf{f}\|}{1-\alpha}, \|\mathbf{g} - \mathbf{f}\| \geq \alpha\tau\right] \\
&\leq \frac{\|\mathbf{g} - \mathbf{f}\|^q}{\alpha^{q-1}\tau^q}\chi(\alpha).
\end{aligned}
$$

For $\bar{\mathbf{g}}$, we use $\bar{\mathbf{g}}^c$ to denote the clipped version of $\bar{\mathbf{g}}$ under the same clipping threshold $\tau$, i.e., $\bar{\mathbf{g}}^c \triangleq \mathrm{clip}_\tau(\bar{\mathbf{g}}) = \min\left\{1, \frac{\tau}{\|\bar{\mathbf{g}}\|}\right\}\bar{\mathbf{g}}$. By our assumption on $\bar{\mathbf{g}}$, the following results hold

$$
\mathbb{E}\left[\mathbf{g}^c \mid \mathcal{F}\right] = \mathbb{E}\left[\bar{\mathbf{g}}^c \mid \mathcal{F}\right] = \mathbb{E}\left[\bar{\mathbf{g}}^c \mid \mathcal{F}, \mathbf{g}\right], \tag{20}
$$

$$
\mathbb{E}\left[\|\mathbf{g} - \mathbf{f}\|^{\mathfrak{p}} \mid \mathcal{F}\right] = \mathbb{E}\left[\|\bar{\mathbf{g}} - \mathbf{f}\|^{\mathfrak{p}} \mid \mathcal{F}\right] \leq \sigma_{\mathfrak{l}}^{\mathfrak{p}}. \tag{21}
$$

We first prove inequalities for $\mathbf{d}^{\mathrm{u}}$.

- Inequality 1. Note that $\|\mathbf{g}^c\| \leq \tau$, implying $\|\mathbf{d}^{\mathrm{u}}\| = \|\mathbf{g}^c - \mathbb{E}\left[\mathbf{g}^c \mid \mathcal{F}\right]\| \leq 2\tau$.

- Inequality 2. We observe that

$$
\begin{aligned}
\mathbb{E}\left[\|\mathbf{d}^{\mathrm{u}}\|^2 \mid \mathcal{F}\right] &= \mathbb{E}\left[\|\mathbf{g}^c - \mathbb{E}\left[\mathbf{g}^c \mid \mathcal{F}\right]\|^2 \mid \mathcal{F}\right] \overset{(20)}{=} \mathbb{E}\left[\|\mathbb{E}\left[\mathbf{g}^c - \bar{\mathbf{g}}^c \mid \mathcal{F}, \mathbf{g}\right]\|^2 \mid \mathcal{F}\right] \\
&\overset{(a)}{\leq} \mathbb{E}\left[\|\mathbf{g}^c - \bar{\mathbf{g}}^c\|^2 \mid \mathcal{F}\right] \leq (2\tau)^{2-\mathfrak{p}}\mathbb{E}\left[\|\mathbf{g}^c - \bar{\mathbf{g}}^c\|^{\mathfrak{p}} \mid \mathcal{F}\right] \\
&\overset{(b)}{\leq} (2\tau)^{2-\mathfrak{p}}\mathbb{E}\left[\|\mathbf{g} - \bar{\mathbf{g}}\|^{\mathfrak{p}} \mid \mathcal{F}\right] \overset{(c)}{\leq} 4\sigma_{\mathfrak{l}}^{\mathfrak{p}}\tau^{2-\mathfrak{p}},
\end{aligned} \tag{22}
$$

where $(a)$ is by the convexity of $\|\cdot\|^2$ and the tower property, $(b)$ holds because $\mathrm{clip}_\tau$ is a nonexpansive mapping, and $(c)$ follows by when $\mathfrak{p} > 1$

$$
\|\mathbf{g} - \bar{\mathbf{g}}\|^{\mathfrak{p}} \leq 2^{\mathfrak{p}-1}\left(\|\mathbf{g} - \mathbf{f}\|^{\mathfrak{p}} + \|\bar{\mathbf{g}} - \mathbf{f}\|^{\mathfrak{p}}\right) \Rightarrow \mathbb{E}\left[\|\mathbf{g} - \bar{\mathbf{g}}\|^{\mathfrak{p}} \mid \mathcal{F}\right] \overset{(16),(21)}{\leq} 2^{\mathfrak{p}}\sigma_{\mathfrak{l}}^{\mathfrak{p}}.
$$

The third and fourth inequalities are more technical. Let $\mathbf{e} \in \mathbb{S}^{d-1}$ be a unit vector, we know

$$
\mathbf{e}^\top\mathbb{E}\left[\mathbf{d}^{\mathrm{u}}\left(\mathbf{d}^{\mathrm{u}}\right)^\top \mid \mathcal{F}\right]\mathbf{e} = \mathbb{E}\left[|\langle \mathbf{e}, \mathbf{d}^{\mathrm{u}}\rangle|^2 \mid \mathcal{F}\right] = \mathbb{E}\left[|\langle \mathbf{e}, \mathbf{g}^c - \mathbb{E}\left[\mathbf{g}^c \mid \mathcal{F}\right]\rangle|^2 \mid \mathcal{F}\right]. \tag{23}
$$

We will bound this term in two approaches.

On the one hand, we have

$$
\begin{aligned}
&\mathbb{E}\left[|\langle \mathbf{e}, \mathbf{g}^c - \mathbb{E}\left[\mathbf{g}^c \mid \mathcal{F}\right]\rangle|^2 \mid \mathcal{F}\right] \\
=&\mathbb{E}\left[|\langle \mathbf{e}, \mathbf{g}^c - \mathbf{f}\rangle|^2 \mid \mathcal{F}\right] - \mathbb{E}\left[|\langle \mathbf{e}, \mathbf{f} - \mathbb{E}\left[\mathbf{g}^c \mid \mathcal{F}\right]\rangle|^2 \mid \mathcal{F}\right] \\
\leq&\mathbb{E}\left[|\langle \mathbf{e}, \mathbf{g}^c - \mathbf{f}\rangle|^2 \mid \mathcal{F}\right] \overset{(17)}{=} \mathbb{E}\left[|\langle \mathbf{e}, h\mathbf{g} - \mathbf{f}\rangle|^2 \mid \mathcal{F}\right] \\
=&\mathbb{E}\left[|h\langle \mathbf{e}, \mathbf{g} - \mathbf{f}\rangle - (1-h)\langle \mathbf{e}, \mathbf{f}\rangle|^2 \mid \mathcal{F}\right] \leq \mathbb{E}\left[h|\langle \mathbf{e}, \mathbf{g} - \mathbf{f}\rangle|^2 + (1-h)|\langle \mathbf{e}, \mathbf{f}\rangle|^2 \mid \mathcal{F}\right] \\
\leq&\mathbb{E}\left[h|\langle \mathbf{e}, \mathbf{g} - \mathbf{f}\rangle|^2 + (1-h)\|\mathbf{f}\|^2 \mid \mathcal{F}\right], \tag{24}
\end{aligned}
$$

where the last step is by $|\langle \mathbf{e}, \mathbf{f}\rangle| \leq \|\mathbf{e}\|\|\mathbf{f}\| = \|\mathbf{f}\|$ and $1 - h \geq 0$. By Cauchy-Schwarz inequality again,

$$
|\langle \mathbf{e}, \mathbf{g} - \mathbf{f}\rangle|^{2-\mathfrak{p}} \leq \|\mathbf{g} - \mathbf{f}\|^{2-\mathfrak{p}} \leq (\|\mathbf{g}\| + \|\mathbf{f}\|)^{2-\mathfrak{p}} \overset{\mathfrak{p}>1}{\leq} \|\mathbf{g}\|^{2-\mathfrak{p}} + \|\mathbf{f}\|^{2-\mathfrak{p}}
$$

$$
\Rightarrow h|\langle \mathbf{e}, \mathbf{g} - \mathbf{f}\rangle|^{2-\mathfrak{p}} \leq h^{\mathfrak{p}-1}\|h\mathbf{g}\|^{2-\mathfrak{p}} + h\|\mathbf{f}\|^{2-\mathfrak{p}} \overset{h\leq 1<\mathfrak{p}}{\leq} \|h\mathbf{g}\|^{2-\mathfrak{p}} + \|\mathbf{f}\|^{2-\mathfrak{p}}
$$

$$
\overset{(17)}{=} \|\mathbf{g}^c\|^{2-\mathfrak{p}} + \|\mathbf{f}\|^{2-\mathfrak{p}} \leq \tau^{2-\mathfrak{p}} + \|\mathbf{f}\|^{2-\mathfrak{p}},
$$

which implies

$$\mathbb{E}\left[h\left|\langle \mathbf{e}, \mathbf{g}-\mathbf{f}\rangle\right|^2 \mid \mathcal{F}\right] \leq \left(\tau^{2-\mathfrak{p}}+\|\mathbf{f}\|^{2-\mathfrak{p}}\right)\mathbb{E}\left[\left|\langle \mathbf{e}, \mathbf{g}-\mathbf{f}\rangle\right|^{\mathfrak{p}} \mid \mathcal{F}\right] \overset{(16)}{\leq} \sigma_{\mathfrak{s}}^{\mathfrak{p}}\tau^{2-\mathfrak{p}}+\sigma_{\mathfrak{s}}^{\mathfrak{p}}\|\mathbf{f}\|^{2-\mathfrak{p}}. \quad (25)$$

Combine (23), (24) and (25) to obtain for any unit vector $\mathbf{e} \in \mathbb{S}^{d-1}$,

$$\mathbf{e}^{\top}\mathbb{E}\left[\mathbf{d}^{\mathrm{u}}\left(\mathbf{d}^{\mathrm{u}}\right)^{\top} \mid \mathcal{F}\right]\mathbf{e} \leq \sigma_{\mathfrak{s}}^{\mathfrak{p}}\tau^{2-\mathfrak{p}}+\sigma_{\mathfrak{s}}^{\mathfrak{p}}\|\mathbf{f}\|^{2-\mathfrak{p}}+\|\mathbf{f}\|^2\mathbb{E}\left[1-h \mid \mathcal{F}\right]$$
$$\Rightarrow \left\|\mathbb{E}\left[\mathbf{d}^{\mathrm{u}}\left(\mathbf{d}^{\mathrm{u}}\right)^{\top} \mid \mathcal{F}\right]\right\| \leq \sigma_{\mathfrak{s}}^{\mathfrak{p}}\tau^{2-\mathfrak{p}}+\sigma_{\mathfrak{s}}^{\mathfrak{p}}\|\mathbf{f}\|^{2-\mathfrak{p}}+\|\mathbf{f}\|^2\mathbb{E}\left[1-h \mid \mathcal{F}\right]. \quad (26)$$

On the other hand, we can follow a similar way of proving (22) to show

$$\mathbb{E}\left[\left|\langle \mathbf{e}, \mathbf{g}^{\mathrm{c}} - \mathbb{E}\left[\mathbf{g}^{\mathrm{c}} \mid \mathcal{F}\right]\rangle\right|^2 \mid \mathcal{F}\right] \leq (2\tau)^{2-\mathfrak{p}}\mathbb{E}\left[\left|\langle \mathbf{e}, \mathbf{g}^{\mathrm{c}} - \bar{\mathbf{g}}^{\mathrm{c}}\rangle\right|^{\mathfrak{p}} \mid \mathcal{F}\right]$$
$$\leq 4\tau^{2-\mathfrak{p}}\mathbb{E}\left[\left|\langle \mathbf{e}, \mathbf{g}^{\mathrm{c}} - \mathbf{f}\rangle\right|^{\mathfrak{p}} \mid \mathcal{F}\right]. \quad (27)$$

Similar to (24), there is

$$\mathbb{E}\left[\left|\langle \mathbf{e}, \mathbf{g}^{\mathrm{c}} - \mathbf{f}\rangle\right|^{\mathfrak{p}} \mid \mathcal{F}\right] \leq \mathbb{E}\left[h\left|\langle \mathbf{e}, \mathbf{g}-\mathbf{f}\rangle\right|^{\mathfrak{p}}+(1-h)\|\mathbf{f}\|^{\mathfrak{p}} \mid \mathcal{F}\right]$$
$$\overset{h\leq 1}{\leq} \mathbb{E}\left[\left|\langle \mathbf{e}, \mathbf{g}-\mathbf{f}\rangle\right|^{\mathfrak{p}}+(1-h)\|\mathbf{f}\|^{\mathfrak{p}} \mid \mathcal{F}\right]$$
$$\overset{(16)}{\leq} \sigma_{\mathfrak{s}}^{\mathfrak{p}}+\|\mathbf{f}\|^{\mathfrak{p}}\mathbb{E}\left[1-h \mid \mathcal{F}\right]. \quad (28)$$

Combine (23), (27) and (28) to obtain for any unit vector $\mathbf{e} \in \mathbb{S}^{d-1}$,

$$\mathbf{e}^{\top}\mathbb{E}\left[\mathbf{d}^{\mathrm{u}}\left(\mathbf{d}^{\mathrm{u}}\right)^{\top} \mid \mathcal{F}\right]\mathbf{e} \leq 4\sigma_{\mathfrak{s}}^{\mathfrak{p}}\tau^{2-\mathfrak{p}}+4\tau^{2-\mathfrak{p}}\|\mathbf{f}\|^{\mathfrak{p}}\mathbb{E}\left[1-h \mid \mathcal{F}\right]$$
$$\Rightarrow \left\|\mathbb{E}\left[\mathbf{d}^{\mathrm{u}}\left(\mathbf{d}^{\mathrm{u}}\right)^{\top} \mid \mathcal{F}\right]\right\| \leq 4\sigma_{\mathfrak{s}}^{\mathfrak{p}}\tau^{2-\mathfrak{p}}+4\tau^{2-\mathfrak{p}}\|\mathbf{f}\|^{\mathfrak{p}}\mathbb{E}\left[1-h \mid \mathcal{F}\right]. \quad (29)$$

Recall by our definition $\chi(0) = \mathbb{1}\left[\tau \geq \|\mathbf{f}\|\right]$, we then denote by $\bar{\chi}(0) \triangleq 1 - \chi(0) = \mathbb{1}\left[\tau < \|\mathbf{f}\|\right]$. Therefore,

$$\left\|\mathbb{E}\left[\mathbf{d}^{\mathrm{u}}\left(\mathbf{d}^{\mathrm{u}}\right)^{\top} \mid \mathcal{F}\right]\right\|\chi(0) \overset{(26)}{\leq} \left(\sigma_{\mathfrak{s}}^{\mathfrak{p}}\tau^{2-\mathfrak{p}}+\sigma_{\mathfrak{s}}^{\mathfrak{p}}\|\mathbf{f}\|^{2-\mathfrak{p}}+\|\mathbf{f}\|^2\mathbb{E}\left[1-h \mid \mathcal{F}\right]\right)\chi(0)$$
$$\overset{\mathfrak{p}\leq 2}{\leq} \left(2\sigma_{\mathfrak{s}}^{\mathfrak{p}}\tau^{2-\mathfrak{p}}+\|\mathbf{f}\|^2\mathbb{E}\left[1-h \mid \mathcal{F}\right]\right)\chi(0),$$

and

$$\left\|\mathbb{E}\left[\mathbf{d}^{\mathrm{u}}\left(\mathbf{d}^{\mathrm{u}}\right)^{\top} \mid \mathcal{F}\right]\right\|\bar{\chi}(0) \overset{(29)}{\leq} \left(4\sigma_{\mathfrak{s}}^{\mathfrak{p}}\tau^{2-\mathfrak{p}}+4\tau^{2-\mathfrak{p}}\|\mathbf{f}\|^{\mathfrak{p}}\mathbb{E}\left[1-h \mid \mathcal{F}\right]\right)\bar{\chi}(0)$$
$$\overset{\mathfrak{p}\leq 2}{\leq} \left(4\sigma_{\mathfrak{s}}^{\mathfrak{p}}\tau^{2-\mathfrak{p}}+4\|\mathbf{f}\|^2\mathbb{E}\left[1-h \mid \mathcal{F}\right]\right)\bar{\chi}(0),$$

which together imply

$$\left\|\mathbb{E}\left[\mathbf{d}^{\mathrm{u}}\left(\mathbf{d}^{\mathrm{u}}\right)^{\top} \mid \mathcal{F}\right]\right\| \leq 4\sigma_{\mathfrak{s}}^{\mathfrak{p}}\tau^{2-\mathfrak{p}}+4\|\mathbf{f}\|^2\mathbb{E}\left[1-h \mid \mathcal{F}\right]. \quad (30)$$

Now we are ready to prove inequalities 3 and 4.

- Inequality 3. We use (30) to know
$$\left\|\mathbb{E}\left[\mathbf{d}^{\mathrm{u}}\left(\mathbf{d}^{\mathrm{u}}\right)^{\top} \mid \mathcal{F}\right]\right\| \leq 4\sigma_{\mathfrak{s}}^{\mathfrak{p}}\tau^{2-\mathfrak{p}}+4\|\mathbf{f}\|^2\mathbb{E}\left[1-h \mid \mathcal{F}\right] \leq 4\sigma_{\mathfrak{s}}^{\mathfrak{p}}\tau^{2-\mathfrak{p}}+4\|\mathbf{f}\|^2.$$

- Inequality 4. By (19), we have
$$(1-h)\chi(\alpha) \leq \frac{\|\mathbf{g}-\mathbf{f}\|^{\mathfrak{p}}}{\alpha^{\mathfrak{p}-1}\tau^{\mathfrak{p}}}\chi(\alpha)$$
$$\Rightarrow \mathbb{E}\left[1-h \mid \mathcal{F}\right]\chi(\alpha) \overset{\chi(\alpha)\in\mathcal{F}}{=} \mathbb{E}\left[(1-h)\chi(\alpha) \mid \mathcal{F}\right] \leq \mathbb{E}\left[\frac{\|\mathbf{g}-\mathbf{f}\|^{\mathfrak{p}}}{\alpha^{\mathfrak{p}-1}\tau^{\mathfrak{p}}}\chi(\alpha) \mid \mathcal{F}\right]$$
$$\overset{(16)}{\leq} \frac{\sigma_{\mathfrak{l}}^{\mathfrak{p}}\chi(\alpha)}{\alpha^{\mathfrak{p}-1}\tau^{\mathfrak{p}}} \leq \frac{\sigma_{\mathfrak{l}}^{\mathfrak{p}}}{\alpha^{\mathfrak{p}-1}\tau^{\mathfrak{p}}}.$$

Now we use (30) to know

$$\left\| \mathbb{E}\left[\mathbf{d}^{\mathrm{u}}\left(\mathbf{d}^{\mathrm{u}}\right)^{\top} \mid \mathcal{F}\right]\right\| \chi(\alpha) \leq 4\sigma_{\mathfrak{s}}^{\mathfrak{p}}\tau^{2-\mathfrak{p}}\chi(\alpha) + 4\left\|\mathbf{f}\right\|^{2}\mathbb{E}\left[1-h \mid \mathcal{F}\right]\chi(\alpha)$$
$$\leq 4\sigma_{\mathfrak{s}}^{\mathfrak{p}}\tau^{2-\mathfrak{p}} + 4\alpha^{1-\mathfrak{p}}\sigma_{\mathfrak{l}}^{\mathfrak{p}}\left\|\mathbf{f}\right\|^{2}\tau^{-\mathfrak{p}}.$$

Finally, we prove the last two inequalities related to $\mathbf{d}^{\mathrm{b}}$. Still let $\mathbf{e}$ represent a unit vector in $\mathbb{R}^{d}$, then by the definition of $\mathbf{d}^{\mathrm{b}}$,

$$\left\langle\mathbf{e},\mathbf{d}^{\mathrm{b}}\right\rangle \overset{(17)}{=} \left\langle\mathbf{e},\mathbb{E}\left[h\mathbf{g} \mid \mathcal{F}\right]-\mathbf{f}\right\rangle = \mathbb{E}\left[(h-1)\left\langle\mathbf{e},\mathbf{g}\right\rangle \mid \mathcal{F}\right]$$
$$= \mathbb{E}\left[(h-1)\left\langle\mathbf{e},\mathbf{g}-\mathbf{f}\right\rangle \mid \mathcal{F}\right] - \left\langle\mathbf{e},\mathbf{f}\right\rangle\mathbb{E}\left[1-h \mid \mathcal{F}\right]$$
$$\overset{(d)}{\leq} \mathbb{E}\left[(1-h)\left|\left\langle\mathbf{e},\mathbf{g}-\mathbf{f}\right\rangle\right| \mid \mathcal{F}\right] + \left\|\mathbf{f}\right\|\mathbb{E}\left[1-h \mid \mathcal{F}\right]$$
$$\overset{(e)}{\leq} \left(\mathbb{E}\left[(1-h)^{\frac{\mathfrak{p}}{\mathfrak{p}-1}} \mid \mathcal{F}\right]\right)^{1-\frac{1}{\mathfrak{p}}}\sigma_{\mathfrak{s}} + \left\|\mathbf{f}\right\|\mathbb{E}\left[1-h \mid \mathcal{F}\right]$$
$$\Rightarrow \left\|\mathbf{d}^{\mathrm{b}}\right\| \leq \left(\mathbb{E}\left[(1-h)^{\frac{\mathfrak{p}}{\mathfrak{p}-1}} \mid \mathcal{F}\right]\right)^{1-\frac{1}{\mathfrak{p}}}\sigma_{\mathfrak{s}} + \left\|\mathbf{f}\right\|\mathbb{E}\left[1-h \mid \mathcal{F}\right], \tag{31}$$

where $(d)$ is by $h \leq 1$ and $-\left\langle\mathbf{e},\mathbf{f}\right\rangle \leq \left\|\mathbf{e}\right\|\left\|\mathbf{f}\right\| = \left\|\mathbf{f}\right\|$, and $(e)$ is by Hölder's inequality and (16).

- Inequality 5. Noticing that $\frac{\mathfrak{p}}{\mathfrak{p}-1} \geq 1$ and $1-h \leq 1$, we then have

$$(1-h)^{\frac{\mathfrak{p}}{\mathfrak{p}-1}} \leq 1-h \overset{(18)}{\leq} \frac{\left\|\mathbf{g}\right\|^{\mathfrak{p}}}{\tau^{\mathfrak{p}}} \leq \frac{2^{\mathfrak{p}-1}\left(\left\|\mathbf{g}-\mathbf{f}\right\|^{\mathfrak{p}}+\left\|\mathbf{f}\right\|^{\mathfrak{p}}\right)}{\tau^{\mathfrak{p}}},$$

which implies

$$\mathbb{E}\left[(1-h)^{\frac{\mathfrak{p}}{\mathfrak{p}-1}} \mid \mathcal{F}\right] \leq \mathbb{E}\left[1-h \mid \mathcal{F}\right] \overset{(16)}{\leq} \frac{2^{\mathfrak{p}-1}\left(\sigma_{\mathfrak{l}}^{\mathfrak{p}}+\left\|\mathbf{f}\right\|^{\mathfrak{p}}\right)}{\tau^{\mathfrak{p}}}.$$

Combine (31) and the above inequality to have

$$\left\|\mathbf{d}^{\mathrm{b}}\right\| \leq \left(\frac{2^{\mathfrak{p}-1}\left(\sigma_{\mathfrak{l}}^{\mathfrak{p}}+\left\|\mathbf{f}\right\|^{\mathfrak{p}}\right)}{\tau^{\mathfrak{p}}}\right)^{1-\frac{1}{\mathfrak{p}}}\sigma_{\mathfrak{s}} + \left\|\mathbf{f}\right\|\frac{2^{\mathfrak{p}-1}\left(\sigma_{\mathfrak{l}}^{\mathfrak{p}}+\left\|\mathbf{f}\right\|^{\mathfrak{p}}\right)}{\tau^{\mathfrak{p}}}$$
$$\overset{\mathfrak{p}\leq 2}{\leq} \sqrt{2}\left(\sigma_{\mathfrak{l}}^{\mathfrak{p}-1}+\left\|\mathbf{f}\right\|^{\mathfrak{p}-1}\right)\sigma_{\mathfrak{s}}\tau^{1-\mathfrak{p}} + 2\left(\sigma_{\mathfrak{l}}^{\mathfrak{p}}+\left\|\mathbf{f}\right\|^{\mathfrak{p}}\right)\left\|\mathbf{f}\right\|\tau^{-\mathfrak{p}}.$$

- Inequality 6. Recall that $\chi(\alpha) \in \{0,1\} \in \mathcal{F}$, which implies

$$\left\|\mathbf{d}^{\mathrm{b}}\right\|\chi(\alpha) \overset{(31)}{\leq} \left(\mathbb{E}\left[((1-h)\chi(\alpha))^{\frac{\mathfrak{p}}{\mathfrak{p}-1}} \mid \mathcal{F}\right]\right)^{1-\frac{1}{\mathfrak{p}}}\sigma_{\mathfrak{s}} + \left\|\mathbf{f}\right\|\mathbb{E}\left[(1-h)\chi(\alpha) \mid \mathcal{F}\right]$$
$$\overset{\frac{1}{\mathfrak{p}-1}\geq 1}{\leq} \left(\mathbb{E}\left[((1-h)\chi(\alpha))^{\mathfrak{p}} \mid \mathcal{F}\right]\right)^{1-\frac{1}{\mathfrak{p}}}\sigma_{\mathfrak{s}} + \left\|\mathbf{f}\right\|\mathbb{E}\left[(1-h)\chi(\alpha) \mid \mathcal{F}\right]$$
$$\overset{(19)}{\leq} \left(\mathbb{E}\left[\frac{\left\|\mathbf{g}-\mathbf{f}\right\|^{\mathfrak{p}}}{\tau^{\mathfrak{p}}}\chi(\alpha) \mid \mathcal{F}\right]\right)^{1-\frac{1}{\mathfrak{p}}}\sigma_{\mathfrak{s}} + \left\|\mathbf{f}\right\|\mathbb{E}\left[\frac{\left\|\mathbf{g}-\mathbf{f}\right\|^{\mathfrak{p}}}{\alpha^{\mathfrak{p}-1}\tau^{\mathfrak{p}}}\chi(\alpha) \mid \mathcal{F}\right]$$
$$\overset{(16)}{\leq} \left(\sigma_{\mathfrak{s}}\sigma_{\mathfrak{l}}^{\mathfrak{p}-1}\tau^{1-\mathfrak{p}} + \alpha^{1-\mathfrak{p}}\sigma_{\mathfrak{l}}^{\mathfrak{p}}\left\|\mathbf{f}\right\|\tau^{-\mathfrak{p}}\right)\chi(\alpha)$$
$$\leq \sigma_{\mathfrak{s}}\sigma_{\mathfrak{l}}^{\mathfrak{p}-1}\tau^{1-\mathfrak{p}} + \alpha^{1-\mathfrak{p}}\sigma_{\mathfrak{l}}^{\mathfrak{p}}\left\|\mathbf{f}\right\|\tau^{-\mathfrak{p}}.$$

$\square$

# E    FULL THEOREMS FOR UPPER BOUNDS AND PROOFS

In this section, we provide the full description of each theorem given in the main paper with the proof. Besides, we also present new anytime convergence of Stabilized Clipped SGD. All intermediate results used in the analysis are deferred to be proved in Appendix F.

Before starting, we recall that $D = \|\mathbf{x}_\star - \mathbf{x}_1\|$ denotes the distance between the optimal solution and the initial point.

For high-probability convergence, as proposed in (2), a repeatedly used quantity in the clipping threshold is

$$\tau_\star = \left( \min \left\{ \frac{\sigma_\mathfrak{s} \sigma_\mathfrak{l}^{\mathfrak{p}-1}}{\ln \frac{3}{\delta}}, \frac{\sigma_\mathfrak{s}^2}{\sigma_\mathfrak{l}^{2-\mathfrak{p}} \mathbb{1}\left[\mathfrak{p} < 2\right]} \right\} \right)^{\frac{1}{\mathfrak{p}}}, \tag{32}$$

where $\delta \in (0,1]$ is the failure probability, $\mathfrak{p} \in (1,2]$ and $0 \leq \sigma_\mathfrak{s} \leq \sigma_\mathfrak{l}$ are introduced in Assumption 4. Another useful value mentioned before in (3) is

$$\varphi_\star = \max \left\{ \sqrt{d_{\text{eff}}} \ln \frac{3}{\delta}, d_{\text{eff}} \mathbb{1}\left[\mathfrak{p} < 2\right] \right\}, \tag{33}$$

where $d_{\text{eff}} = \sigma_\mathfrak{l}^2 / \sigma_\mathfrak{s}^2$ is called generalized effective dimension defined in (1) satisfying

$$d_{\text{eff}} \in \{0\} \cup [1, \pi d/2], \tag{34}$$

in which $d_{\text{eff}} = 0$ if and only if $\sigma_\mathfrak{l} = \sigma_\mathfrak{s} = 0$, i.e., the noiseless case. Lastly, it is noteworthy that the following equation always holds

$$\varphi_\star = \frac{\sigma_\mathfrak{l}^\mathfrak{p}}{\tau_\star^\mathfrak{p}}. \tag{35}$$

For in-expectation convergence, we will consider a larger quantity in the clipping threshold as mentioned in Section 6:

$$\widetilde{\tau}_\star = \frac{\sigma_\mathfrak{s}^{\frac{2}{\mathfrak{p}}}}{\sigma_\mathfrak{l}^{\frac{2}{\mathfrak{p}}-1} \mathbb{1}\left[\mathfrak{p} < 2\right]}. \tag{36}$$

We also recall

$$\widetilde{\varphi}_\star = d_{\text{eff}} \mathbb{1}\left[\mathfrak{p} < 2\right]. \tag{37}$$

Note that there is

$$\widetilde{\varphi}_\star = \frac{\sigma_\mathfrak{l}^\mathfrak{p}}{\widetilde{\tau}_\star^\mathfrak{p}}. \tag{38}$$

### E.1    GENERAL CONVEX CASE

We provide different convergence rates for general convex objectives. Recall that $\bar{\mathbf{x}}_{T+1}^{\text{cvx}}$ stands for the average iterate after $T$ steps, i.e.,

$$\bar{\mathbf{x}}_{T+1}^{\text{cvx}} = \frac{1}{T} \sum_{t=1}^{T} \mathbf{x}_{t+1}. \tag{39}$$

Moreover, note that Clipped SGD and Stabilized Clipped SGD are the same when the stepsize is constant, as mentioned in Appendix C. Hence, everything in this subsection is proved based on the analysis for Stabilized Clipped SGD.

#### E.1.1    HIGH-PROBABILITY CONVERGENCE

**Known $T$.** We begin with the situation where the time horizon $T$ is known in advance. Theorem 10 below shows the refined high-probability rate for Clipped SGD.

**Theorem 10** (Full statement of Theorem 1). *Under Assumptions 1, 2 (with $\mu = 0$), 3 and 4, for any $T \in \mathbb{N}$ and $\delta \in (0,1]$, setting $\eta_t = \eta_\star, \tau_t = \max\left\{ \frac{G}{1-\alpha}, \tau_\star T^{\frac{1}{\mathfrak{p}}} \right\}, \forall t \in [T]$ where $\alpha = 1/2$,*

$$\eta_\star = \min \left\{ \frac{D/G}{\varphi + \ln \frac{3}{\delta}}, \frac{D/G}{\sqrt{T}}, \frac{D}{\left( \sigma_\mathfrak{s}^{\frac{2}{\mathfrak{p}}-1} \sigma_\mathfrak{l}^{2-\frac{2}{\mathfrak{p}}} + \sigma_\mathfrak{s}^{\frac{1}{\mathfrak{p}}} \sigma_\mathfrak{l}^{1-\frac{1}{\mathfrak{p}}} \ln^{1-\frac{1}{\mathfrak{p}}} \frac{3}{\delta} \right) T^{\frac{1}{\mathfrak{p}}}} \right\}, \tag{40}$$

and $\varphi \leq \varphi_\star$ is a constant defined in (46) and equals $\varphi_\star$ when $T = \Omega\left(\frac{G^{\mathfrak{p}}}{\sigma_{\mathfrak{l}}^{\mathfrak{p}}} \varphi_\star\right)$, then Clipped SGD (Algorithm 1) guarantees that with probability at least $1 - \delta$, $F(\bar{\mathbf{x}}_{T+1}^{\mathrm{cvx}}) - F_\star$ converges at the rate of

$$
\mathcal{O}\left(\frac{\left(\varphi + \ln\frac{3}{\delta}\right) GD}{T} + \frac{GD}{\sqrt{T}} + \frac{\left(\sigma_{\mathfrak{s}}^{\frac{2}{\mathfrak{p}}-1} \sigma_{\mathfrak{l}}^{2-\frac{2}{\mathfrak{p}}} + \sigma_{\mathfrak{s}}^{\frac{1}{\mathfrak{p}}} \sigma_{\mathfrak{l}}^{1-\frac{1}{\mathfrak{p}}} \ln^{1-\frac{1}{\mathfrak{p}}}\frac{3}{\delta}\right) D}{T^{1-\frac{1}{\mathfrak{p}}}}\right).
$$

*Remark* 11. There are two points we want to emphasize:

First, the choice $\alpha = 1/2$ is not essential and can be changed to any $\alpha \in (0,1)$, only resulting in a different hidden constant in the $\mathcal{O}$ notation. In the proof, we try to keep $\alpha$ until the very last step. Moreover, we would like to mention that a small $\alpha$ may lead to better practical performance as suggested in Remark 2 of Parletta et al. (2025).

Second, these rates are presented while assuming the knowledge of all problem-dependent parameters, as ubiquitously done in the optimization literature. However, not all problem-dependent parameters are necessary if one only wants to ensure convergence. For example, in the above Theorem 10, taking $\eta_t = \min\left\{\frac{\lambda}{G\sqrt{T}}, \frac{\lambda}{\tau_t}\right\}, \tau_t = \max\left\{2G, \tau T^{\frac{1}{\mathfrak{p}}}\right\}, \forall t \in [T]$ where $\lambda, \tau > 0$ (like Theorem 3 in Liu & Zhou (2023)) is sufficient to prove that Clipped SGD converges. Therefore, when proving these theorems, we also try to keep a general version of the stepsize scheduling and the clipping threshold until the very last step.

*Proof.* First, a constant stepsize fulfills the requirement of Lemma 4. In addition, our choices of $\eta_t$ and $\tau_t$ also satisfy Conditions 1 and 2 (with $\alpha = 1/2$) in Lemma 6. Therefore, given $T \in \mathbb{N}$ and $\delta \in (0,1]$, Lemmas 4 and 6 together yield with probability at least $1 - \delta$,

$$
\frac{\|\mathbf{x}_\star - \mathbf{x}_{T+1}\|^2}{2\eta_{T+1}} + \sum_{t=1}^{T} F(\mathbf{x}_{t+1}) - F_\star \leq \frac{D^2}{\eta_{T+1}} + 2A_T^{\mathrm{cvx}}
$$

$$
\Rightarrow F(\bar{\mathbf{x}}_{T+1}^{\mathrm{cvx}}) - F_\star \leq \frac{D^2}{\eta_{T+1}T} + \frac{2A_T^{\mathrm{cvx}}}{T}, \tag{41}
$$

where $A_T^{\mathrm{cvx}}$ is a constant in the order of

$$
\mathcal{O}\left(\max_{t\in[T]} \eta_t \tau_t^2 \ln^2\frac{3}{\delta} + \sum_{t=1}^{T} \sigma_{\mathfrak{l}}^{\mathfrak{p}} \eta_t \tau_t^{2-\mathfrak{p}} + \left(\sum_{t=1}^{T} \frac{\sigma_{\mathfrak{s}} \sigma_{\mathfrak{l}}^{\mathfrak{p}-1} \sqrt{\eta_t}}{\tau_t^{\mathfrak{p}-1}} + \frac{\sigma_{\mathfrak{l}}^{\mathfrak{p}} G \sqrt{\eta_t}}{\alpha^{\mathfrak{p}-1} \tau_t^{\mathfrak{p}}}\right)^2 + \sum_{t=1}^{T} G^2 \eta_t\right). \tag{42}
$$

Our left task is to bound $A_T^{\mathrm{cvx}}$. When $\eta_t = \eta, \tau_t = \tau, \forall t \in [T]$ where $\eta > 0$ and $\tau \geq \frac{G}{1-\alpha}$ (as required by Condition 2 in Lemma 6), we can simplify (42) into

$$
A_T^{\mathrm{cvx}} = \mathcal{O}\left(\eta\left(\tau^2 \ln^2\frac{3}{\delta} + \sigma_{\mathfrak{l}}^{\mathfrak{p}} \tau^{2-\mathfrak{p}} T + \frac{\sigma_{\mathfrak{s}}^2 \sigma_{\mathfrak{l}}^{2\mathfrak{p}-2}}{\tau^{2\mathfrak{p}-2}} T^2 + \frac{\sigma_{\mathfrak{l}}^{2\mathfrak{p}} G^2}{\alpha^{2\mathfrak{p}-2} \tau^{2\mathfrak{p}}} T^2 + G^2 T\right)\right). \tag{43}
$$

One more step, under changing $\tau$ to $\max\left\{\frac{G}{1-\alpha}, \tau\right\}$ (the second $\tau$ is only required to be nonnegative), we can further write (43) into

$$
\begin{aligned}
A_T^{\mathrm{cvx}} = & \mathcal{O}\left(\eta\left(\frac{\sigma_{\mathfrak{l}}^{2\mathfrak{p}} G^2 T^2}{\min\{\alpha^{2\mathfrak{p}-2}, (1-\alpha)^2\} \left(\max\left\{\frac{G}{1-\alpha}, \tau\right\}\right)^{2\mathfrak{p}}} + \frac{G^2 \ln^2\frac{3}{\delta}}{(1-\alpha)^2} + G^2 T\right)\right. \\
& \left. + \eta\left(\tau^2 \ln^2\frac{3}{\delta} + \sigma_{\mathfrak{l}}^{\mathfrak{p}} \tau^{2-\mathfrak{p}} T + \frac{\sigma_{\mathfrak{s}}^2 \sigma_{\mathfrak{l}}^{2\mathfrak{p}-2}}{\tau^{2\mathfrak{p}-2}} T^2\right)\right),
\end{aligned}
$$

where we use

$$\sigma_{\mathfrak{l}}^{\mathfrak{p}} \left( \max \left\{ \frac{G}{1-\alpha}, \tau \right\} \right)^{2-\mathfrak{p}} T \leq \frac{\sigma_{\mathfrak{l}}^{\mathfrak{p}} G^2 T}{(1-\alpha)^2 \left( \max \left\{ \frac{G}{1-\alpha}, \tau \right\} \right)^{\mathfrak{p}}} + \sigma_{\mathfrak{l}}^{\mathfrak{p}} \tau^{2-\mathfrak{p}} T$$

$$\leq \frac{\sigma_{\mathfrak{l}}^{2\mathfrak{p}} G^2 T^2}{(1-\alpha)^2 \left( \max \left\{ \frac{G}{1-\alpha}, \tau \right\} \right)^{2\mathfrak{p}}} + \frac{G^2}{4(1-\alpha)^2} + \sigma_{\mathfrak{l}}^{\mathfrak{p}} \tau^{2-\mathfrak{p}} T$$

$$\leq \frac{\sigma_{\mathfrak{l}}^{2\mathfrak{p}} G^2 T^2}{\min \left\{ \alpha^{2\mathfrak{p}-2}, (1-\alpha)^2 \right\} \left( \max \left\{ \frac{G}{1-\alpha}, \tau \right\} \right)^{2\mathfrak{p}}} + \frac{G^2 \ln^2 \frac{3}{\delta}}{4(1-\alpha)^2} + \sigma_{\mathfrak{l}}^{\mathfrak{p}} \tau^{2-\mathfrak{p}} T.$$

Furthermore, we replace the current $\tau$ with $\tau T^{\frac{1}{\mathfrak{p}}}$ to obtain

$$A_T^{\mathrm{cvx}} = \mathcal{O} \left( \eta \left( \inf_{\beta \in [0,1/2]} \frac{(1-\alpha)^{2\beta\mathfrak{p}} \sigma_{\mathfrak{l}}^{2\mathfrak{p}} G^{2-2\beta\mathfrak{p}} T^{2\beta}}{\min \left\{ \alpha^{2\mathfrak{p}-2}, (1-\alpha)^2 \right\} \tau^{2(1-\beta)\mathfrak{p}}} + \frac{G^2 \ln^2 \frac{3}{\delta}}{(1-\alpha)^2} + G^2 T \right) \right.$$

$$\left. + \eta \left( \tau^2 \ln^2 \frac{3}{\delta} + \sigma_{\mathfrak{l}}^{\mathfrak{p}} \tau^{2-\mathfrak{p}} + \frac{\sigma_{\mathfrak{s}}^2 \sigma_{\mathfrak{l}}^{2\mathfrak{p}-2}}{\tau^{2\mathfrak{p}-2}} \right) T^{\frac{2}{\mathfrak{p}}} \right), \tag{44}$$

where the first term appears due to, for any $\beta \in [0, 1/2]$,

$$\frac{\sigma_{\mathfrak{l}}^{2\mathfrak{p}} G^2 T^2}{\left( \max \left\{ \frac{G}{1-\alpha}, \tau T^{\frac{1}{\mathfrak{p}}} \right\} \right)^{2\mathfrak{p}}} \leq \frac{\sigma_{\mathfrak{l}}^{2\mathfrak{p}} G^2 T^2}{\left( \frac{G}{1-\alpha} \right)^{2\beta\mathfrak{p}} \left( \tau T^{\frac{1}{\mathfrak{p}}} \right)^{2(1-\beta)\mathfrak{p}}} = \frac{(1-\alpha)^{2\beta\mathfrak{p}} \sigma_{\mathfrak{l}}^{2\mathfrak{p}} G^{2-2\beta\mathfrak{p}} T^{2\beta}}{\tau^{2(1-\beta)\mathfrak{p}}}.$$

Now, we plug $\tau = \tau_\star$ (see (32)) into (44) to have, under $\tau_t = \max \left\{ \frac{G}{1-\alpha}, \tau_\star T^{\frac{1}{\mathfrak{p}}} \right\}, \forall t \in [T]$,

$$A_T^{\mathrm{cvx}} = \mathcal{O} \left( \eta \left( \frac{G^2 \varphi^2}{\min \left\{ \alpha^{2\mathfrak{p}-2}, (1-\alpha)^2 \right\}} + \frac{G^2 \ln^2 \frac{3}{\delta}}{(1-\alpha)^2} \right) + \eta G^2 T \right.$$

$$\left. + \eta \left( \sigma_{\mathfrak{s}}^{\frac{4}{\mathfrak{p}}-2} \sigma_{\mathfrak{l}}^{4-\frac{4}{\mathfrak{p}}} + \sigma_{\mathfrak{s}}^{\frac{2}{\mathfrak{p}}} \sigma_{\mathfrak{l}}^{2-\frac{2}{\mathfrak{p}}} \ln^{2-\frac{2}{\mathfrak{p}}} \frac{3}{\delta} \right) T^{\frac{2}{\mathfrak{p}}} \right), \tag{45}$$

where the first term is obtained by noticing

$$\frac{(1-\alpha)^{2\beta\mathfrak{p}} \sigma_{\mathfrak{l}}^{2\mathfrak{p}} G^{2-2\beta\mathfrak{p}} T^{2\beta}}{\tau_\star^{2(1-\beta)\mathfrak{p}}} = G^2 \cdot \left( (1-\alpha)^{\beta\mathfrak{p}} \frac{\sigma_{\mathfrak{l}}^{(1-\beta)\mathfrak{p}}}{\tau_\star^{(1-\beta)\mathfrak{p}}} \left( \frac{\sigma_{\mathfrak{l}}}{G} \right)^{\beta\mathfrak{p}} T^\beta \right)^2$$

$$\overset{(35)}{=} G^2 \cdot \left( (1-\alpha)^{\beta\mathfrak{p}} \varphi_\star^{1-\beta} \left( \frac{\sigma_{\mathfrak{l}}}{G} \right)^{\beta\mathfrak{p}} T^\beta \right)^2$$

$$\Rightarrow \inf_{\beta \in [0,1/2]} \frac{(1-\alpha)^{2\beta\mathfrak{p}} \sigma_{\mathfrak{l}}^{2\mathfrak{p}} G^{2-2\beta\mathfrak{p}} T^{2\beta}}{\tau_\star^{2(1-\beta)\mathfrak{p}}} \leq G^2 \varphi^2,$$

in which

$$\varphi \triangleq \inf_{\beta \in [0,1/2]} (1-\alpha)^{\beta\mathfrak{p}} \varphi_\star^{1-\beta} \left( \frac{\sigma_{\mathfrak{l}}}{G} \right)^{\beta\mathfrak{p}} T^\beta = \min \left\{ \varphi_\star, \sqrt{(1-\alpha)^{\mathfrak{p}} \varphi_\star \left( \frac{\sigma_{\mathfrak{l}}}{G} \right)^{\mathfrak{p}} T} \right\} \leq \varphi_\star. \tag{46}$$

Note that we have $\varphi = \varphi_\star$ when $T \geq \frac{G^{\mathfrak{p}} \varphi_\star}{(1-\alpha)^{\mathfrak{p}} \sigma_{\mathfrak{l}}^{\mathfrak{p}}} = \Omega \left( \frac{G^{\mathfrak{p}}}{\sigma_{\mathfrak{l}}^{\mathfrak{p}}} \varphi_\star \right)$.

By (41), (45) and $\alpha = 1/2$, we can find

$$F(\bar{\mathbf{x}}_{T+1}^{\mathrm{cvx}}) - F_\star \leq \mathcal{O} \left( \frac{D^2}{\eta T} + \frac{\eta \left( \varphi^2 + \ln^2 \frac{3}{\delta} \right) G^2}{T} + \eta G^2 \right.$$

$$\left. + \eta \left( \sigma_{\mathfrak{s}}^{\frac{4}{\mathfrak{p}}-2} \sigma_{\mathfrak{l}}^{4-\frac{4}{\mathfrak{p}}} + \sigma_{\mathfrak{s}}^{\frac{2}{\mathfrak{p}}} \sigma_{\mathfrak{l}}^{2-\frac{2}{\mathfrak{p}}} \ln^{2-\frac{2}{\mathfrak{p}}} \frac{3}{\delta} \right) T^{\frac{2}{\mathfrak{p}}-1} \right).$$

Plug in $\eta = \eta_\star$ (see (40)) to conclude that $F(\bar{\mathbf{x}}_{T+1}^{\mathrm{cvx}}) - F_\star$ converges at the rate of

$$\mathcal{O}\left( \frac{\left(\varphi + \ln \frac{3}{\delta}\right) GD}{T} + \frac{GD}{\sqrt{T}} + \frac{\left( \sigma_{\mathfrak{s}}^{\frac{2}{\mathfrak{p}}-1} \sigma_{\mathfrak{l}}^{2-\frac{2}{\mathfrak{p}}} + \sigma_{\mathfrak{s}}^{\frac{1}{\mathfrak{p}}} \sigma_{\mathfrak{l}}^{1-\frac{1}{\mathfrak{p}}} \ln^{1-\frac{1}{\mathfrak{p}}} \frac{3}{\delta} \right) D}{T^{1-\frac{1}{\mathfrak{p}}}} \right).$$

$\square$

**Recover the existing rate in** Liu & Zhou (2023). Remarkably, our above analysis is essentially tighter than Liu & Zhou (2023). To see this claim, we bound $A_T^{\mathrm{cvx}}$ in the following way (take the same $\alpha = 1/2$ as in Liu & Zhou (2023) for a fair comparison):

$$A_T^{\mathrm{cvx}} \overset{(43)}{=} \mathcal{O}\left( \eta \left( \tau^2 \ln^2 \frac{3}{\delta} + \sigma_{\mathfrak{l}}^{\mathfrak{p}} \tau^{2-\mathfrak{p}} T + \frac{\sigma_{\mathfrak{s}}^2 \sigma_{\mathfrak{l}}^{2\mathfrak{p}-2}}{\tau^{2\mathfrak{p}-2}} T^2 + \frac{\sigma_{\mathfrak{l}}^{2\mathfrak{p}} G^2}{\tau^{2\mathfrak{p}}} T^2 + G^2 T \right) \right)$$

$$\overset{(a)}{\leq} \mathcal{O}\left( \eta \left( \tau^2 \ln^2 \frac{3}{\delta} + \sigma_{\mathfrak{l}}^{\mathfrak{p}} \tau^{2-\mathfrak{p}} T + \frac{\sigma_{\mathfrak{s}}^2 \sigma_{\mathfrak{l}}^{2\mathfrak{p}-2}}{\tau^{2\mathfrak{p}-2}} T^2 + \frac{\sigma_{\mathfrak{l}}^{2\mathfrak{p}}}{\tau^{2\mathfrak{p}-2}} T^2 + G^2 T \right) \right)$$

$$\overset{(b)}{=} \mathcal{O}\left( \eta \left( \tau^2 \ln^2 \frac{3}{\delta} + \sigma_{\mathfrak{l}}^{\mathfrak{p}} \tau^{2-\mathfrak{p}} T + \frac{\sigma_{\mathfrak{l}}^{2\mathfrak{p}}}{\tau^{2\mathfrak{p}-2}} T^2 + G^2 T \right) \right),$$

where $(a)$ is by $\tau \geq \frac{G}{1-\alpha} = 2G$ and $(b)$ holds due to $\sigma_{\mathfrak{s}} \leq \sigma_{\mathfrak{l}}$. Under the choice of $\eta = \min\left\{ \frac{\lambda}{G\sqrt{T}}, \frac{\lambda}{\tau} \right\}$ used in Theorem 3 of Liu & Zhou (2023), we have

$$A_T^{\mathrm{cvx}} \leq \mathcal{O}\left( \frac{\lambda^2}{\eta} \left( \ln^2 \frac{3}{\delta} + \frac{\sigma_{\mathfrak{l}}^{\mathfrak{p}}}{\tau^{\mathfrak{p}}} T + \frac{\sigma_{\mathfrak{l}}^{2\mathfrak{p}}}{\tau^{2\mathfrak{p}}} T^2 + 1 \right) \right) \leq \mathcal{O}\left( \frac{\lambda^2}{\eta} \left( \ln^2 \frac{3}{\delta} + \frac{\sigma_{\mathfrak{l}}^{2\mathfrak{p}}}{\tau^{2\mathfrak{p}}} T^2 \right) \right),$$

where the second step is due to $\frac{2\sigma_{\mathfrak{l}}^{\mathfrak{p}}}{\tau^{\mathfrak{p}}} T \leq \frac{\sigma_{\mathfrak{l}}^{2\mathfrak{p}}}{\tau^{2\mathfrak{p}}} T^2 + 1$ (by AM-GM inequality) and $1 \leq \ln^2 \frac{3}{\delta}$. Lastly, we replace $\tau$ with $\max\left\{ 2G, \tau T^{\frac{1}{\mathfrak{p}}} \right\}$ given in Theorem 3 of Liu & Zhou (2023) to obtain

$$A_T^{\mathrm{cvx}} = \mathcal{O}\left( \frac{\lambda^2}{\eta} \left( \ln^2 \frac{3}{\delta} + \frac{\sigma_{\mathfrak{l}}^{2\mathfrak{p}}}{\tau^{2\mathfrak{p}}} \right) \right).$$

Combine with (41) to finally have

$$F(\bar{\mathbf{x}}_{T+1}^{\mathrm{cvx}}) - F_\star \leq \mathcal{O}\left( \frac{D^2 + \lambda^2 \left( \ln^2 \frac{3}{\delta} + \frac{\sigma_{\mathfrak{l}}^{2\mathfrak{p}}}{\tau^{2\mathfrak{p}}} \right)}{\eta T} \right),$$

which is the same rate as given in Liu & Zhou (2023) (see their equation (7)), implying that our analysis is indeed more refined than Liu & Zhou (2023).

**Unknown $T$.** We move to the case of unknown $T$. Theorem 11 in the following gives the anytime high-probability rate for Stabilized Clipped SGD.

**Theorem 11.** *Under Assumptions 1, 2 (with $\mu = 0$), 3 and 4, for any $T \in \mathbb{N}$ and $\delta \in (0, 1]$, setting $\eta_t = \min\left\{ \gamma_\star, \frac{\eta_\star}{\sqrt{t}}, \frac{\lambda_\star}{\tau_\star t^{\frac{1}{\mathfrak{p}}}} \right\}, \tau_t = \max\left\{ \frac{G}{1-\alpha}, \tau_\star t^{\frac{1}{\mathfrak{p}}} \right\}, \forall t \in [T]$ where $\alpha = 1/2$,*

$$\gamma_\star = \frac{D/G}{\varphi_\star \psi_\star + \ln \frac{3}{\delta}}, \quad \eta_\star = D/G, \quad \lambda_\star = \frac{D}{\sqrt{\ln^2 \frac{3}{\delta} + \frac{\sigma_{\mathfrak{l}}^{\mathfrak{p}}}{\tau_\star^{\mathfrak{p}}} + \frac{\sigma_{\mathfrak{s}}^2 \sigma_{\mathfrak{l}}^{2\mathfrak{p}-2}}{\tau_\star^{2\mathfrak{p}}}}}, \quad (47)$$

*and $\psi_\star \triangleq 1 + \ln \varphi_\star$, then Stabilized Clipped SGD (Algorithm 2) guarantees that with probability at least $1 - \delta$, $F(\bar{\mathbf{x}}_{T+1}^{\mathrm{cvx}}) - F_\star$ converges at the rate of*

$$\mathcal{O}\left( \frac{\left(\varphi_\star \psi_\star + \ln \frac{3}{\delta}\right) GD}{T} + \frac{GD}{\sqrt{T}} + \frac{\left( \sigma_{\mathfrak{s}}^{\frac{2}{\mathfrak{p}}-1} \sigma_{\mathfrak{l}}^{2-\frac{2}{\mathfrak{p}}} + \sigma_{\mathfrak{s}}^{\frac{1}{\mathfrak{p}}} \sigma_{\mathfrak{l}}^{1-\frac{1}{\mathfrak{p}}} \ln^{1-\frac{1}{\mathfrak{p}}} \frac{3}{\delta} \right) D}{T^{1-\frac{1}{\mathfrak{p}}}} \right).$$

*Proof.* By the same argument for (41) in the proof of Theorem 10, we have with probability at least $1 - \delta$,

$$F(\bar{\mathbf{x}}_{T+1}^{\text{cvx}}) - F_\star \leq \frac{D^2}{\eta_{T+1}T} + \frac{2A_T^{\text{cvx}}}{T}, \tag{48}$$

where $A_T^{\text{cvx}}$ is a constant in the order of

$$\mathcal{O}\left(\underbrace{\max_{t \in [T]} \eta_t \tau_t^2 \ln^2 \frac{3}{\delta}}_{\text{I}} + \underbrace{\sum_{t=1}^{T} \sigma_{\mathfrak{l}}^{\mathfrak{p}} \eta_t \tau_t^{2-\mathfrak{p}}}_{\text{II}} + \left(\underbrace{\sum_{t=1}^{T} \frac{\sigma_{\mathfrak{s}} \sigma_{\mathfrak{l}}^{\mathfrak{p}-1} \sqrt{\eta_t}}{\tau_t^{\mathfrak{p}-1}}}_{\text{III}} + \underbrace{\sum_{t=1}^{T} \frac{\sigma_{\mathfrak{l}}^{\mathfrak{p}} G \sqrt{\eta_t}}{\alpha^{\mathfrak{p}-1} \tau_t^{\mathfrak{p}}}}_{\text{IV}}\right)^2 + \underbrace{\sum_{t=1}^{T} G^2 \eta_t}_{\text{V}}\right). \tag{49}$$

When $\eta_t = \min\left\{\gamma, \frac{\eta}{\sqrt{t}}, \frac{\lambda}{\tau t^{\frac{1}{\mathfrak{p}}}}\right\}, \tau_t = \max\left\{\frac{G}{1-\alpha}, \tau t^{\frac{1}{\mathfrak{p}}}\right\}, \forall t \in [T]$ for nonnegative $\gamma, \eta, \lambda$ and $\tau$, we can bound the above five terms as follows.

- Term I. We have

$$\max_{t \in [T]} \eta_t \tau_t^2 \ln^2 \frac{3}{\delta} \leq \max_{t \in [T]} \left(\frac{\eta_t G^2}{(1-\alpha)^2} + \eta_t \left(\tau t^{\frac{1}{\mathfrak{p}}}\right)^2\right) \ln^2 \frac{3}{\delta}$$

$$\leq \max_{t \in [T]} \left(\frac{\gamma G^2}{(1-\alpha)^2} + \lambda \tau t^{\frac{1}{\mathfrak{p}}}\right) \ln^2 \frac{3}{\delta} = \mathcal{O}\left(\frac{\gamma G^2 \ln^2 \frac{3}{\delta}}{(1-\alpha)^2} + \lambda \tau \ln^2\left(\frac{3}{\delta}\right) T^{\frac{1}{\mathfrak{p}}}\right). \tag{50}$$

- Term II. For any $t \in [T]$, we have

$$\sigma_{\mathfrak{l}}^{\mathfrak{p}} \eta_t \tau_t^{2-\mathfrak{p}} \leq \frac{\sigma_{\mathfrak{l}}^{\mathfrak{p}} G^2 \eta_t}{(1-\alpha)^2 \tau_t^{\mathfrak{p}}} + \sigma_{\mathfrak{l}}^{\mathfrak{p}} \left(\tau t^{\frac{1}{\mathfrak{p}}}\right)^{2-\mathfrak{p}} \eta_t \leq \frac{\sigma_{\mathfrak{l}}^{\mathfrak{p}} G^2 \sqrt{\gamma \eta_t}}{(1-\alpha)^2 \tau_t^{\mathfrak{p}}} + \frac{\sigma_{\mathfrak{l}}^{\mathfrak{p}} \lambda}{\left(\tau t^{\frac{1}{\mathfrak{p}}}\right)^{\mathfrak{p}-1}},$$

which implies that

$$\sum_{t=1}^{T} \sigma_{\mathfrak{l}}^{\mathfrak{p}} \eta_t \tau_t^{2-\mathfrak{p}} \leq \frac{\sqrt{\gamma} G}{1-\alpha} \left(\sum_{t=1}^{T} \frac{\sigma_{\mathfrak{l}}^{\mathfrak{p}} G \sqrt{\eta_t}}{(1-\alpha) \tau_t^{\mathfrak{p}}}\right) + \sum_{t=1}^{T} \frac{\sigma_{\mathfrak{l}}^{\mathfrak{p}} \lambda}{\left(\tau t^{\frac{1}{\mathfrak{p}}}\right)^{\mathfrak{p}-1}}$$

$$\leq \frac{\gamma G^2}{4(1-\alpha)^2} + \left(\sum_{t=1}^{T} \frac{\sigma_{\mathfrak{l}}^{\mathfrak{p}} G \sqrt{\eta_t}}{(1-\alpha) \tau_t^{\mathfrak{p}}}\right)^2 + \sum_{t=1}^{T} \frac{\sigma_{\mathfrak{l}}^{\mathfrak{p}} \lambda}{\left(\tau t^{\frac{1}{\mathfrak{p}}}\right)^{\mathfrak{p}-1}}$$

$$\leq \mathcal{O}\left(\frac{\gamma G^2 \ln^2 \frac{3}{\delta}}{(1-\alpha)^2} + \left(\frac{\alpha^{\mathfrak{p}-1}}{1-\alpha} \cdot \text{Term IV}\right)^2 + \frac{\lambda \sigma_{\mathfrak{l}}^{\mathfrak{p}}}{\tau^{\mathfrak{p}-1}} T^{\frac{1}{\mathfrak{p}}}\right). \tag{51}$$

- Term III. For any $t \in [T]$, we have

$$\frac{\sqrt{\eta_t}}{\tau_t^{\mathfrak{p}-1}} \overset{\mathfrak{p} \geq 1}{\lesssim} \frac{\sqrt{\lambda/(\tau t^{\frac{1}{\mathfrak{p}}})}}{(\tau t^{\frac{1}{\mathfrak{p}}})^{\mathfrak{p}-1}} = \frac{\sqrt{\lambda}}{(\tau t^{\frac{1}{\mathfrak{p}}})^{\mathfrak{p}-\frac{1}{2}}},$$

which implies

$$\sum_{t=1}^{T} \frac{\sigma_{\mathfrak{s}} \sigma_{\mathfrak{l}}^{\mathfrak{p}-1} \sqrt{\eta_t}}{\tau_t^{\mathfrak{p}-1}} \leq \mathcal{O}\left(\frac{\sqrt{\lambda} \sigma_{\mathfrak{s}} \sigma_{\mathfrak{l}}^{\mathfrak{p}-1}}{\tau^{\mathfrak{p}-\frac{1}{2}}} T^{\frac{1}{2\mathfrak{p}}}\right). \tag{52}$$

- Term IV. For any $\beta \in [0, 1]$, we have

$$\sum_{t=1}^{T} \frac{\sigma_{\mathfrak{l}}^{\mathfrak{p}} G \sqrt{\eta_t}}{\alpha^{\mathfrak{p}-1} \tau_t^{\mathfrak{p}}} \leq \sum_{t=1}^{T} \frac{\sigma_{\mathfrak{l}}^{\mathfrak{p}} G \gamma^{\frac{1-\beta}{2}} \eta^{\frac{\beta}{2}}}{\alpha^{\mathfrak{p}-1} (\tau t^{\frac{1}{\mathfrak{p}}})^{\mathfrak{p}} t^{\frac{\beta}{4}}} = \mathcal{O}\left(\frac{\sqrt{\gamma} \sigma_{\mathfrak{l}}^{\mathfrak{p}} G}{\alpha^{\mathfrak{p}-1} \tau^{\mathfrak{p}}} \left(\frac{\eta}{\gamma}\right)^{\frac{\beta}{2}} \psi(\beta, T)\right), \tag{53}$$

where

$$\psi(\beta, T) \triangleq \begin{cases} 1 + \ln T & \beta = 0 \\ 1 + \frac{4}{\beta} & \beta \in (0, 1] \end{cases}. \tag{54}$$

- **Term V.** We have

$$\sum_{t=1}^{T} G^2 \eta_t \leq \sum_{t=1}^{T} \frac{\eta G^2}{\sqrt{t}} = \mathcal{O}\left(\eta G^2 \sqrt{T}\right). \tag{55}$$

We plug (50), (51), (52), (53) and (55) back into (49) to know

$$A_T^{\mathrm{cvx}} \leq \mathcal{O}\left(\gamma\left(\frac{\sigma_{\mathfrak{l}}^{2\mathfrak{p}} G^2 (\eta/\gamma)^\beta \psi^2(\beta, T)}{\min\left\{\alpha^{2\mathfrak{p}-2}, (1-\alpha)^2\right\} \tau^{2\mathfrak{p}}} + \frac{G^2 \ln^2 \frac{3}{\delta}}{(1-\alpha)^2}\right) + \eta G^2 \sqrt{T}\right.$$
$$\left. + \lambda\left(\tau \ln^2 \frac{3}{\delta} + \frac{\sigma_{\mathfrak{l}}^{\mathfrak{p}}}{\tau^{\mathfrak{p}-1}} + \frac{\sigma_{\mathfrak{s}}^2 \sigma_{\mathfrak{l}}^{2\mathfrak{p}-2}}{\tau^{2\mathfrak{p}-1}}\right) T^{\frac{1}{\mathfrak{p}}}\right), \forall \beta \in [0, 1].$$

Combine the above result with $\eta_t = \min\left\{\gamma, \frac{\eta}{\sqrt{t}}, \frac{\lambda}{\tau t^{\frac{1}{\mathfrak{p}}}}\right\}$ and (48) to obtain

$$F(\bar{\mathbf{x}}_{T+1}^{\mathrm{cvx}}) - F_\star \leq \mathcal{O}\left(\frac{\frac{D^2}{\gamma} + \gamma\left(\frac{\sigma_{\mathfrak{l}}^{2\mathfrak{p}} G^2 (\eta/\gamma)^\beta \psi^2(\beta, T)}{\min\{\alpha^{2\mathfrak{p}-2},(1-\alpha)^2\}\tau^{2\mathfrak{p}}} + \frac{G^2 \ln^2 \frac{3}{\delta}}{(1-\alpha)^2}\right)}{T} + \frac{\frac{D^2}{\eta} + \eta G^2}{\sqrt{T}}\right.$$
$$\left. + \frac{\frac{D^2 \tau}{\lambda} + \lambda\left(\tau \ln^2\left(\frac{3}{\delta}\right) + \frac{\sigma_{\mathfrak{l}}^{\mathfrak{p}}}{\tau^{\mathfrak{p}-1}} + \frac{\sigma_{\mathfrak{s}}^2 \sigma_{\mathfrak{l}}^{2\mathfrak{p}-2}}{\tau^{2\mathfrak{p}-1}}\right)}{T^{1-\frac{1}{\mathfrak{p}}}}\right), \forall \beta \in [0, 1]. \tag{56}$$

Finally, we conclude after plugging in $\tau = \tau_\star, \gamma = \gamma_\star, \eta = \eta_\star, \lambda = \lambda_\star$ (see (32) and (47)), $\alpha = 1/2$, and the following fact:

$$\inf_{\beta \in [0,1]} \gamma_\star \left(\frac{\eta_\star}{\gamma_\star}\right)^\beta \psi^2(\beta, T) \overset{(47), \beta \leq 1}{\leq} \frac{D/G}{\varphi_\star \psi_\star} \inf_{\beta \in [0,1]} (\varphi_\star \psi_\star)^\beta \psi^2(\beta, T)$$
$$\leq \frac{D/G}{\varphi_\star \psi_\star} (\varphi_\star \psi_\star)^{\beta_\star} \psi^2(\beta_\star, T) \quad \text{where} \quad \beta_\star = \frac{2}{\max\{\ln(\varphi_\star \psi_\star), 2\}}$$
$$\overset{(54)}{\leq} \frac{D/G}{\varphi_\star \psi_\star} \cdot e^2 \cdot (1 + 2\max\{\ln(\varphi_\star \psi_\star), 2\})^2$$
$$= \mathcal{O}\left(\frac{D/G}{\varphi_\star \psi_\star} \cdot \left(1 + \ln^2 \varphi_\star + \ln^2 \psi_\star\right)\right) = \mathcal{O}\left(\frac{D/G}{\varphi_\star} \cdot \psi_\star\right),$$

where the last step is by $\ln \psi_\star \leq 2\sqrt{\psi_\star}, 1 + \ln^2 \varphi_\star \leq \psi_\star^2$ (since $\psi_\star = 1 + \ln \varphi_\star$ and $\varphi_\star \geq 1$), and $\psi_\star \geq 1$. $\qquad\square$

We first compare Theorem 11 with our Theorem 10. As one can see, the only difference is the term $\varphi$ versus the term $\varphi_\star \psi_\star$, the former of which satisfies $\varphi \leq \varphi_\star$. This change should be expected as the precise value of $\varphi$ depends on $T$ (see (46)). Moreover, recall that $\varphi = \varphi_\star$ once $T$ exceeds $\Omega\left(\frac{G^{\mathfrak{p}}}{\sigma_{\mathfrak{l}}^{\mathfrak{p}}} \varphi_\star\right)$. Hence, roughly speaking, the only loss in Theorem 11 is an extra multiplicative term $\psi_\star$, which never grows with $T$ and is in the order of

$$1 + \ln \varphi_\star \overset{(33)}{=} 1 + \ln\left(\max\left\{\sqrt{d_{\mathrm{eff}}} \ln \frac{3}{\delta}, d_{\mathrm{eff}} \mathbb{1}\left[\mathfrak{p} < 2\right]\right\}\right).$$

This positive result, i.e., no extra $\mathrm{poly}(\ln T)$ term, is due to the stabilization technique, as discussed in Appendix C.

Without considering the extra stabilized step, following a similar analysis given in Appendix F later, one can show that for any general stepsize $\eta_t$ and any clipping threshold $\tau_t \geq \frac{G}{1-\alpha}$, Clipped SGD guarantees with probability at least $1 - \delta$ (assuming that $\eta_t$ is nonincreasing for simplicity),

$$F(\bar{\mathbf{x}}_{T+1}^{\mathrm{cvx}}) - F_\star \leq \left(\frac{D^2 + \tilde{A}_T^{\mathrm{cvx}}}{\eta_T T}\right), \tag{57}$$

where $\tilde{A}_T^{\mathrm{cvx}}$ is in the order of

$$
\mathcal{O}\left(\max_{t\in[T]}\eta_t^2\tau_t^2\ln^2\frac{3}{\delta} + \sum_{t=1}^{T}\sigma_{\mathfrak{l}}^{\mathfrak{p}}\eta_t^2\tau_t^{2-\mathfrak{p}} + \left(\sum_{t=1}^{T}\frac{\sigma_{\mathfrak{s}}\sigma_{\mathfrak{l}}^{\mathfrak{p}-1}\eta_t}{\tau_t^{\mathfrak{p}-1}} + \sum_{t=1}^{T}\frac{\sigma_{\mathfrak{l}}^{\mathfrak{p}}G\eta_t}{\alpha^{\mathfrak{p}-1}\tau_t^{\mathfrak{p}}}\right)^2 + \sum_{t=1}^{T}G^2\eta_t^2\right). \quad (58)
$$

As a sanity check, when $\eta_t = \eta, \tau_t = \tau, \forall t \in [T]$, $\tilde{A}_T^{\mathrm{cvx}}/\eta$ coincides with $A_T^{\mathrm{cvx}}$ given in (43). If $T$ is unknown, even ignoring all other terms and only focusing on $\sum_{t=1}^{T}G^2\eta_t^2$ in (58), the final rate of Clipped SGD by (57) will contain a term $\sum_{t=1}^{T}G^2\eta_t^2/(\eta_T T)$, which is however well-known to give an extra $\mathrm{poly}(\ln T)$ factor for a time-varying stepsize $\eta_t$.

Now let us compare Theorem 11 to Theorem 1 in Liu & Zhou (2023). The latter gives the current best anytime rate for Clipped SGD as follows (actually, this can be obtained by (57) and (58) above):

$$
F(\bar{\mathbf{x}}_{T+1}^{\mathrm{cvx}}) - F_\star \leq \mathcal{O}\left(\left(\ln\frac{1}{\delta} + \ln^2 T\right)\left(\frac{GD}{\sqrt{T}} + \frac{\sigma_{\mathfrak{l}}D}{T^{1-\frac{1}{\mathfrak{p}}}}\right)\right).
$$

Similar to our comparison when $T$ is known in Section 4, one can see that our Theorem 11 is better (at least in the case of large $T$).

### E.1.2 IN-EXPECTATION CONVERGENCE

**Known** $T$. Now we consider the in-expectation convergence. Theorem 12 gives the first rate $\mathcal{O}(\sigma_{\mathfrak{l}}d_{\mathrm{eff}}^{\frac{1}{2}-\frac{1}{\mathfrak{p}}}DT^{\frac{1}{\mathfrak{p}}-1})$ faster than the existing lower bound $\Omega(\sigma_{\mathfrak{l}}DT^{\frac{1}{\mathfrak{p}}-1})$ (Nemirovski & Yudin, 1983; Vural et al., 2022).

**Theorem 12** (Full statement of Theorem 3). *Under Assumptions 1, 2 (with $\mu = 0$), 3 and 4, for any $T \in \mathbb{N}$, setting $\eta_t = \eta_\star, \tau_t = \max\left\{\frac{G}{1-\alpha}, \tilde{\tau}_\star T^{\frac{1}{\mathfrak{p}}}\right\}, \forall t \in [T]$ where $\alpha = 1/2$,*

$$
\eta_\star = \min\left\{\frac{D/G}{\tilde{\varphi}}, \frac{D/G}{\sqrt{T}}, \frac{D}{\sigma_{\mathfrak{s}}^{\frac{2}{\mathfrak{p}}-1}\sigma_{\mathfrak{l}}^{2-\frac{2}{\mathfrak{p}}}T^{\frac{1}{\mathfrak{p}}}}\right\}, \quad (59)
$$

*and $\tilde{\varphi} \leq \tilde{\varphi}_\star$ is a constant defined in (60) and equals $\tilde{\varphi}_\star$ when $T = \Omega\left(\frac{G^{\mathfrak{p}}}{\sigma_{\mathfrak{l}}^{\mathfrak{p}}}\tilde{\varphi}_\star\right)$, then Clipped SGD (Algorithm 1) guarantees that $\mathbb{E}\left[F(\bar{\mathbf{x}}_{T+1}^{\mathrm{cvx}}) - F_\star\right]$ converges at the rate of*

$$
\mathcal{O}\left(\frac{\tilde{\varphi}GD}{T} + \frac{GD}{\sqrt{T}} + \frac{\sigma_{\mathfrak{s}}^{\frac{2}{\mathfrak{p}}-1}\sigma_{\mathfrak{l}}^{2-\frac{2}{\mathfrak{p}}}D}{T^{1-\frac{1}{\mathfrak{p}}}}\right).
$$

*Proof.* By Lemmas 5 and 6, we can follow a similar argument until (44) in the proof of Theorem 10 to have

$$
\mathbb{E}\left[F(\bar{\mathbf{x}}_{T+1}^{\mathrm{cvx}}) - F_\star\right] \leq \frac{D^2}{\eta_{T+1}T} + \frac{2B_T^{\mathrm{cvx}}}{T},
$$

where, under $\eta_t = \eta, \tau_t = \max\left\{\frac{G}{1-\alpha}, \tau T^{\frac{1}{\mathfrak{p}}}\right\}, \forall t \in [T]$ for $\eta, \tau > 0$,

$$
\begin{aligned}
B_T^{\mathrm{cvx}} \leq &\mathcal{O}\left(\eta\left(\inf_{\beta\in[0,1/2]}\frac{(1-\alpha)^{2\beta\mathfrak{p}}\sigma_{\mathfrak{l}}^{2\mathfrak{p}}G^{2-2\beta\mathfrak{p}}T^{2\beta}}{\min\left\{\alpha^{2\mathfrak{p}-2},(1-\alpha)^2\right\}\tau^{2(1-\beta)\mathfrak{p}}} + \frac{G^2}{(1-\alpha)^2} + G^2T\right)\right. \\
&\left. + \eta\left(\sigma_{\mathfrak{l}}^{\mathfrak{p}}\tau^{2-\mathfrak{p}} + \frac{\sigma_{\mathfrak{s}}^2\sigma_{\mathfrak{l}}^{2\mathfrak{p}-2}}{\tau^{2\mathfrak{p}-2}}\right)T^{\frac{2}{\mathfrak{p}}}\right).
\end{aligned}
$$

Now, we plug $\tau = \tilde{\tau}_\star$ (see (36)) into the above inequality to have under the choice of $\tau_t = \max\left\{\frac{G}{1-\alpha}, \tilde{\tau}_\star T^{\frac{1}{\mathfrak{p}}}\right\}, \forall t \in [T]$,

$$
B_T^{\mathrm{cvx}} \leq \mathcal{O}\left(\eta\left(\frac{G^2\tilde{\varphi}^2}{\min\left\{\alpha^{2\mathfrak{p}-2},(1-\alpha)^2\right\}} + \frac{G^2}{(1-\alpha)^2} + G^2T + \sigma_{\mathfrak{s}}^{\frac{4}{\mathfrak{p}}-2}\sigma_{\mathfrak{l}}^{4-\frac{4}{\mathfrak{p}}}T^{\frac{2}{\mathfrak{p}}}\right)\right),
$$

where the first term is obtained by noticing

$$\frac{(1-\alpha)^{2\beta\mathfrak{p}}\sigma_{\mathfrak{l}}^{2\mathfrak{p}}G^{2-2\beta\mathfrak{p}}T^{2\beta}}{\widetilde{\tau}_{\star}^{2(1-\beta)\mathfrak{p}}} = G^2 \cdot \left( (1-\alpha)^{\beta\mathfrak{p}}\frac{\sigma_{\mathfrak{l}}^{(1-\beta)\mathfrak{p}}}{\widetilde{\tau}_{\star}^{(1-\beta)\mathfrak{p}}} \left(\frac{\sigma_{\mathfrak{l}}}{G}\right)^{\beta\mathfrak{p}} T^{\beta} \right)^2$$

$$\overset{(38)}{=} G^2 \cdot \left( (1-\alpha)^{\beta\mathfrak{p}}\widetilde{\varphi}_{\star}^{1-\beta} \left(\frac{\sigma_{\mathfrak{l}}}{G}\right)^{\beta\mathfrak{p}} T^{\beta} \right)^2$$

$$\Rightarrow \inf_{\beta\in[0,1/2]} \frac{(1-\alpha)^{2\beta\mathfrak{p}}\sigma_{\mathfrak{l}}^{2\mathfrak{p}}G^{2-2\beta\mathfrak{p}}T^{2\beta}}{\widetilde{\tau}_{\star}^{2(1-\beta)\mathfrak{p}}} \leq G^2\widetilde{\varphi}^2,$$

in which

$$\widetilde{\varphi} \triangleq \inf_{\beta\in[0,1/2]} (1-\alpha)^{\beta\mathfrak{p}}\widetilde{\varphi}_{\star}^{1-\beta} \left(\frac{\sigma_{\mathfrak{l}}}{G}\right)^{\beta\mathfrak{p}} T^{\beta} = \min\left\{ \widetilde{\varphi}_{\star}, \sqrt{(1-\alpha)^{\mathfrak{p}}\widetilde{\varphi}_{\star} \left(\frac{\sigma_{\mathfrak{l}}}{G}\right)^{\mathfrak{p}} T} \right\} \leq \widetilde{\varphi}_{\star}. \quad (60)$$

Note that we have $\widetilde{\varphi} = \widetilde{\varphi}_{\star}$ when $T \geq \frac{G^{\mathfrak{p}}\widetilde{\varphi}_{\star}}{(1-\alpha)^{\mathfrak{p}}\sigma_{\mathfrak{l}}^{\mathfrak{p}}} = \Omega\left(\frac{G^{\mathfrak{p}}}{\sigma_{\mathfrak{l}}^{\mathfrak{p}}}\widetilde{\varphi}_{\star}\right)$.

By the above results and $\alpha = 1/2$, we find

$$\mathbb{E}\left[F(\bar{\mathbf{x}}_{T+1}^{\mathrm{cvx}}) - F_{\star}\right] \leq \mathcal{O}\left( \frac{D^2}{\eta T} + \frac{\eta\widetilde{\varphi}^2 G^2}{T} + \eta G^2 + \eta\sigma_{\mathsf{s}}^{\frac{4}{\mathfrak{p}}-2}\sigma_{\mathfrak{l}}^{4-\frac{4}{\mathfrak{p}}}T^{\frac{2}{\mathfrak{p}}-1} \right).$$

Plug in $\eta = \eta_{\star}$ (see (59)) to conclude that $\mathbb{E}\left[F(\bar{\mathbf{x}}_{T+1}^{\mathrm{cvx}}) - F_{\star}\right]$ converges at the rate of

$$\mathcal{O}\left( \frac{\widetilde{\varphi}GD}{T} + \frac{GD}{\sqrt{T}} + \frac{\sigma_{\mathsf{s}}^{\frac{2}{\mathfrak{p}}-1}\sigma_{\mathfrak{l}}^{2-\frac{2}{\mathfrak{p}}}D}{T^{1-\frac{1}{\mathfrak{p}}}} \right).$$

$\square$

**Unknown $T$.** Next, we consider the in-expectation convergence for Stabilized Clipped SGD. This anytime rate is also faster than the lower bound $\Omega(\sigma_{\mathfrak{l}}DT^{\frac{1}{\mathfrak{p}}-1})$.

**Theorem 13.** *Under Assumptions 1, 2 (with $\mu = 0$), 3 and 4, for any $T \in \mathbb{N}$, setting $\eta_t = \min\left\{\gamma_{\star}, \frac{\eta_{\star}}{\sqrt{t}}, \frac{\lambda_{\star}}{\widetilde{\tau}_{\star}t^{\frac{1}{\mathfrak{p}}}}\right\}, \tau_t = \max\left\{\frac{G}{1-\alpha}, \widetilde{\tau}_{\star}t^{\frac{1}{\mathfrak{p}}}\right\}, \forall t \in [T]$ where $\alpha = 1/2$,*

$$\gamma_{\star} = \frac{D/G}{\widetilde{\varphi}_{\star}\widetilde{\psi}_{\star} + 1}, \quad \eta_{\star} = D/G, \quad \lambda_{\star} = \frac{D}{\sqrt{\frac{\sigma_{\mathfrak{l}}^{\mathfrak{p}}}{\widetilde{\tau}_{\star}^{\mathfrak{p}}} + \frac{\sigma_{\mathsf{s}}^2\sigma_{\mathfrak{l}}^{2\mathfrak{p}-2}}{\widetilde{\tau}_{\star}^{2\mathfrak{p}}}}}, \quad (61)$$

*and $\widetilde{\psi}_{\star} \triangleq 1 + \ln\widetilde{\varphi}_{\star}$, then Stabilized Clipped SGD (Algorithm 2) guarantees that $\mathbb{E}\left[F(\bar{\mathbf{x}}_{T+1}^{\mathrm{cvx}}) - F_{\star}\right]$ converges at the rate of*

$$\mathcal{O}\left( \frac{\widetilde{\varphi}_{\star}\widetilde{\psi}_{\star}GD}{T} + \frac{GD}{\sqrt{T}} + \frac{\sigma_{\mathsf{s}}^{\frac{2}{\mathfrak{p}}-1}\sigma_{\mathfrak{l}}^{2-\frac{2}{\mathfrak{p}}}D}{T^{1-\frac{1}{\mathfrak{p}}}} \right).$$

*Proof.* By Lemmas 5 and 6, we can follow a similar argument until (56) in the proof of Theorem 11 to have when $\eta_t = \min\left\{\gamma, \frac{\eta}{\sqrt{t}}, \frac{\lambda}{\tau t^{\frac{1}{\mathfrak{p}}}}\right\}, \tau_t = \max\left\{\frac{G}{1-\alpha}, \tau t^{\frac{1}{\mathfrak{p}}}\right\}, \forall t \in [T]$,

$$F(\bar{\mathbf{x}}_{T+1}^{\mathrm{cvx}}) - F_{\star} \leq \mathcal{O}\left( \frac{\frac{D^2}{\gamma} + \gamma\left(\frac{\sigma_{\mathfrak{l}}^{2\mathfrak{p}}G^2(\eta/\gamma)^{\beta}\psi^2(\beta,T)}{\min\{\alpha^{2\mathfrak{p}-2},(1-\alpha)^2\}\tau^{2\mathfrak{p}}} + \frac{G^2}{(1-\alpha)^2}\right)}{T} + \frac{\frac{D^2}{\eta} + \eta G^2}{\sqrt{T}} \right.$$

$$\left. + \frac{\frac{D^2\tau}{\lambda} + \lambda\left(\frac{\sigma_{\mathfrak{l}}^{\mathfrak{p}}}{\tau^{\mathfrak{p}-1}} + \frac{\sigma_{\mathsf{s}}^2\sigma_{\mathfrak{l}}^{2\mathfrak{p}-2}}{\tau^{2\mathfrak{p}-1}}\right)}{T^{1-\frac{1}{\mathfrak{p}}}} \right), \forall \beta \in [0,1],$$

where $\psi(\beta, T) = \begin{cases} 1 + \ln T & \beta = 0 \\ 1 + \frac{4}{\beta} & \beta \in (0, 1] \end{cases}$ is defined in (54).

Finally, we conclude after plugging in $\tau = \widetilde{\tau}_\star, \gamma = \gamma_\star, \eta = \eta_\star, \lambda = \lambda_\star$ (see (36) and (61)), $\alpha = 1/2$, and the following fact:

$$
\inf_{\beta \in [0,1]} \gamma_\star \left( \frac{\eta_\star}{\gamma_\star} \right)^\beta \psi^2(\beta, T) \overset{(61)}{\leq} \frac{D/G}{\widetilde{\varphi}_\star \widetilde{\psi}_\star} \inf_{\beta \in [0,1]} \left( \widetilde{\varphi}_\star \widetilde{\psi}_\star \right)^\beta \psi^2(\beta, T)
$$

$$
\leq \frac{D/G}{\widetilde{\varphi}_\star \widetilde{\psi}_\star} \left( \widetilde{\varphi}_\star \widetilde{\psi}_\star \right)^{\beta_\star} \psi^2(\beta_\star, T) \quad \text{where} \quad \beta_\star = \frac{2}{\max \left\{ \ln \left( \widetilde{\varphi}_\star \widetilde{\psi}_\star \right), 2 \right\}}
$$

$$
\leq \frac{D/G}{\widetilde{\varphi}_\star \widetilde{\psi}_\star} \cdot e^2 \cdot \left( 1 + 2 \max \left\{ \ln \left( \widetilde{\varphi}_\star \widetilde{\psi}_\star \right), 2 \right\} \right)^2
$$

$$
= \mathcal{O} \left( \frac{D/G}{\widetilde{\varphi}_\star \widetilde{\psi}_\star} \cdot \left( 1 + \ln^2 \widetilde{\varphi}_\star + \ln^2 \widetilde{\psi}_\star \right) \right) = \mathcal{O} \left( \frac{D/G}{\widetilde{\varphi}_\star} \cdot \widetilde{\psi}_\star \right),
$$

where the last step is by $\ln \widetilde{\psi}_\star \leq 2 \sqrt{\widetilde{\psi}_\star}$, $1 + \ln^2 \widetilde{\varphi}_\star \leq \widetilde{\psi}_\star^2$ (since $\widetilde{\psi}_\star = 1 + \ln \widetilde{\varphi}_\star$ and $\widetilde{\varphi}_\star \geq 1$), and $\widetilde{\psi}_\star \geq 1$. $\qquad \square$

Compared to Theorem 12, we only incur an extra multiplicative term $\widetilde{\psi}_\star = 1 + \ln \widetilde{\varphi}_\star = 1 + \ln \left( d_{\text{eff}} \mathbb{1} \left[ \mathfrak{p} < 2 \right] \right)$ in the higher-order $\mathcal{O}(T^{-1})$ part.

## E.2 STRONGLY CONVEX CASE

We turn our attention to strongly convex objectives. In this setting, we recall that $\bar{\mathbf{x}}_{T+1}^{\text{str}}$ denotes the following weighted average iterate after $T$ steps:

$$
\bar{\mathbf{x}}_{T+1}^{\text{str}} = \frac{\sum_{t=1}^{T} (t+4)(t+5) \mathbf{x}_{t+1}}{\sum_{t=1}^{T} (t+4)(t+5)}. \tag{62}
$$

### E.2.1 HIGH-PROBABILITY CONVERGENCE

Still, we first consider the high-probability convergence rate. Theorem 14 gives the anytime high-probability rate of Clipped SGD improving upon Liu & Zhou (2023).

**Theorem 14** (Full statement of Theorem 2). *Under Assumptions 1, 2 (with $\mu > 0$), 3 and 4, for any $T \in \mathbb{N}$ and $\delta \in (0, 1]$, setting $\eta_t = \frac{6}{\mu t}, \tau_t = \max \left\{ \frac{G}{1-\alpha}, \tau_\star t^{\frac{1}{\mathfrak{p}}} \right\}, \forall t \in [T]$ where $\alpha = 1/2$, then Clipped SGD (Algorithm 1) guarantees that with probability at least $1 - \delta$, both $F(\bar{\mathbf{x}}_{T+1}^{\text{str}}) - F_\star$ and $\mu \| \mathbf{x}_{T+1} - \mathbf{x}_\star \|^2$ converge at the rate of*

$$
\mathcal{O} \left( \frac{\mu D^2}{T^3} + \frac{\left( \varphi^2 + \ln^2 \frac{3}{\delta} \right) G^2}{\mu T^2} + \frac{G^2}{\mu T} + \frac{\sigma_{\mathfrak{s}}^{\frac{4}{\mathfrak{p}} - 2} \sigma_{\mathfrak{l}}^{4 - \frac{4}{\mathfrak{p}}} + \sigma_{\mathfrak{s}}^{\frac{2}{\mathfrak{p}}} \sigma_{\mathfrak{l}}^{2 - \frac{2}{\mathfrak{p}}} \ln^{2 - \frac{2}{\mathfrak{p}}} \frac{3}{\delta}}{\mu T^{2 - \frac{2}{\mathfrak{p}}}} \right),
$$

*where $\varphi \leq \varphi_\star$ is a constant defined in (46) and equals $\varphi_\star$ when $T = \Omega \left( \frac{G^{\mathfrak{p}}}{\sigma_{\mathfrak{l}}^{\mathfrak{p}}} \varphi_\star \right)$.*

*Proof.* First, the choice of $\eta_t = \frac{6}{\mu t}, \forall t \in [T]$ satisfies $\eta_t \leq \frac{\eta}{\mu}, \forall t \in [T]$ for $\eta = 6$, fulfilling the requirement of Lemma 7. In addition, our choices of $\eta_t$ and $\tau_t$ also meet Conditions 1 and 2 (with $\alpha = 1/2$) in Lemma 9. Therefore, given $T \in \mathbb{N}$ and $\delta \in (0, 1]$, Lemmas 7 and 9 together yield that with probability at least $1 - \delta$,

$$
\frac{\Gamma_{T+1} \| \mathbf{x}_\star - \mathbf{x}_{T+1} \|^2}{2} + \sum_{t=1}^{T} \Gamma_t \eta_t \left( F(\mathbf{x}_{t+1}) - F_\star \right) \leq 4D^2 + 2A_T^{\text{str}}
$$

$$
\Rightarrow \frac{\Gamma_{T+1} \| \mathbf{x}_\star - \mathbf{x}_{T+1} \|^2}{2 \sum_{t=1}^{T} \Gamma_t \eta_t} + \frac{\sum_{t=1}^{T} \Gamma_t \eta_t \left( F(\mathbf{x}_{t+1}) - F_\star \right)}{\sum_{t=1}^{T} \Gamma_t \eta_t} \leq \frac{4D^2 + 2A_T^{\text{str}}}{\sum_{t=1}^{T} \Gamma_t \eta_t}, \tag{63}
$$

where $\Gamma_t = \prod_{s=2}^t \frac{1+\mu\eta_{s-1}}{1+\mu\eta_s/2}$ is introduced in (95) and $A_T^{\mathrm{str}}$ is a constant in the order of

$$
\mathcal{O}\left( \max_{t\in[T]} \Gamma_t \eta_t^2 \tau_t^2 \ln^2 \frac{3}{\delta} + \sum_{t=1}^T \sigma_{\mathfrak{l}}^{\mathfrak{p}} \Gamma_t \eta_t^2 \tau_t^{2-\mathfrak{p}} + \sum_{t=1}^T \left( \sigma_{\mathfrak{s}}^{\mathfrak{p}} \Gamma_t \eta_t^2 \tau_t^{2-\mathfrak{p}} + \frac{\sigma_{\mathfrak{l}}^{\mathfrak{p}} G^2 \Gamma_t \eta_t^2}{\alpha^{\mathfrak{p}-1} \tau_t^{\mathfrak{p}}} \right) \ln \frac{3}{\delta} \right.
$$
$$
\left. + \sum_{t=1}^T \left( \frac{\sigma_{\mathfrak{s}}^2 \sigma_{\mathfrak{l}}^{2\mathfrak{p}-2} \Gamma_t \eta_t}{\tau_t^{2\mathfrak{p}-2}} + \frac{\sigma_{\mathfrak{l}}^{2\mathfrak{p}} G^2 \Gamma_t \eta_t}{\alpha^{2\mathfrak{p}-2} \tau_t^{2\mathfrak{p}}} \right) \frac{1}{\mu} + \sum_{t=1}^T G^2 \Gamma_t \eta_t^2 \right). \tag{64}
$$

We use $\eta_t = \frac{6}{\mu t}, \forall t \in [T]$ to compute

$$
\Gamma_t = \prod_{s=2}^t \frac{1+\mu\eta_{s-1}}{1+\mu\eta_s/2} = \prod_{s=2}^t \frac{s}{s-1} \cdot \frac{s+5}{s+3} = \frac{t(t+4)(t+5)}{30}, \forall t \in [T+1]. \tag{65}
$$

So for any $t \in [T]$,

$$
\Gamma_t \eta_t = \frac{(t+4)(t+5)}{5\mu} \le \frac{6t^2}{\mu} \quad \text{and} \quad \Gamma_t \eta_t^2 = \frac{6(t+4)(t+5)}{5\mu^2 t} \le \frac{36t}{\mu^2}, \tag{66}
$$

implying

$$
\sum_{t=1}^T \Gamma_t \eta_t = \sum_{t=1}^T \frac{(t+4)(t+5)}{5\mu} = \frac{T(T^2 + 15T + 74)}{15\mu}. \tag{67}
$$

Lastly, let us bound (63). For the L.H.S. of (63), we have

$$
\frac{\Gamma_{T+1}}{2\sum_{t=1}^T \Gamma_t \eta_t} \overset{(65),(67)}{=} \frac{\mu(T+1)(T+5)(T+6)}{4T(T^2 + 15T + 74)} \ge \min_{T\in\mathbb{N}} \frac{\mu(T+1)(T+5)(T+6)}{4T(T^2 + 15T + 74)} = \frac{3\mu}{16}.
$$

In addition, we observe that

$$
\frac{\sum_{t=1}^T \Gamma_t \eta_t \mathbf{x}_{t+1}}{\sum_{t=1}^T \Gamma_t \eta_t} \overset{(66)}{=} \frac{\sum_{t=1}^T (t+4)(t+5)\mathbf{x}_{t+1}}{\sum_{t=1}^T (t+4)(t+5)} \overset{(62)}{=} \bar{\mathbf{x}}_{T+1}^{\mathrm{str}}.
$$

The above two results and the convexity of $F$ together lead us to

$$
\frac{3\mu \|\mathbf{x}_\star - \mathbf{x}_{T+1}\|^2}{16} + F(\bar{\mathbf{x}}_{T+1}^{\mathrm{str}}) - F_\star \le \text{L.H.S. of (63)}. \tag{68}
$$

For the R.H.S. of (63), we plug (67) back into (63) to have

$$
\text{R.H.S. of (63)} \le \mathcal{O}\left( \frac{\mu D^2 + \mu A_T^{\mathrm{str}}}{T^3} \right). \tag{69}
$$

One more step, we use (66) to upper bound (64) and obtain

$$
\mu A_T^{\mathrm{str}} \le \frac{1}{\mu} \cdot \mathcal{O}\left( \max_{t\in[T]} \tau_t^2 t \ln^2 \frac{3}{\delta} + \sum_{t=1}^T \sigma_{\mathfrak{l}}^{\mathfrak{p}} \tau_t^{2-\mathfrak{p}} t + \sum_{t=1}^T \left( \sigma_{\mathfrak{s}}^{\mathfrak{p}} \tau_t^{2-\mathfrak{p}} t + \frac{\sigma_{\mathfrak{l}}^{\mathfrak{p}} G^2 t}{\alpha^{\mathfrak{p}-1} \tau_t^{\mathfrak{p}}} \right) \ln \frac{3}{\delta} \right.
$$
$$
\left. + \sum_{t=1}^T \left( \frac{\sigma_{\mathfrak{s}}^2 \sigma_{\mathfrak{l}}^{2\mathfrak{p}-2} t^2}{\tau_t^{2\mathfrak{p}-2}} + \frac{\sigma_{\mathfrak{l}}^{2\mathfrak{p}} G^2 t^2}{\alpha^{2\mathfrak{p}-2} \tau_t^{2\mathfrak{p}}} \right) + G^2 T^2 \right),
$$

When $\tau_t = \max\left\{ \frac{G}{1-\alpha}, \tau t^{\frac{1}{\mathfrak{p}}} \right\}, \forall t \in [T]$, we notice that for any $t \in [T]$,

$$
\sigma_{\mathfrak{l}}^{\mathfrak{p}} \tau_t^{2-\mathfrak{p}} t \le \frac{\sigma_{\mathfrak{l}}^{\mathfrak{p}} G^2 t}{(1-\alpha)^2 \tau_t^{\mathfrak{p}}} + \sigma_{\mathfrak{l}}^{\mathfrak{p}} \tau^{2-\mathfrak{p}} t^{\frac{2}{\mathfrak{p}}} \le \frac{\sigma_{\mathfrak{l}}^{2\mathfrak{p}} G^2 t^2}{\min\{\alpha^{2\mathfrak{p}-2}, (1-\alpha)^2\} \tau_t^{2\mathfrak{p}}} + \frac{G^2}{4(1-\alpha)^2} + \sigma_{\mathfrak{l}}^{\mathfrak{p}} \tau^{2-\mathfrak{p}} t^{\frac{2}{\mathfrak{p}}},
$$
$$
\sigma_{\mathfrak{s}}^{\mathfrak{p}} \tau_t^{2-\mathfrak{p}} t \le \frac{\sigma_{\mathfrak{s}}^{\mathfrak{p}} G^2 t}{(1-\alpha)^2 \tau_t^{\mathfrak{p}}} + \sigma_{\mathfrak{s}}^{\mathfrak{p}} \tau^{2-\mathfrak{p}} t^{\frac{2}{\mathfrak{p}}} \le \frac{\sigma_{\mathfrak{s}}^{2\mathfrak{p}} G^2 t^2}{\min\{\alpha^{2\mathfrak{p}-2}, (1-\alpha)^2\} \tau_t^{2\mathfrak{p}} \ln \frac{3}{\delta}} + \frac{G^2 \ln \frac{3}{\delta}}{4(1-\alpha)^2} + \sigma_{\mathfrak{s}}^{\mathfrak{p}} \tau^{2-\mathfrak{p}} t^{\frac{2}{\mathfrak{p}}},
$$

which implies that

$$
\mu A_T^{\mathrm{str}} \leq \frac{1}{\mu} \cdot \mathcal{O} \left( \frac{G^2 \ln^2 \frac{3}{\delta}}{(1-\alpha)^2} T + \underbrace{\max_{t \in [T]} \tau_t^2 t \ln^2 \frac{3}{\delta}}_{\mathrm{I}} + \underbrace{\sum_{t=1}^T \sigma_\mathfrak{l}^\mathfrak{p} \tau^{2-\mathfrak{p}} t^{\frac{2}{\mathfrak{p}}}}_{\mathrm{II}} + \underbrace{\sum_{t=1}^T \left( \sigma_\mathfrak{s}^\mathfrak{p} \tau^{2-\mathfrak{p}} t^{\frac{2}{\mathfrak{p}}} + \frac{\sigma_\mathfrak{l}^\mathfrak{p} G^2 t}{\alpha^{\mathfrak{p}-1} \tau_t^\mathfrak{p}} \right) \ln \frac{3}{\delta}}_{\mathrm{III}} \right.
$$

$$
\left. + \underbrace{\sum_{t=1}^T \left( \frac{\sigma_\mathfrak{s}^2 \sigma_\mathfrak{l}^{2\mathfrak{p}-2} t^2}{\tau_t^{2\mathfrak{p}-2}} + \frac{\sigma_\mathfrak{l}^{2\mathfrak{p}} G^2 t^2}{\min\left\{\alpha^{2\mathfrak{p}-2}, (1-\alpha)^2\right\} \tau_t^{2\mathfrak{p}}} \right)}_{\mathrm{IV}} + G^2 T^2 \right). \tag{70}
$$

We control the above four terms as follows.

- Term I. We have

$$
\max_{t \in [T]} \tau_t^2 t \ln^2 \frac{3}{\delta} = \tau_T^2 T \ln^2 \frac{3}{\delta} \leq \frac{G^2 \ln^2 \frac{3}{\delta}}{(1-\alpha)^2} T + \tau^2 \ln^2 \left( \frac{3}{\delta} \right) T^{1+\frac{2}{\mathfrak{p}}}. \tag{71}
$$

- Term II. We have

$$
\sum_{t=1}^T \sigma_\mathfrak{l}^\mathfrak{p} \tau^{2-\mathfrak{p}} t^{\frac{2}{\mathfrak{p}}} \leq \sigma_\mathfrak{l}^\mathfrak{p} \tau^{2-\mathfrak{p}} T^{1+\frac{2}{\mathfrak{p}}}. \tag{72}
$$

- Term III. We have

$$
\sum_{t=1}^T \sigma_\mathfrak{s}^\mathfrak{p} \tau^{2-\mathfrak{p}} t^{\frac{2}{\mathfrak{p}}} \leq \sigma_\mathfrak{s}^\mathfrak{p} \tau^{2-\mathfrak{p}} T^{1+\frac{2}{\mathfrak{p}}},
$$

and for any $\beta \in [0, 1/2]$

$$
\sum_{t=1}^T \frac{\sigma_\mathfrak{l}^\mathfrak{p} G^2 t}{\tau_t^\mathfrak{p}} \leq \sum_{t=1}^T \frac{\sigma_\mathfrak{l}^\mathfrak{p} G^2 t}{\left( \frac{G}{1-\alpha} \right)^{\beta \mathfrak{p}} \left( \tau t^{\frac{1}{\mathfrak{p}}} \right)^{(1-\beta)\mathfrak{p}}} \leq \mathcal{O} \left( \frac{(1-\alpha)^{\beta \mathfrak{p}} \sigma_\mathfrak{l}^\mathfrak{p} G^{2-\beta \mathfrak{p}}}{\tau^{(1-\beta)\mathfrak{p}}} T^{1+\beta} \right).
$$

Thus, for any $\beta \in [0, 1/2]$,

$$
\sum_{t=1}^T \left( \sigma_\mathfrak{s}^\mathfrak{p} \tau_t^{2-\mathfrak{p}} t + \frac{\sigma_\mathfrak{l}^\mathfrak{p} G^2 t}{\alpha^{\mathfrak{p}-1} \tau_t^\mathfrak{p}} \right) \ln \frac{3}{\delta}
$$

$$
\leq \mathcal{O} \left( \left( \sigma_\mathfrak{s}^\mathfrak{p} \tau^{2-\mathfrak{p}} T^{1+\frac{2}{\mathfrak{p}}} + \frac{(1-\alpha)^{\beta \mathfrak{p}} \sigma_\mathfrak{l}^\mathfrak{p} G^{2-\beta \mathfrak{p}}}{\alpha^{\mathfrak{p}-1} \tau^{(1-\beta)\mathfrak{p}}} T^{1+\beta} \right) \ln \frac{3}{\delta} \right). \tag{73}
$$

- Term IV. We have

$$
\sum_{t=1}^T \frac{\sigma_\mathfrak{s}^2 \sigma_\mathfrak{l}^{2\mathfrak{p}-2} t^2}{\tau_t^{2\mathfrak{p}-2}} \overset{\mathfrak{p} \geq 1}{\lesssim} \frac{\sigma_\mathfrak{s}^2 \sigma_\mathfrak{l}^{2\mathfrak{p}-2}}{\tau^{2\mathfrak{p}-2}} T^{1+\frac{2}{\mathfrak{p}}},
$$

and for any $\beta \in [0, 1/2]$,

$$
\sum_{t=1}^T \frac{\sigma_\mathfrak{l}^{2\mathfrak{p}} G^2 t^2}{\tau_t^{2\mathfrak{p}}} \leq \sum_{t=1}^T \frac{\sigma_\mathfrak{l}^{2\mathfrak{p}} G^2 t^2}{\left( \frac{G}{1-\alpha} \right)^{2\beta \mathfrak{p}} \left( \tau t^{\frac{1}{\mathfrak{p}}} \right)^{2(1-\beta)\mathfrak{p}}} \leq \mathcal{O} \left( \frac{(1-\alpha)^{2\beta \mathfrak{p}} \sigma_\mathfrak{l}^{2\mathfrak{p}} G^{2-2\beta \mathfrak{p}}}{\tau^{2(1-\beta)\mathfrak{p}}} T^{1+2\beta} \right).
$$

Hence, for any $\beta \in [0, 1/2]$,

$$
\sum_{t=1}^T \left( \frac{\sigma_\mathfrak{s}^2 \sigma_\mathfrak{l}^{2\mathfrak{p}-2} t^2}{\tau_t^{2\mathfrak{p}-2}} + \frac{\sigma_\mathfrak{l}^{2\mathfrak{p}} G^2 t^2}{\min\left\{\alpha^{2\mathfrak{p}-2}, (1-\alpha)^2\right\} \tau_t^{2\mathfrak{p}}} \right)
$$

$$
\leq \mathcal{O} \left( \frac{\sigma_\mathfrak{s}^2 \sigma_\mathfrak{l}^{2\mathfrak{p}-2}}{\tau^{2\mathfrak{p}-2}} T^{1+\frac{2}{\mathfrak{p}}} + \frac{(1-\alpha)^{2\beta \mathfrak{p}} \sigma_\mathfrak{l}^{2\mathfrak{p}} G^{2-2\beta \mathfrak{p}}}{\min\left\{\alpha^{2\mathfrak{p}-2}, (1-\alpha)^2\right\} \tau^{2(1-\beta)\mathfrak{p}}} T^{1+2\beta} \right). \tag{74}
$$

Next, for any fixed $\beta \in [0, 1/2]$,

R.H.S. of (71) + R.H.S. of (74)

$$
= \frac{G^2 \ln^2 \frac{3}{\delta}}{(1-\alpha)^2} T + \frac{(1-\alpha)^{2\beta\mathfrak{p}}\sigma_{\mathfrak{l}}^{2\mathfrak{p}}G^{2-2\beta\mathfrak{p}}}{\min\left\{\alpha^{2\mathfrak{p}-2}, (1-\alpha)^2\right\}\tau^{2(1-\beta)\mathfrak{p}}} T^{1+2\beta} + \tau^2 \ln^2\left(\frac{3}{\delta}\right) T^{1+\frac{2}{\mathfrak{p}}} + \frac{\sigma_{\mathfrak{s}}^2 \sigma_{\mathfrak{l}}^{2\mathfrak{p}-2}}{\tau^{2\mathfrak{p}-2}} T^{1+\frac{2}{\mathfrak{p}}}
$$

$$
\overset{(a)}{\geq} \frac{2(1-\alpha)^{\beta\mathfrak{p}}\sigma_{\mathfrak{l}}^{\mathfrak{p}}G^{2-\beta\mathfrak{p}}\ln\frac{3}{\delta}}{(1-\alpha)\min\left\{\alpha^{\mathfrak{p}-1}, (1-\alpha)\right\}\tau^{(1-\beta)\mathfrak{p}}} T^{1+\beta} + 2\sigma_{\mathfrak{s}}\sigma_{\mathfrak{l}}^{\mathfrak{p}-1}\tau^{2-\mathfrak{p}}\ln\left(\frac{3}{\delta}\right) T^{1+\frac{2}{\mathfrak{p}}}
$$

$$
\overset{(b)}{\geq} \frac{(1-\alpha)^{\beta\mathfrak{p}}\sigma_{\mathfrak{l}}^{\mathfrak{p}}G^{2-\beta\mathfrak{p}}\ln\frac{3}{\delta}}{\alpha^{\mathfrak{p}-1}\tau^{(1-\beta)\mathfrak{p}}} T^{1+\beta} + \sigma_{\mathfrak{s}}^{\mathfrak{p}}\tau^{2-\mathfrak{p}}\ln\left(\frac{3}{\delta}\right) T^{1+\frac{2}{\mathfrak{p}}}
$$

=R.H.S. of (73),

where $(a)$ is by AM-GM inequality and $(b)$ is due to $\alpha < 1$, $\sigma_{\mathfrak{l}} \geq \sigma_{\mathfrak{s}}$ and $\mathfrak{p} \geq 1$. Therefore, after plugging (71), (72), (73), and (74) back into (70), we have for any $\beta \in [0, 1/2]$,

$$
\mu A_T^{\mathrm{str}} \leq \frac{1}{\mu} \cdot \mathcal{O}\left(\left(\frac{(1-\alpha)^{2\beta\mathfrak{p}}\sigma_{\mathfrak{l}}^{2\mathfrak{p}}G^{2-2\beta\mathfrak{p}}T^{2\beta}}{\min\left\{\alpha^{2\mathfrak{p}-2}, (1-\alpha)^2\right\}\tau^{2(1-\beta)\mathfrak{p}}} + \frac{G^2 \ln^2 \frac{3}{\delta}}{(1-\alpha)^2}\right) T + G^2 T^2 \right.
$$

$$
\left. + \left(\tau^2 \ln^2 \frac{3}{\delta} + \sigma_{\mathfrak{l}}^{\mathfrak{p}}\tau^{2-\mathfrak{p}} + \frac{\sigma_{\mathfrak{s}}^2 \sigma_{\mathfrak{l}}^{2\mathfrak{p}-2}}{\tau^{2\mathfrak{p}-2}}\right) T^{1+\frac{2}{\mathfrak{p}}}\right).
$$

Combine the above bound on $\mu A_T^{\mathrm{str}}$ and (69) to have for any $\beta \in [0, 1/2]$,

$$
\text{R.H.S. of (63)} \leq \mathcal{O}\left(\frac{\mu D^2}{T^3} + \frac{\frac{(1-\alpha)^{2\beta\mathfrak{p}}\sigma_{\mathfrak{l}}^{2\mathfrak{p}}G^{2-2\beta\mathfrak{p}}T^{2\beta}}{\min\{\alpha^{2\mathfrak{p}-2}, (1-\alpha)^2\}\tau^{2(1-\beta)\mathfrak{p}}} + \frac{G^2 \ln^2 \frac{3}{\delta}}{(1-\alpha)^2}}{\mu T^2}\right.
$$

$$
\left. + \frac{G^2}{\mu T} + \frac{\tau^2 \ln^2 \frac{3}{\delta} + \sigma_{\mathfrak{l}}^{\mathfrak{p}}\tau^{2-\mathfrak{p}} + \frac{\sigma_{\mathfrak{s}}^2 \sigma_{\mathfrak{l}}^{2\mathfrak{p}-2}}{\tau^{2\mathfrak{p}-2}}}{\mu T^{2-\frac{2}{\mathfrak{p}}}}\right). \tag{75}
$$

We put (68) and (75) together, then use $\alpha = 1/2$ and $\tau = \tau_\star$ (see (32)), and follow the same argument of (46) to finally obtain

$$
\frac{3\mu \|\mathbf{x}_\star - \mathbf{x}_{T+1}\|^2}{16} + F(\bar{\mathbf{x}}_{T+1}^{\mathrm{str}}) - F_\star
$$

$$
\leq \mathcal{O}\left(\frac{\mu D^2}{T^3} + \frac{\left(\varphi^2 + \ln^2 \frac{3}{\delta}\right)G^2}{\mu T^2} + \frac{G^2}{\mu T} + \frac{\sigma_{\mathfrak{s}}^{\frac{4}{\mathfrak{p}}-2}\sigma_{\mathfrak{l}}^{4-\frac{4}{\mathfrak{p}}} + \sigma_{\mathfrak{s}}^{\frac{2}{\mathfrak{p}}}\sigma_{\mathfrak{l}}^{2-\frac{2}{\mathfrak{p}}}\ln^{2-\frac{2}{\mathfrak{p}}}\frac{3}{\delta}}{\mu T^{2-\frac{2}{\mathfrak{p}}}}\right).
$$

$\square$

### E.2.2 IN-EXPECTATION CONVERGENCE

Next, we consider the in-expectation convergence. Note that Theorem 15 is also the first result that breaks the existing lower bound $\Omega(\sigma_{\mathfrak{l}}^2 T^{\frac{2}{\mathfrak{p}}-2})$ (Zhang et al., 2020).

**Theorem 15** (Full statement of Theorem 4). *Under Assumptions 1, 2 (with $\mu > 0$), 3 and 4, for any $T \in \mathbb{N}$, setting $\eta_t = \frac{6}{\mu t}, \tau_t = \max\left\{\frac{G}{1-\alpha}, \widetilde{\tau}_\star t^{\frac{1}{\mathfrak{p}}}\right\}, \forall t \in [T]$ where $\alpha = 1/2$, then Clipped SGD (Algorithm 1) guarantees that both $\mathbb{E}\left[F(\bar{\mathbf{x}}_{T+1}^{\mathrm{str}}) - F_\star\right]$ and $\mu\mathbb{E}\left[\|\mathbf{x}_{T+1} - \mathbf{x}_\star\|^2\right]$ converge at the rate of*

$$
\mathcal{O}\left(\frac{\mu D^2}{T^3} + \frac{\widetilde{\varphi}^2 G^2}{\mu T^2} + \frac{G^2}{\mu T} + \frac{\sigma_{\mathfrak{s}}^{\frac{4}{\mathfrak{p}}-2}\sigma_{\mathfrak{l}}^{4-\frac{4}{\mathfrak{p}}}}{\mu T^{2-\frac{2}{\mathfrak{p}}}}\right),
$$

*where $\widetilde{\varphi} \leq \widetilde{\varphi}_\star$ is a constant defined in (60) and equals $\widetilde{\varphi}_\star$ when $T = \Omega\left(\frac{G^{\mathfrak{p}}}{\sigma_{\mathfrak{l}}^{\mathfrak{p}}}\widetilde{\varphi}_\star\right)$.*

*Proof.* By Lemmas 8 and 9, we can follow a similar argument until (70) in the proof of Theorem 14 to have

$$\frac{3\mu\mathbb{E}\left[\|\mathbf{x}_\star - \mathbf{x}_{T+1}\|^2\right]}{16} + \mathbb{E}\left[F(\bar{\mathbf{x}}_{T+1}^{\text{str}}) - F_\star\right] \leq \mathcal{O}\left(\frac{\mu D^2 + \mu B_T^{\text{str}}}{T^3}\right), \tag{76}$$

where

$$\mu B_T^{\text{str}} \leq \frac{1}{\mu} \cdot \mathcal{O}\left(\sum_{t=1}^T \sigma_l^{\mathfrak{p}} \tau_t^{2-\mathfrak{p}} t + \sum_{t=1}^T \left(\frac{\sigma_s^2 \sigma_l^{2\mathfrak{p}-2} t^2}{\tau_t^{2\mathfrak{p}-2}} + \frac{\sigma_l^{2\mathfrak{p}} G^2 t^2}{\alpha^{2\mathfrak{p}-2} \tau_t^{2\mathfrak{p}}}\right) + G^2 T^2\right).$$

When $\tau_t = \max\left\{\frac{G}{1-\alpha}, \tau t^{\frac{1}{\mathfrak{p}}}\right\}, \forall t \in [T]$, we notice that for any $t \in [T]$,

$$\sigma_l^{\mathfrak{p}} \tau_t^{2-\mathfrak{p}} t \leq \frac{\sigma_l^{\mathfrak{p}} G^2 t}{(1-\alpha)^2 \tau_t^{\mathfrak{p}}} + \sigma_l^{\mathfrak{p}} \tau^{2-\mathfrak{p}} t^{\frac{2}{\mathfrak{p}}} \leq \frac{\sigma_l^{2\mathfrak{p}} G^2 t^2}{\min\left\{\alpha^{2\mathfrak{p}-2}, (1-\alpha)^2\right\} \tau_t^{2\mathfrak{p}}} + \frac{G^2}{4(1-\alpha)^2} + \sigma_l^{\mathfrak{p}} \tau^{2-\mathfrak{p}} t^{\frac{2}{\mathfrak{p}}},$$

which implies that

$$\mu B_T^{\text{str}} \leq \frac{1}{\mu} \cdot \mathcal{O}\left(\frac{G^2 T}{(1-\alpha)^2} + \sum_{t=1}^T \sigma_l^{\mathfrak{p}} \tau^{2-\mathfrak{p}} t^{\frac{2}{\mathfrak{p}}} + \sum_{t=1}^T \left(\frac{\sigma_s^2 \sigma_l^{2\mathfrak{p}-2} t^2}{\tau_t^{2\mathfrak{p}-2}} + \frac{\sigma_l^{2\mathfrak{p}} G^2 t^2}{\min\left\{\alpha^{2\mathfrak{p}-2}, (1-\alpha)^2\right\} \tau_t^{2\mathfrak{p}}}\right) + G^2 T^2\right).$$

We know

$$\sum_{t=1}^T \sigma_l^{\mathfrak{p}} \tau^{2-\mathfrak{p}} t^{\frac{2}{\mathfrak{p}}} \overset{(72)}{\leq} \sigma_l^{\mathfrak{p}} \tau^{2-\mathfrak{p}} T^{1+\frac{2}{\mathfrak{p}}},$$

and for any $\beta \in [0, 1/2]$,

$$\sum_{t=1}^T \left(\frac{\sigma_s^2 \sigma_l^{2\mathfrak{p}-2} t^2}{\tau_t^{2\mathfrak{p}-2}} + \frac{\sigma_l^{2\mathfrak{p}} G^2 t^2}{\min\left\{\alpha^{2\mathfrak{p}-2}, (1-\alpha)^2\right\} \tau_t^{2\mathfrak{p}}}\right)$$
$$\overset{(74)}{\leq} \mathcal{O}\left(\frac{\sigma_s^2 \sigma_l^{2\mathfrak{p}-2}}{\tau^{2\mathfrak{p}-2}} T^{1+\frac{2}{\mathfrak{p}}} + \frac{(1-\alpha)^{2\beta\mathfrak{p}} \sigma_l^{2\mathfrak{p}} G^{2-2\beta\mathfrak{p}}}{\min\left\{\alpha^{2\mathfrak{p}-2}, (1-\alpha)^2\right\} \tau^{2(1-\beta)\mathfrak{p}}} T^{1+2\beta}\right).$$

Therefore, we can bound

$$\mu B_T^{\text{str}} \leq \frac{1}{\mu} \cdot \mathcal{O}\left(\left(\frac{(1-\alpha)^{2\beta\mathfrak{p}} \sigma_l^{2\mathfrak{p}} G^{2-2\beta\mathfrak{p}} T^{2\beta}}{\min\left\{\alpha^{2\mathfrak{p}-2}, (1-\alpha)^2\right\} \tau^{2(1-\beta)\mathfrak{p}}} + \frac{G^2}{(1-\alpha)^2}\right) T + G^2 T^2\right.$$
$$\left. + \left(\sigma_l^{\mathfrak{p}} \tau^{2-\mathfrak{p}} + \frac{\sigma_s^2 \sigma_l^{2\mathfrak{p}-2}}{\tau^{2\mathfrak{p}-2}}\right) T^{1+\frac{2}{\mathfrak{p}}}\right), \forall \beta \in [0, 1/2]. \tag{77}$$

We put (76) and (77) together, then use $\alpha = 1/2$ and $\tau = \tilde{\tau}_\star$ (see (36)), and follow the same argument of (60) to finally obtain

$$\frac{3\mu\mathbb{E}\left[\|\mathbf{x}_\star - \mathbf{x}_{T+1}\|^2\right]}{16} + \mathbb{E}\left[F(\bar{\mathbf{x}}_{T+1}^{\text{str}}) - F_\star\right]$$
$$\leq \mathcal{O}\left(\frac{\mu D^2}{T^3} + \frac{\tilde{\varphi}^2 G^2}{\mu T^2} + \frac{G^2}{\mu T} + \frac{\sigma_s^{\frac{4}{\mathfrak{p}}-2} \sigma_l^{4-\frac{4}{\mathfrak{p}}}}{\mu T^{2-\frac{2}{\mathfrak{p}}}}\right).$$

$\square$

# F THEORETICAL ANALYSIS

This section provides the missing analysis for every lemma used in the proof in Section E. As discussed in Section 5, our refined analysis has two core parts: better application of Freedman's inequality and finer bounds for clipping error.

Before starting, we summarize the frequently used notation in the proof:

- $\mathbf{x}_\star \in \mathbb{X}$, the optimal solution in the domain of the problem $\mathbb{X}$.
- $D = \|\mathbf{x}_\star - \mathbf{x}_1\|$, distance between the optimal solution and the initial point.
- $\mathcal{F}_t = \sigma(\xi_1, \cdots, \xi_t)$, the natural filtration induced by i.i.d. samples $\xi_1$ to $\xi_t$ from $\mathbb{D}$.
- $\mathbf{g}_t = \mathbf{g}(\mathbf{x}_t, \xi_t)$, the stochastic gradient accessed at the $t$-th iteration for point $\mathbf{x}_t$.
- $\tau_t$, the clipping threshold used at the $t$-th iteration.
- $\mathbf{g}_t^{\mathrm{c}} = \mathrm{clip}_{\tau_t}(\mathbf{g}_t) = \min\left\{1, \frac{\tau_t}{\|\mathbf{g}_t\|}\right\} \mathbf{g}_t$, the clipped stochastic gradient.
- $\mathbf{d}_t^{\mathrm{c}} = \mathbf{g}_t^{\mathrm{c}} - \nabla f(\mathbf{x}_t)$, difference between the clipped stochastic gradient and the true gradient.
- $\mathbf{d}_t^{\mathrm{u}} = \mathbf{g}_t^{\mathrm{c}} - \mathbb{E}\left[\mathbf{g}_t^{\mathrm{c}} \mid \mathcal{F}_{t-1}\right]$, the unbiased part in $\mathbf{d}_t^{\mathrm{c}}$.
- $\mathbf{d}_t^{\mathrm{b}} = \mathbb{E}\left[\mathbf{g}_t^{\mathrm{c}} \mid \mathcal{F}_{t-1}\right] - \nabla f(\mathbf{x}_t)$, the biased part in $\mathbf{d}_t^{\mathrm{c}}$.

## F.1 General Lemmas

We give two general lemmas in this subsection.

First, we apply Theorem 9 to obtain the following error bounds specialized for clipped gradient methods. As mentioned, the technical condition required in Theorem 9 automatically holds for clipped gradient methods.

**Lemma 2** (Full statement of Lemma 1). *Under Assumption 4 and assuming $0 < \tau_t \in \mathcal{F}_{t-1}$, then for $\mathbf{d}_t^{\mathrm{u}} = \mathbf{g}_t^{\mathrm{c}} - \mathbb{E}\left[\mathbf{g}_t^{\mathrm{c}} \mid \mathcal{F}_{t-1}\right]$, $\mathbf{d}_t^{\mathrm{b}} = \mathbb{E}\left[\mathbf{g}_t^{\mathrm{c}} \mid \mathcal{F}_{t-1}\right] - \nabla f(\mathbf{x}_t)$, and $\chi_t(\alpha) = \mathbb{1}\left[(1-\alpha)\tau_t \geq \|\nabla f(\mathbf{x}_t)\|\right], \forall \alpha \in [0, 1)$, there are:*

1. $\|\mathbf{d}_t^{\mathrm{u}}\| \leq 2\tau_t$.

2. $\mathbb{E}\left[\|\mathbf{d}_t^{\mathrm{u}}\|^2 \mid \mathcal{F}_{t-1}\right] \leq 4\sigma_{\mathfrak{l}}^{\mathfrak{p}} \tau_t^{2-\mathfrak{p}}$.

3. $\left\|\mathbb{E}\left[\mathbf{d}_t^{\mathrm{u}} \left(\mathbf{d}_t^{\mathrm{u}}\right)^\top \mid \mathcal{F}_{t-1}\right]\right\| \leq 4\sigma_{\mathfrak{s}}^{\mathfrak{p}} \tau_t^{2-\mathfrak{p}} + 4\|\nabla f(\mathbf{x}_t)\|^2$.

4. $\left\|\mathbb{E}\left[\mathbf{d}_t^{\mathrm{u}} \left(\mathbf{d}_t^{\mathrm{u}}\right)^\top \mid \mathcal{F}_{t-1}\right]\right\| \chi_t(\alpha) \leq 4\sigma_{\mathfrak{s}}^{\mathfrak{p}} \tau_t^{2-\mathfrak{p}} + 4\alpha^{1-\mathfrak{p}} \sigma_{\mathfrak{l}}^{\mathfrak{p}} \|\nabla f(\mathbf{x}_t)\|^2 \tau_t^{-\mathfrak{p}}$.

5. $\|\mathbf{d}_t^{\mathrm{b}}\| \leq \sqrt{2}\left(\sigma_{\mathfrak{l}}^{\mathfrak{p}-1} + \|\nabla f(\mathbf{x}_t)\|^{\mathfrak{p}-1}\right)\sigma_{\mathfrak{s}} \tau_t^{1-\mathfrak{p}} + 2\left(\sigma_{\mathfrak{l}}^{\mathfrak{p}} + \|\nabla f(\mathbf{x}_t)\|^{\mathfrak{p}}\right)\|\nabla f(\mathbf{x}_t)\| \tau_t^{-\mathfrak{p}}$.

6. $\|\mathbf{d}_t^{\mathrm{b}}\| \chi_t(\alpha) \leq \sigma_{\mathfrak{s}} \sigma_{\mathfrak{l}}^{\mathfrak{p}-1} \tau_t^{1-\mathfrak{p}} + \alpha^{1-\mathfrak{p}} \sigma_{\mathfrak{l}}^{\mathfrak{p}} \|\nabla f(\mathbf{x}_t)\| \tau_t^{-\mathfrak{p}}$.

*Proof.* We invoke Theorem 9 with $\mathcal{F} = \mathcal{F}_{t-1}$, $\mathbf{g} = \mathbf{g}_t$, $\mathbf{f} = \nabla f(\mathbf{x}_t)$, $\bar{\mathbf{g}} = \mathbf{g}(\mathbf{x}_t, \xi_{t+1})$, $\tau = \tau_t$, $\mathbf{d}^{\mathrm{u}} = \mathbf{d}_t^{\mathrm{u}}$, $\mathbf{d}^{\mathrm{b}} = \mathbf{d}_t^{\mathrm{b}}$, and $\chi(\alpha) = \chi_t(\alpha)$ to conclude. $\square$

Compared to Lemma 1, the clipping threshold $\tau_t$ could be time-varying and random. Inequalities 4 and 6 provide a further (though minor) generalization by a new parameter $\alpha$, which might be useful in practice as mentioned in Remark 11. Especially, setting $\alpha = 1/2$ will recover Lemma 1. Moreover, as discussed in Section 5, Inequalities 2, 4 and 6 are all finer than existing bounds for clipping error under heavy-tailed noise.

We then discuss Inequalities 3 and 5 not provided in Lemma 1. As far as we know, both of them are new in the literature. As one can see, we do not require $\|\nabla f(\mathbf{x}_t)\|$ (which turns out to be $G$ under Assumption 3) to set up $\tau_t$ now, which we believe could be useful for future work.

Next, we give two one-step descent inequalities for our algorithms. The analysis is standard in the literature, which we reproduce here for completeness.

**Lemma 3.** *Under Assumptions 2 and 3, for any $\mathbf{y} \in \mathbb{X}$ and $t \in \mathbb{N}$:*

- *Clipped SGD (Algorithm 1) guarantees*

$$F(\mathbf{x}_{t+1}) - F(\mathbf{y}) \leq \frac{\|\mathbf{y} - \mathbf{x}_t\|^2}{2\eta_t} - \frac{(1 + \mu\eta_t)\|\mathbf{y} - \mathbf{x}_{t+1}\|^2}{2\eta_t} + \langle \mathbf{d}_t^{\mathrm{c}}, \mathbf{y} - \mathbf{x}_t \rangle + \eta_t \|\mathbf{d}_t^{\mathrm{c}}\|^2 + 4\eta_t G^2.$$

- *Stabilized Clipped SGD (Algorithm 2) guarantees, if $\eta_t$ is nonincreasing,*

$$F(\mathbf{x}_{t+1}) - F(\mathbf{y}) \le \frac{\|\mathbf{y} - \mathbf{x}_t\|^2}{2\eta_t} - \frac{(1 + \mu\eta_{t+1}) \|\mathbf{y} - \mathbf{x}_{t+1}\|^2}{2\eta_{t+1}} + \left(\frac{1}{\eta_{t+1}} - \frac{1}{\eta_t}\right)\frac{\|\mathbf{y} - \mathbf{x}_1\|^2}{2}$$
$$+ \langle \mathbf{d}_t^c, \mathbf{y} - \mathbf{x}_t \rangle + \eta_t \|\mathbf{d}_t^c\|^2 + 4\eta_t G^2.$$

*Proof.* By the convexity of $f$,

$$f(\mathbf{x}_{t+1}) - f(\mathbf{x}_t) \le \langle \nabla f(\mathbf{x}_{t+1}), \mathbf{x}_{t+1} - \mathbf{x}_t \rangle$$
$$= \langle \nabla f(\mathbf{x}_{t+1}) - \nabla f(\mathbf{x}_t), \mathbf{x}_{t+1} - \mathbf{x}_t \rangle + \langle \nabla f(\mathbf{x}_t), \mathbf{x}_{t+1} - \mathbf{x}_t \rangle.$$

Recall that $\mathbf{d}_t^c = \mathbf{g}_t^c - \nabla f(\mathbf{x}_t)$, we hence have for any $\mathbf{y} \in \mathbb{X}$,

$$\langle \nabla f(\mathbf{x}_t), \mathbf{x}_{t+1} - \mathbf{x}_t \rangle = \langle \mathbf{d}_t^c, \mathbf{x}_t - \mathbf{x}_{t+1} \rangle + \langle \mathbf{g}_t^c, \mathbf{x}_{t+1} - \mathbf{y} \rangle + \langle \mathbf{d}_t^c, \mathbf{y} - \mathbf{x}_t \rangle + \langle \nabla f(\mathbf{x}_t), \mathbf{y} - \mathbf{x}_t \rangle$$
$$\le \langle \mathbf{d}_t^c, \mathbf{x}_t - \mathbf{x}_{t+1} \rangle + \langle \mathbf{g}_t^c, \mathbf{x}_{t+1} - \mathbf{y} \rangle + \langle \mathbf{d}_t^c, \mathbf{y} - \mathbf{x}_t \rangle + f(\mathbf{y}) - f(\mathbf{x}_t),$$

where the inequality is, again, due to the convexity of $f$. Combine the above two results to obtain

$$f(\mathbf{x}_{t+1}) - f(\mathbf{y}) \le \underbrace{\langle \nabla f(\mathbf{x}_{t+1}) - \nabla f(\mathbf{x}_t), \mathbf{x}_{t+1} - \mathbf{x}_t \rangle}_{\text{I}} + \underbrace{\langle \mathbf{d}_t^c, \mathbf{x}_t - \mathbf{x}_{t+1} \rangle}_{\text{II}}$$
$$+ \underbrace{\langle \mathbf{g}_t^c, \mathbf{x}_{t+1} - \mathbf{y} \rangle}_{\text{III}} + \langle \mathbf{d}_t^c, \mathbf{y} - \mathbf{x}_t \rangle. \tag{78}$$

Next, we bound these three terms separately.

- Term I. By Cauchy-Schwarz inequality, $G$-Lipschitz property of $f$, and AM-GM inequality, there is

$$\langle \nabla f(\mathbf{x}_{t+1}) - \nabla f(\mathbf{x}_t), \mathbf{x}_{t+1} - \mathbf{x}_t \rangle \le \|\nabla f(\mathbf{x}_{t+1}) - \nabla f(\mathbf{x}_t)\| \|\mathbf{x}_{t+1} - \mathbf{x}_t\|$$
$$\le 2G \|\mathbf{x}_{t+1} - \mathbf{x}_t\| \le 4\eta_t G^2 + \frac{\|\mathbf{x}_{t+1} - \mathbf{x}_t\|^2}{4\eta_t}. \tag{79}$$

- Term II. By Cauchy-Schwarz inequality and AM-GM inequality, we know

$$\langle \mathbf{d}_t^c, \mathbf{x}_t - \mathbf{x}_{t+1} \rangle \le \|\mathbf{d}_t^c\| \|\mathbf{x}_{t+1} - \mathbf{x}_t\| \le \eta_t \|\mathbf{d}_t^c\|^2 + \frac{\|\mathbf{x}_{t+1} - \mathbf{x}_t\|^2}{4\eta_t}. \tag{80}$$

- Term III. For Clipped SGD, by the optimality condition of the update rule, there exists $\nabla r(\mathbf{x}_{t+1}) \in \partial r(\mathbf{x}_{t+1})$ such that

$$\left\langle \nabla r(\mathbf{x}_{t+1}) + \mathbf{g}_t^c + \frac{\mathbf{x}_{t+1} - \mathbf{x}_t}{\eta_t}, \mathbf{x}_{t+1} - \mathbf{y} \right\rangle \le 0,$$

which implies

$$\langle \mathbf{g}_t^c, \mathbf{x}_{t+1} - \mathbf{y} \rangle$$
$$\le \frac{1}{\eta_t} \langle \mathbf{x}_t - \mathbf{x}_{t+1}, \mathbf{x}_{t+1} - \mathbf{y} \rangle + \langle \nabla r(\mathbf{x}_{t+1}), \mathbf{y} - \mathbf{x}_{t+1} \rangle$$
$$= \frac{\|\mathbf{y} - \mathbf{x}_t\|^2 - \|\mathbf{y} - \mathbf{x}_{t+1}\|^2 - \|\mathbf{x}_{t+1} - \mathbf{x}_t\|^2}{2\eta_t} + \langle \nabla r(\mathbf{x}_{t+1}), \mathbf{y} - \mathbf{x}_{t+1} \rangle$$
$$\le \frac{\|\mathbf{y} - \mathbf{x}_t\|^2 - \|\mathbf{y} - \mathbf{x}_{t+1}\|^2 - \|\mathbf{x}_{t+1} - \mathbf{x}_t\|^2}{2\eta_t} + r(\mathbf{y}) - r(\mathbf{x}_{t+1}) - \frac{\mu}{2}\|\mathbf{y} - \mathbf{x}_{t+1}\|^2, \tag{81}$$

where the last step is due to the $\mu$-strong convexity of $r$ (Assumption 2). For Stabilized Clipped SGD, a similar argument yields that when $\eta_t \ge \eta_{t+1}$,

$$\langle \mathbf{g}_t^c, \mathbf{x}_{t+1} - \mathbf{y} \rangle \le \frac{\|\mathbf{y} - \mathbf{x}_t\|^2}{2\eta_t} - \frac{\|\mathbf{y} - \mathbf{x}_{t+1}\|^2}{2\eta_{t+1}} - \frac{\|\mathbf{x}_{t+1} - \mathbf{x}_t\|^2}{2\eta_t} + \left(\frac{1}{\eta_{t+1}} - \frac{1}{\eta_t}\right)\frac{\|\mathbf{y} - \mathbf{x}_1\|^2}{2}$$
$$+ r(\mathbf{y}) - r(\mathbf{x}_{t+1}) - \frac{\mu}{2}\|\mathbf{y} - \mathbf{x}_{t+1}\|^2. \tag{82}$$

We plug (79), (80), and (81) (resp. (82)) back into (78) and rearrange terms to obtain the desired result for Clipped SGD (resp. Stabilized Clipped SGD). □

### F.2 LEMMAS FOR GENERAL CONVEX FUNCTIONS

In this section, we focus on the general convex case, i.e., $\mu = 0$ in Assumption 2. As mentioned before in Appendix C, it is enough to only analyze the Stabilized Clipped SGD method since it is the same as the original Clipped SGD when the stepsize is constant.

#### F.2.1 TWO CORE INEQUALITIES

Before moving to the formal proof, we first introduce two quantities that will be used in the analysis:

$$R_t \triangleq \max_{s \in [t]} \frac{\|\mathbf{x}_\star - \mathbf{x}_s\|}{\sqrt{\eta_s}}, \forall t \in [T], \quad \text{and} \quad N_t \triangleq \left\langle \sqrt{\eta_t} \mathbf{d}_t^{\mathrm{u}}, \frac{\mathbf{x}_\star - \mathbf{x}_t}{R_t \sqrt{\eta_t}} \right\rangle, \forall t \in [T]. \tag{83}$$

Note that $R_t \in \mathcal{F}_{t-1}$ and $N_t \in \mathcal{F}_t$ by their definitions. Importantly, $N_t$ is a real-valued MDS due to

$$\mathbb{E}\left[N_t \mid \mathcal{F}_{t-1}\right] = \left\langle \sqrt{\eta_t} \mathbb{E}\left[\mathbf{d}_t^{\mathrm{u}} \mid \mathcal{F}_{t-1}\right], \frac{\mathbf{x}_\star - \mathbf{x}_t}{R_t \sqrt{\eta_t}} \right\rangle = 0, \forall t \in [T]. \tag{84}$$

Now we are ready to dive into the analysis. We first introduce the following Lemma 4, which characterizes the progress made by Stabilized Clipped SGD after $T$ iterations.

**Lemma 4.** *Under Assumptions 1, 2 (with $\mu = 0$) and 3, if $\eta_t$ is nonincreasing, then for any $T \in \mathbb{N}$, Stabilized Clipped SGD (Algorithm 2) guarantees*

$$\frac{\|\mathbf{x}_\star - \mathbf{x}_{T+1}\|^2}{2\eta_{T+1}} + \sum_{t=1}^{T} F(\mathbf{x}_{t+1}) - F_\star \leq \frac{D^2}{\eta_{T+1}} + 2I_T^{\mathrm{cvx}},$$

*where*

$$I_T^{\mathrm{cvx}} \triangleq 8 \max_{t \in [T]} \left(\sum_{s=1}^{t} N_s\right)^2 + 2 \sum_{t=1}^{T} \eta_t \|\mathbf{d}_t^{\mathrm{u}}\|^2 + 4 \left(\sum_{t=1}^{T} \left\|\sqrt{\eta_t} \mathbf{d}_t^{\mathrm{b}}\right\|\right)^2 + 4G^2 \sum_{t=1}^{T} \eta_t.$$

*Proof.* We invoke Lemma 3 for Stabilized Clipped SGD with $\mu = 0$ and $\mathbf{y} = \mathbf{x}_\star$, then replace the subscript $t$ with $s$, and use $\|\mathbf{x}_\star - \mathbf{x}_1\| = D$ to have

$$F(\mathbf{x}_{s+1}) - F_\star \leq \frac{\|\mathbf{x}_\star - \mathbf{x}_s\|^2}{2\eta_s} - \frac{\|\mathbf{x}_\star - \mathbf{x}_{s+1}\|^2}{2\eta_{s+1}} + \left(\frac{1}{\eta_{s+1}} - \frac{1}{\eta_s}\right) \frac{D^2}{2} + \langle \mathbf{d}_s^{\mathrm{c}}, \mathbf{x}_\star - \mathbf{x}_s \rangle + \eta_s \|\mathbf{d}_s^{\mathrm{c}}\|^2 + 4\eta_s G^2,$$

sum up which over $s$ from 1 to $t \leq T$ to obtain

$$\frac{\|\mathbf{x}_\star - \mathbf{x}_{t+1}\|^2}{2\eta_{t+1}} + \sum_{s=1}^{t} F(\mathbf{x}_{s+1}) - F_\star \leq \frac{D^2}{2\eta_{t+1}} + \sum_{s=1}^{t} \langle \mathbf{d}_s^{\mathrm{c}}, \mathbf{x}_\star - \mathbf{x}_s \rangle + \sum_{s=1}^{t} \eta_s \|\mathbf{d}_s^{\mathrm{c}}\|^2 + 4G^2 \sum_{s=1}^{t} \eta_s. \tag{85}$$

We recall the decomposition $\mathbf{d}_s^{\mathrm{c}} = \mathbf{d}_s^{\mathrm{u}} + \mathbf{d}_s^{\mathrm{b}}$ to have

$$\sum_{s=1}^{t} \langle \mathbf{d}_s^{\mathrm{c}}, \mathbf{x}_\star - \mathbf{x}_s \rangle = \sum_{s=1}^{t} \langle \mathbf{d}_s^{\mathrm{u}}, \mathbf{x}_\star - \mathbf{x}_s \rangle + \sum_{s=1}^{t} \langle \mathbf{d}_s^{\mathrm{b}}, \mathbf{x}_\star - \mathbf{x}_s \rangle \overset{(83)}{=} \sum_{s=1}^{t} R_s N_s + \sum_{s=1}^{t} \sqrt{\eta_s} \left\langle \mathbf{d}_s^{\mathrm{b}}, \frac{\mathbf{x}_\star - \mathbf{x}_s}{\sqrt{\eta_s}} \right\rangle.$$

We can bound

$$\sum_{s=1}^{t} R_s N_s \leq \left|\sum_{s=1}^{t} R_s N_s\right| \overset{\text{Lemma 13}}{\leq} 2R_t \max_{S \in [t]} \left|\sum_{s=1}^{S} N_s\right|.$$

In addition, Cauchy-Schwarz inequality gives us

$$\sum_{s=1}^{t} \left\langle \sqrt{\eta_s} \mathbf{d}_s^{\mathrm{b}}, \frac{\mathbf{x}_\star - \mathbf{x}_s}{\sqrt{\eta_s}} \right\rangle \leq \sum_{s=1}^{t} \left\|\sqrt{\eta_s} \mathbf{d}_s^{\mathrm{b}}\right\| \frac{\|\mathbf{x}_\star - \mathbf{x}_s\|}{\sqrt{\eta_s}} \overset{(83)}{\leq} R_t \sum_{s=1}^{t} \left\|\sqrt{\eta_s} \mathbf{d}_s^{\mathrm{b}}\right\|.$$

As such, we know

$$\sum_{s=1}^{t} \langle \mathbf{d}_s^{\mathrm{c}}, \mathbf{x}_\star - \mathbf{x}_s \rangle \le 2R_t \max_{S \in [t]} \left| \sum_{s=1}^{S} N_s \right| + R_t \sum_{s=1}^{t} \left\| \sqrt{\eta_s} \mathbf{d}_s^{\mathrm{b}} \right\|$$

$$\le \frac{R_t^2}{4} + 8 \max_{S \in [t]} \left( \sum_{s=1}^{S} N_s \right)^2 + 2 \left( \sum_{s=1}^{t} \left\| \sqrt{\eta_s} \mathbf{d}_s^{\mathrm{b}} \right\| \right)^2, \tag{86}$$

where the second inequality is by $R_t X \le \frac{R_t^2}{8} + 2X^2$ (due to AM-GM inequality) for $X = 2 \max_{S \in [t]} \left| \sum_{s=1}^{S} N_s \right|$ and $\sum_{s=1}^{t} \left\| \sqrt{\eta_s} \mathbf{d}_s^{\mathrm{b}} \right\|$, respectively.

Plug (86) back into (85) to get

$$\frac{\| \mathbf{x}_\star - \mathbf{x}_{t+1} \|^2}{2\eta_{t+1}} + \sum_{s=1}^{t} F(\mathbf{x}_{s+1}) - F_\star$$

$$\le \frac{R_t^2}{4} + \frac{D^2}{2\eta_{t+1}} + 8 \max_{S \in [t]} \left( \sum_{s=1}^{S} N_s \right)^2 + 2 \left( \sum_{s=1}^{t} \left\| \sqrt{\eta_s} \mathbf{d}_s^{\mathrm{b}} \right\| \right)^2 + \sum_{s=1}^{t} \eta_s \| \mathbf{d}_s^{\mathrm{c}} \|^2 + 4G^2 \sum_{s=1}^{t} \eta_s$$

$$\le \frac{R_t^2}{4} + \frac{D^2}{2\eta_{t+1}} + \underbrace{8 \max_{S \in [t]} \left( \sum_{s=1}^{S} N_s \right)^2 + 2 \sum_{s=1}^{t} \eta_s \| \mathbf{d}_s^{\mathrm{u}} \|^2 + 4 \left( \sum_{s=1}^{t} \left\| \sqrt{\eta_s} \mathbf{d}_s^{\mathrm{b}} \right\| \right)^2 + 4G^2 \sum_{s=1}^{t} \eta_s}_{\triangleq I_t^{\mathrm{cvx}}},$$

$$\tag{87}$$

where the last step is by

$$\sum_{s=1}^{t} \eta_s \| \mathbf{d}_s^{\mathrm{c}} \|^2 = \sum_{s=1}^{t} \eta_s \left\| \mathbf{d}_s^{\mathrm{u}} + \mathbf{d}_s^{\mathrm{b}} \right\|^2 \le 2 \sum_{s=1}^{t} \eta_s \| \mathbf{d}_s^{\mathrm{u}} \|^2 + 2 \sum_{s=1}^{t} \eta_s \left\| \mathbf{d}_s^{\mathrm{b}} \right\|^2$$

$$\le 2 \sum_{s=1}^{t} \eta_s \| \mathbf{d}_s^{\mathrm{u}} \|^2 + 2 \left( \sum_{s=1}^{t} \left\| \sqrt{\eta_s} \mathbf{d}_s^{\mathrm{b}} \right\| \right)^2.$$

Now we let $a_t \triangleq \frac{\| \mathbf{x}_\star - \mathbf{x}_t \|^2}{2\eta_t}, \forall t \in [T+1]$, $b_t \triangleq \sum_{s=1}^{t} F(\mathbf{x}_{s+1}) - F_\star, \forall t \in [T]$ and $c_t \triangleq \frac{D^2}{2\eta_t} + I_{t-1}^{\mathrm{cvx}}, \forall t \in [T+1]$ where $I_0^{\mathrm{cvx}} = 0$. Note that $b_t$ is nonnegative, $c_t$ is nondecreasing as $\eta_t$ is nonincreasing, and

$$a_1 = \frac{\| \mathbf{x}_\star - \mathbf{x}_1 \|^2}{2\eta_1} = \frac{D^2}{2\eta_1} \le \frac{D^2}{\eta_1} = 2c_1.$$

Moreover, (87) is saying that

$$a_{t+1} + b_t \le \frac{\max_{s \in [t]} a_s}{2} + c_{t+1}, \forall t \in [T].$$

Thus, we can invoke Lemma 14 to obtain

$$a_{T+1} + b_T \le 2c_{T+1},$$

which means

$$\frac{\| \mathbf{x}_\star - \mathbf{x}_{T+1} \|^2}{2\eta_{T+1}} + \sum_{t=1}^{T} F(\mathbf{x}_{t+1}) - F_\star \le \frac{D^2}{\eta_{T+1}} + 2I_T^{\mathrm{cvx}}.$$

$\square$

Equipped with Lemma 4, we prove the following in-expectation convergence result for Stabilized Clipped SGD.

**Lemma 5.** *Under the same setting in Lemma 4, Stabilized Clipped SGD (Algorithm 2) guarantees*

$$\frac{\mathbb{E}\left[\|\mathbf{x}_\star - \mathbf{x}_{T+1}\|^2\right]}{2\eta_{T+1}} + \sum_{t=1}^T \mathbb{E}\left[F(\mathbf{x}_{t+1}) - F_\star\right] \leq \frac{D^2}{\eta_{T+1}} + 2J_T^{\mathrm{cvx}},$$

*where*

$$J_T^{\mathrm{cvx}} \triangleq 34 \sum_{t=1}^T \eta_t \mathbb{E}\left[\|\mathbf{d}_t^{\mathrm{u}}\|^2\right] + 4\mathbb{E}\left[\left(\sum_{t=1}^T \|\sqrt{\eta_t}\mathbf{d}_t^{\mathrm{b}}\|\right)^2\right] + 4G^2 \sum_{t=1}^T \eta_t.$$

*Proof.* We invoke Lemma 4 and take expectations to obtain

$$\frac{\mathbb{E}\left[\|\mathbf{x}_\star - \mathbf{x}_{T+1}\|^2\right]}{2\eta_{T+1}} + \sum_{t=1}^T \mathbb{E}\left[F(\mathbf{x}_{t+1}) - F_\star\right] \leq \frac{D^2}{\eta_{T+1}} + 2\mathbb{E}\left[I_T^{\mathrm{cvx}}\right],$$

where, by the definition of $I_T^{\mathrm{cvx}}$,

$$\mathbb{E}\left[I_T^{\mathrm{cvx}}\right] = 8\mathbb{E}\left[\max_{t\in[T]}\left(\sum_{s=1}^t N_s\right)^2\right] + 2\sum_{t=1}^T \eta_t \mathbb{E}\left[\|\mathbf{d}_t^{\mathrm{u}}\|^2\right] + 4\mathbb{E}\left[\left(\sum_{t=1}^T \|\sqrt{\eta_t}\mathbf{d}_t^{\mathrm{b}}\|\right)^2\right] + 4G^2 \sum_{t=1}^T \eta_t.$$

Recall that $N_t, \forall t \in [T]$ is a MDS (see (84)). Therefore, by Lemma 12, there is

$$\mathbb{E}\left[\max_{t\in[T]}\left(\sum_{s=1}^t N_s\right)^2\right] \leq 4\sum_{t=1}^T \mathbb{E}\left[N_s^2\right] \overset{(83)}{\leq} 4\sum_{t=1}^T \eta_t \mathbb{E}\left[\|\mathbf{d}_t^{\mathrm{u}}\|^2\right].$$

Finally, we have

$$\mathbb{E}\left[I_T^{\mathrm{cvx}}\right] \leq 34\sum_{t=1}^T \eta_t \mathbb{E}\left[\|\mathbf{d}_t^{\mathrm{u}}\|^2\right] + 4\mathbb{E}\left[\left(\sum_{t=1}^T \|\sqrt{\eta_t}\mathbf{d}_t^{\mathrm{b}}\|\right)^2\right] + 4G^2 \sum_{t=1}^T \eta_t = J_T^{\mathrm{cvx}}.$$

$\square$

### F.2.2 Bounding Residual Terms

With Lemmas 4 and 5, our next goal is naturally to bound the residual terms $I_T^{\mathrm{cvx}}$ and $J_T^{\mathrm{cvx}}$. Note that the $G^2 \sum_{t=1}^T \eta_t$ part is standard in nonsmooth optimization. Hence, all important things are to control the other terms left.

We now provide the bound in the following Lemma 6, a tighter estimation for the residual term compared to prior works (e.g., Liu & Zhou (2023)), which is achieved due to our finer bounds for clipping error under heavy-tailed noise.

**Lemma 6.** *Under Assumptions 3, 4 and the following two conditions:*

1. *$\eta_t$ and $\tau_t$ are deterministic for all $t \in [T]$.*

2. *$\tau_t \geq \frac{G}{1-\alpha}$ holds for some constant $\alpha \in (0,1)$ and all $t \in [T]$.*

*We have:*

1. *for any $\delta \in (0,1]$, with probability at least $1 - \delta$, $I_T^{\mathrm{cvx}} \leq A_T^{\mathrm{cvx}}$ where $I_T^{\mathrm{cvx}}$ is defined in Lemma 4 and $A_T^{\mathrm{cvx}}$ is a constant in the order of*

$$\mathcal{O}\left(\max_{t\in[T]}\eta_t\tau_t^2\ln^2\frac{3}{\delta} + \sum_{t=1}^T \frac{\sigma_{\mathfrak{l}}^{\mathfrak{p}}\eta_t}{\tau_t^{\mathfrak{p}-2}} + \left(\sum_{t=1}^T \frac{\sigma_{\mathfrak{s}}\sigma_{\mathfrak{l}}^{\mathfrak{p}-1}\sqrt{\eta_t}}{\tau_t^{\mathfrak{p}-1}} + \frac{\sigma_{\mathfrak{l}}^{\mathfrak{p}}G\sqrt{\eta_t}}{\alpha^{\mathfrak{p}-1}\tau_t^{\mathfrak{p}}}\right)^2 + \sum_{t=1}^T G^2\eta_t\right).$$

2. $J_T^{\mathrm{cvx}} \leq B_T^{\mathrm{cvx}}$ where $J_T^{\mathrm{cvx}}$ is defined in Lemma 5 and $B_T^{\mathrm{cvx}}$ is a constant in the order of

$$
\mathcal{O}\left(\sum_{t=1}^{T} \frac{\sigma_{\mathfrak{l}}^{\mathfrak{p}} \eta_t}{\tau_t^{\mathfrak{p}-2}} + \left(\sum_{t=1}^{T} \frac{\sigma_{\mathfrak{s}} \sigma_{\mathfrak{l}}^{\mathfrak{p}-1} \sqrt{\eta_t}}{\tau_t^{\mathfrak{p}-1}} + \frac{\sigma_{\mathfrak{l}}^{\mathfrak{p}} G \sqrt{\eta_t}}{\alpha^{\mathfrak{p}-1} \tau_t^{\mathfrak{p}}}\right)^2 + \sum_{t=1}^{T} G^2 \eta_t\right).
$$

*Proof.* We observe that for any $t \in [T]$, $\tau_t \geq \frac{G}{1-\alpha} \geq \frac{\|\nabla f(\mathbf{x}_t)\|}{1-\alpha}$ holds almost surely due to Condition 2 and Assumption 3, implying that $\chi_t(\alpha)$ in Lemma 2 equals 1 for all $t \in [T]$. Then Lemma 2 and Assumption 3 together yield the following inequalities holding for any $t \in [T]$:

$$
\|\sqrt{\eta_t} \mathbf{d}_t^{\mathrm{u}}\| \overset{\text{Inequality 1}}{\leq} 2\sqrt{\eta_t} \tau_t \leq 2 \max_{t \in [T]} \sqrt{\eta_t} \tau_t, \tag{88}
$$

$$
\mathbb{E}\left[\|\sqrt{\eta_t} \mathbf{d}_t^{\mathrm{u}}\|^2 \mid \mathcal{F}_{t-1}\right] \overset{\text{Inequality 2}}{\leq} \frac{4\sigma_{\mathfrak{l}}^{\mathfrak{p}} \eta_t}{\tau_t^{\mathfrak{p}-2}}, \tag{89}
$$

$$
\left\|\mathbb{E}\left[\eta_t \mathbf{d}_t^{\mathrm{u}} (\mathbf{d}_t^{\mathrm{u}})^{\top} \mid \mathcal{F}_{t-1}\right]\right\| \overset{\text{Inequality 4}}{\leq} \frac{4\sigma_{\mathfrak{s}}^{\mathfrak{p}} \eta_t}{\tau_t^{\mathfrak{p}-2}} + \frac{4\sigma_{\mathfrak{l}}^{\mathfrak{p}} G^2 \eta_t}{\alpha^{\mathfrak{p}-1} \tau_t^{\mathfrak{p}}}, \tag{90}
$$

$$
\|\sqrt{\eta_t} \mathbf{d}_t^{\mathrm{b}}\| \overset{\text{Inequality 6}}{\leq} \frac{\sigma_{\mathfrak{s}} \sigma_{\mathfrak{l}}^{\mathfrak{p}-1} \sqrt{\eta_t}}{\tau_t^{\mathfrak{p}-1}} + \frac{\sigma_{\mathfrak{l}}^{\mathfrak{p}} G \sqrt{\eta_t}}{\alpha^{\mathfrak{p}-1} \tau_t^{\mathfrak{p}}}. \tag{91}
$$

We first bound $I_T^{\mathrm{cvx}}$ in high probability.

- Recall that $N_t = \left\langle \sqrt{\eta_t} \mathbf{d}_t^{\mathrm{u}}, \frac{\mathbf{x}_\star - \mathbf{x}_t}{R_t \sqrt{\eta_t}} \right\rangle, \forall t \in [T]$ is a real-valued MDS (see (84)), whose absolute value can be bounded by Cauchy-Schwarz inequality

$$
|N_t| \leq \|\sqrt{\eta_t} \mathbf{d}_t^{\mathrm{u}}\| \left\|\frac{\mathbf{x}_\star - \mathbf{x}_t}{R_t \sqrt{\eta_t}}\right\| \overset{(83)}{\leq} \|\sqrt{\eta_t} \mathbf{d}_t^{\mathrm{u}}\| \overset{(88)}{\leq} 2 \max_{t \in [T]} \sqrt{\eta_t} \tau_t.
$$

Moreover, its conditional variance can be controlled by

$$
\mathbb{E}\left[N_t^2 \mid \mathcal{F}_{t-1}\right] = \left(\frac{\mathbf{x}_\star - \mathbf{x}_t}{R_t \sqrt{\eta_t}}\right)^{\top} \mathbb{E}\left[\eta_t \mathbf{d}_t^{\mathrm{u}} (\mathbf{d}_t^{\mathrm{u}})^{\top} \mid \mathcal{F}_{t-1}\right] \frac{\mathbf{x}_\star - \mathbf{x}_t}{R_t \sqrt{\eta_t}}
$$

$$
\overset{(83)}{\leq} \left\|\mathbb{E}\left[\eta_t \mathbf{d}_t^{\mathrm{u}} (\mathbf{d}_t^{\mathrm{u}})^{\top} \mid \mathcal{F}_{t-1}\right]\right\| \overset{(90)}{\leq} \frac{4\sigma_{\mathfrak{s}}^{\mathfrak{p}} \eta_t}{\tau_t^{\mathfrak{p}-2}} + \frac{4\sigma_{\mathfrak{l}}^{\mathfrak{p}} G^2 \eta_t}{\alpha^{\mathfrak{p}-1} \tau_t^{\mathfrak{p}}}.
$$

Therefore, Freedman's inequality (Lemma 10) gives that with probability at least $1 - 2\delta/3$,

$$
\left|\sum_{s=1}^{t} N_s\right| \leq \frac{4}{3} \max_{t \in [T]} \sqrt{\eta_t} \tau_t \ln \frac{3}{\delta} + \sqrt{8 \sum_{s=1}^{T} \left(\frac{\sigma_{\mathfrak{s}}^{\mathfrak{p}} \eta_s}{\tau_s^{\mathfrak{p}-2}} + \frac{\sigma_{\mathfrak{l}}^{\mathfrak{p}} G^2 \eta_s}{\alpha^{\mathfrak{p}-1} \tau_s^{\mathfrak{p}}}\right) \ln \frac{3}{\delta}}, \forall t \in [T],
$$

which implies

$$
\max_{t \in [T]} \left(\sum_{s=1}^{t} N_s\right)^2 \leq \frac{2^5}{9} \max_{t \in [T]} \eta_t \tau_t^2 \ln^2 \frac{3}{\delta} + 16 \sum_{t=1}^{T} \left(\frac{\sigma_{\mathfrak{s}}^{\mathfrak{p}} \eta_t}{\tau_t^{\mathfrak{p}-2}} + \frac{\sigma_{\mathfrak{l}}^{\mathfrak{p}} G^2 \eta_t}{\alpha^{\mathfrak{p}-1} \tau_t^{\mathfrak{p}}}\right) \ln \frac{3}{\delta}. \tag{92}
$$

- Note that $\|\sqrt{\eta_t} \mathbf{d}_t^{\mathrm{u}}\|, \forall t \in [T]$ is a sequence of random variables satisfying

$$
\|\sqrt{\eta_t} \mathbf{d}_t^{\mathrm{u}}\| \overset{(88)}{\leq} 2 \max_{t \in [T]} \sqrt{\eta_t} \tau_t \quad \text{and} \quad \mathbb{E}\left[\|\sqrt{\eta_t} \mathbf{d}_t^{\mathrm{u}}\|^2 \mid \mathcal{F}_{t-1}\right] \overset{(89)}{\leq} \frac{4\sigma_{\mathfrak{l}}^{\mathfrak{p}} \eta_t}{\tau_t^{\mathfrak{p}-2}}.
$$

Then by Lemma 11, we have with probability at least $1 - \delta/3$,

$$
\sum_{t=1}^{T} \eta_t \|\mathbf{d}_t^{\mathrm{u}}\|^2 \leq \frac{14}{3} \max_{t \in [T]} \eta_t \tau_t^2 \ln \frac{3}{\delta} + 8 \sum_{t=1}^{T} \frac{\sigma_{\mathfrak{l}}^{\mathfrak{p}} \eta_t}{\tau_t^{\mathfrak{p}-2}} \overset{\ln \frac{3}{\delta} \geq 1}{\leq} \frac{14}{3} \max_{t \in [T]} \eta_t \tau_t^2 \ln^2 \frac{3}{\delta} + 8 \sum_{t=1}^{T} \frac{\sigma_{\mathfrak{l}}^{\mathfrak{p}} \eta_t}{\tau_t^{\mathfrak{p}-2}}. \tag{93}
$$

- Lastly, there is

$$\sum_{t=1}^{T} \left\| \sqrt{\eta_t} \mathbf{d}_t^{\mathrm{b}} \right\| \overset{(91)}{\leq} \sum_{t=1}^{T} \frac{\sigma_{\mathfrak{s}} \sigma_{\mathfrak{l}}^{\mathfrak{p}-1} \sqrt{\eta_t}}{\tau_t^{\mathfrak{p}-1}} + \frac{\sigma_{\mathfrak{l}}^{\mathfrak{p}} G \sqrt{\eta_t}}{\alpha^{\mathfrak{p}-1} \tau_t^{\mathfrak{p}}}. \tag{94}$$

Combine (92), (93) and (94) to have with probability at least $1 - \delta$,

$$I_T^{\mathrm{cvx}} = 8 \max_{t \in [T]} \left( \sum_{s=1}^{t} N_s \right)^2 + 2 \sum_{t=1}^{T} \eta_t \left\| \mathbf{d}_t^{\mathrm{u}} \right\|^2 + 4 \left( \sum_{t=1}^{T} \left\| \sqrt{\eta_t} \mathbf{d}_t^{\mathrm{b}} \right\| \right)^2 + 4G^2 \sum_{t=1}^{T} \eta_t \leq A_T^{\mathrm{cvx}},$$

where

$$A_T^{\mathrm{cvx}} \triangleq \left( \frac{2^8}{9} + \frac{28}{3} \right) \max_{t \in [T]} \eta_t \tau_t^2 \ln^2 \frac{3}{\delta} + 16 \sum_{t=1}^{T} \frac{\sigma_{\mathfrak{l}}^{\mathfrak{p}} \eta_t}{\tau_t^{\mathfrak{p}-2}} + 128 \sum_{t=1}^{T} \left( \frac{\sigma_{\mathfrak{s}}^{\mathfrak{p}} \eta_t}{\tau_t^{\mathfrak{p}-2}} + \frac{\sigma_{\mathfrak{l}}^{\mathfrak{p}} G^2 \eta_t}{\alpha^{\mathfrak{p}-1} \tau_t^{\mathfrak{p}}} \right) \ln \frac{3}{\delta}$$

$$+ 4 \left( \sum_{t=1}^{T} \frac{\sigma_{\mathfrak{s}} \sigma_{\mathfrak{l}}^{\mathfrak{p}-1} \sqrt{\eta_t}}{\tau_t^{\mathfrak{p}-1}} + \frac{\sigma_{\mathfrak{l}}^{\mathfrak{p}} G \sqrt{\eta_t}}{\alpha^{\mathfrak{p}-1} \tau_t^{\mathfrak{p}}} \right)^2 + 4G^2 \sum_{t=1}^{T} \eta_t$$

$$= \mathcal{O} \left( \max_{t \in [T]} \eta_t \tau_t^2 \ln^2 \frac{3}{\delta} + \sum_{t=1}^{T} \frac{\sigma_{\mathfrak{l}}^{\mathfrak{p}} \eta_t}{\tau_t^{\mathfrak{p}-2}} + \sum_{t=1}^{T} \left( \frac{\sigma_{\mathfrak{s}}^{\mathfrak{p}} \eta_t}{\tau_t^{\mathfrak{p}-2}} + \frac{\sigma_{\mathfrak{l}}^{\mathfrak{p}} G^2 \eta_t}{\alpha^{\mathfrak{p}-1} \tau_t^{\mathfrak{p}}} \right) \ln \frac{3}{\delta} \right.$$

$$\left. + \left( \sum_{t=1}^{T} \frac{\sigma_{\mathfrak{s}} \sigma_{\mathfrak{l}}^{\mathfrak{p}-1} \sqrt{\eta_t}}{\tau_t^{\mathfrak{p}-1}} + \frac{\sigma_{\mathfrak{l}}^{\mathfrak{p}} G \sqrt{\eta_t}}{\alpha^{\mathfrak{p}-1} \tau_t^{\mathfrak{p}}} \right)^2 + \sum_{t=1}^{T} G^2 \eta_t \right).$$

Note that by AM-GM inequality

$$\max_{t \in [T]} \eta_t \tau_t^2 \ln^2 \frac{3}{\delta} + \left( \sum_{t=1}^{T} \frac{\sigma_{\mathfrak{s}} \sigma_{\mathfrak{l}}^{\mathfrak{p}-1} \sqrt{\eta_t}}{\tau_t^{\mathfrak{p}-1}} + \frac{\sigma_{\mathfrak{l}}^{\mathfrak{p}} G \sqrt{\eta_t}}{\alpha^{\mathfrak{p}-1} \tau_t^{\mathfrak{p}}} \right)^2$$

$$\geq 2 \left( \max_{t \in [T]} \sqrt{\eta_t} \tau_t \right) \sum_{t=1}^{T} \left( \frac{\sigma_{\mathfrak{s}} \sigma_{\mathfrak{l}}^{\mathfrak{p}-1} \sqrt{\eta_t}}{\tau_t^{\mathfrak{p}-1}} + \frac{\sigma_{\mathfrak{l}}^{\mathfrak{p}} G \sqrt{\eta_t}}{\alpha^{\mathfrak{p}-1} \tau_t^{\mathfrak{p}}} \right) \ln \frac{3}{\delta}$$

$$\overset{(a)}{\geq} 2 \sum_{t=1}^{T} \left( \frac{\sigma_{\mathfrak{s}} \sigma_{\mathfrak{l}}^{\mathfrak{p}-1} \eta_t}{\tau_t^{\mathfrak{p}-2}} + \frac{\sigma_{\mathfrak{l}}^{\mathfrak{p}} G^2 \eta_t}{(1-\alpha) \alpha^{\mathfrak{p}-1} \tau_t^{\mathfrak{p}}} \right) \ln \frac{3}{\delta} \overset{(b)}{\geq} 2 \sum_{t=1}^{T} \left( \frac{\sigma_{\mathfrak{s}}^{\mathfrak{p}} \eta_t}{\tau_t^{\mathfrak{p}-2}} + \frac{\sigma_{\mathfrak{l}}^{\mathfrak{p}} G^2 \eta_t}{\alpha^{\mathfrak{p}-1} \tau_t^{\mathfrak{p}}} \right) \ln \frac{3}{\delta},$$

where $(a)$ is by $\tau_t \geq \frac{G}{1-\alpha}$ in Condition 2 and $(b)$ is due to $\sigma_{\mathfrak{l}} \geq \sigma_{\mathfrak{s}}$, $\mathfrak{p} > 1$ and $\alpha \in (0,1)$. Hence, the order of $A_T^{\mathrm{cvx}}$ can be simplified into

$$\mathcal{O} \left( \max_{t \in [T]} \eta_t \tau_t^2 \ln^2 \frac{3}{\delta} + \sum_{t=1}^{T} \frac{\sigma_{\mathfrak{l}}^{\mathfrak{p}} \eta_t}{\tau_t^{\mathfrak{p}-2}} + \left( \sum_{t=1}^{T} \frac{\sigma_{\mathfrak{s}} \sigma_{\mathfrak{l}}^{\mathfrak{p}-1} \sqrt{\eta_t}}{\tau_t^{\mathfrak{p}-1}} + \frac{\sigma_{\mathfrak{l}}^{\mathfrak{p}} G \sqrt{\eta_t}}{\alpha^{\mathfrak{p}-1} \tau_t^{\mathfrak{p}}} \right)^2 + \sum_{t=1}^{T} G^2 \eta_t \right).$$

Now let us bound $J_T^{\mathrm{cvx}}$. It can be done directly via (89) and (91). Hence, we omit the detail and claim $J_T^{\mathrm{cvx}} \leq B_T^{\mathrm{cvx}}$, where $B_T^{\mathrm{cvx}}$ is a constant in the order of

$$\mathcal{O} \left( \sum_{t=1}^{T} \frac{\sigma_{\mathfrak{l}}^{\mathfrak{p}} \eta_t}{\tau_t^{\mathfrak{p}-2}} + \left( \sum_{t=1}^{T} \frac{\sigma_{\mathfrak{s}} \sigma_{\mathfrak{l}}^{\mathfrak{p}-1} \sqrt{\eta_t}}{\tau_t^{\mathfrak{p}-1}} + \frac{\sigma_{\mathfrak{l}}^{\mathfrak{p}} G \sqrt{\eta_t}}{\alpha^{\mathfrak{p}-1} \tau_t^{\mathfrak{p}}} \right)^2 + \sum_{t=1}^{T} G^2 \eta_t \right).$$

$\square$

### F.3 LEMMAS FOR STRONGLY CONVEX FUNCTIONS

In this section, we move to the strongly convex case, i.e., $\mu > 0$ in Assumption 2. The algorithm that we study is Clipped SGD.

### F.3.1 TWO CORE INEQUALITIES

We begin by introducing some notations that will be used later:

$$\Gamma_t \triangleq \prod_{s=2}^{t} \frac{1 + \mu\eta_{s-1}}{1 + \mu\eta_s/2}, \forall t \in [T+1], \tag{95}$$

which satisfies the equation

$$\Gamma_t(1 + \mu\eta_t) = \Gamma_{t+1}(1 + \mu\eta_{t+1}/2), \forall t \in [T]. \tag{96}$$

Equipped with $\Gamma_t$, we redefine

$$R_t \triangleq \max_{s \in [t]} \sqrt{\Gamma_s(1 + \mu\eta_s/2)} \|\mathbf{x}_\star - \mathbf{x}_s\|, \forall t \in [T], \tag{97}$$

$$N_t \triangleq \left\langle \sqrt{\frac{\Gamma_t}{1 + \mu\eta_t/2}} \eta_t \mathbf{d}_t^{\mathrm{u}}, \frac{\sqrt{\Gamma_t(1 + \mu\eta_t/2)}(\mathbf{x}_\star - \mathbf{x}_t)}{R_t} \right\rangle, \forall t \in [T]. \tag{98}$$

By their definitions, $R_t \in \mathcal{F}_{t-1}$ and $N_t \in \mathcal{F}_t$. Moreover, $N_t$ is still a MDS due to

$$\mathbb{E}[N_t \mid \mathcal{F}_{t-1}] = \left\langle \sqrt{\frac{\Gamma_t}{1 + \mu\eta_t/2}} \eta_t \mathbb{E}[\mathbf{d}_t^{\mathrm{u}} \mid \mathcal{F}_{t-1}], \frac{\sqrt{\Gamma_t(1 + \mu\eta_t/2)}(\mathbf{x}_\star - \mathbf{x}_t)}{R_t} \right\rangle = 0, \forall t \in [T]. \tag{99}$$

Again, we first show the progress made by Clipped SGD after $T$ steps in the following Lemma 7.

**Lemma 7.** *Under Assumptions 1, 2 (with $\mu > 0$) and 3, if $\eta_t \leq \frac{\eta}{\mu}$ for some constant $\eta > 0$, then for any $T \in \mathbb{N}$, Clipped SGD (Algorithm 1) guarantees*

$$\frac{\Gamma_{T+1}\|\mathbf{x}_\star - \mathbf{x}_{T+1}\|^2}{2} + \sum_{t=1}^{T} \Gamma_t\eta_t (F(\mathbf{x}_{t+1}) - F_\star) \leq (1 + \eta/2)D^2 + 2I_T^{\mathrm{str}},$$

*where*

$$I_T^{\mathrm{str}} \triangleq 4 \max_{t \in [T]} \left( \sum_{s=1}^{t} N_s \right)^2 + 2\sum_{t=1}^{T} \Gamma_t\eta_t^2 \|\mathbf{d}_t^{\mathrm{u}}\|^2 + \frac{2\eta+1}{\mu} \sum_{t=1}^{T} \Gamma_t\eta_t \|\mathbf{d}_t^{\mathrm{b}}\|^2 + 4G^2 \sum_{t=1}^{T} \Gamma_t\eta_t^2.$$

*Proof.* We invoke Lemma 3 for Clipped SGD with $\mu > 0$ and $\mathbf{y} = \mathbf{x}_\star$, then replace the subscript $t$ with $s$, and multiply both sides by $\Gamma_s\eta_s$ to have

$$\Gamma_s\eta_s (F(\mathbf{x}_{s+1}) - F_\star)$$

$$\leq \frac{\Gamma_s\|\mathbf{x}_\star - \mathbf{x}_s\|^2}{2} - \frac{\Gamma_s(1 + \mu\eta_s)\|\mathbf{x}_\star - \mathbf{x}_{s+1}\|^2}{2} + \langle\Gamma_s\eta_s\mathbf{d}_s^{\mathrm{c}}, \mathbf{x}_\star - \mathbf{x}_s\rangle + \Gamma_s\eta_s^2 \|\mathbf{d}_s^{\mathrm{c}}\|^2 + 4\Gamma_s\eta_s^2 G^2$$

$$\overset{(96)}{=} \frac{\Gamma_s\|\mathbf{x}_\star - \mathbf{x}_s\|^2}{2} - \frac{\Gamma_{s+1}(1 + \mu\eta_{s+1}/2)\|\mathbf{x}_\star - \mathbf{x}_{s+1}\|^2}{2} + \langle\Gamma_s\eta_s\mathbf{d}_s^{\mathrm{c}}, \mathbf{x}_\star - \mathbf{x}_s\rangle + \Gamma_s\eta_s^2 \|\mathbf{d}_s^{\mathrm{c}}\|^2 + 4\Gamma_s\eta_s^2 G^2,$$

sum up which over $s$ from 1 to $t \leq T$ to obtain

$$\frac{\Gamma_{t+1}(1 + \mu\eta_{t+1}/2)\|\mathbf{x}_\star - \mathbf{x}_{t+1}\|^2}{2} + \sum_{s=1}^{t} \Gamma_s\eta_s (F(\mathbf{x}_{s+1}) - F_\star)$$

$$\leq \frac{\Gamma_1\|\mathbf{x}_\star - \mathbf{x}_1\|^2}{2} - \frac{\mu}{4}\sum_{s=2}^{t} \Gamma_s\eta_s \|\mathbf{x}_\star - \mathbf{x}_s\|^2 + \sum_{s=1}^{t} \langle\Gamma_s\eta_s\mathbf{d}_s^{\mathrm{c}}, \mathbf{x}_\star - \mathbf{x}_s\rangle + \sum_{s=1}^{t} \Gamma_s\eta_s^2 \|\mathbf{d}_s^{\mathrm{c}}\|^2 + 4G^2\sum_{s=1}^{t} \Gamma_s\eta_s^2$$

$$= \frac{D^2}{2} - \frac{\mu}{4}\sum_{s=2}^{t} \Gamma_s\eta_s \|\mathbf{x}_\star - \mathbf{x}_s\|^2 + \sum_{s=1}^{t} \langle\Gamma_s\eta_s\mathbf{d}_s^{\mathrm{c}}, \mathbf{x}_\star - \mathbf{x}_s\rangle + \sum_{s=1}^{t} \Gamma_s\eta_s^2 \|\mathbf{d}_s^{\mathrm{c}}\|^2 + 4G^2\sum_{s=1}^{t} \Gamma_s\eta_s^2, \tag{100}$$

where the last step holds by $\Gamma_1 = 1$ and $\|\mathbf{x}_\star - \mathbf{x}_1\| = D$.

We recall the decomposition $\mathbf{d}_s^{\mathrm{c}} = \mathbf{d}_s^{\mathrm{u}} + \mathbf{d}_s^{\mathrm{b}}$ to have

$$\sum_{s=1}^{t} \langle \Gamma_s \eta_s \mathbf{d}_s^{\mathrm{c}}, \mathbf{x}_\star - \mathbf{x}_s \rangle = \sum_{s=1}^{t} \langle \Gamma_s \eta_s \mathbf{d}_s^{\mathrm{u}}, \mathbf{x}_\star - \mathbf{x}_s \rangle + \sum_{s=1}^{t} \langle \Gamma_s \eta_s \mathbf{d}_s^{\mathrm{b}}, \mathbf{x}_\star - \mathbf{x}_s \rangle$$

$$\stackrel{(97),(98)}{=} \sum_{s=1}^{t} R_s N_s + \sum_{s=1}^{t} \langle \Gamma_s \eta_s \mathbf{d}_s^{\mathrm{b}}, \mathbf{x}_\star - \mathbf{x}_s \rangle.$$

By Lemma 13 and AM-GM inequality, there is

$$\sum_{s=1}^{t} R_s N_s \leq 2 R_t \max_{S \in [t]} \left| \sum_{s=1}^{S} N_s \right| \leq \frac{R_t^2}{4} + 4 \max_{S \in [t]} \left( \sum_{s=1}^{S} N_s \right)^2.$$

In addition, we use Cauchy-Schwarz inequality and AM-GM inequality to bound

$$\sum_{s=1}^{t} \langle \Gamma_s \eta_s \mathbf{d}_s^{\mathrm{b}}, \mathbf{x}_\star - \mathbf{x}_s \rangle \leq \sum_{s=1}^{t} \Gamma_s \eta_s \left\| \mathbf{d}_s^{\mathrm{b}} \right\| \left\| \mathbf{x}_\star - \mathbf{x}_s \right\| \leq \sum_{s=1}^{t} \frac{\Gamma_s \eta_s \left\| \mathbf{d}_s^{\mathrm{b}} \right\|^2}{\mu} + \frac{\mu \Gamma_s \eta_s \left\| \mathbf{x}_\star - \mathbf{x}_s \right\|^2}{4}.$$

As such, we obtain

$$\sum_{s=1}^{t} \langle \Gamma_s \eta_s \mathbf{d}_s^{\mathrm{c}}, \mathbf{x}_\star - \mathbf{x}_s \rangle \leq \frac{R_t^2}{4} + 4 \max_{S \in [t]} \left( \sum_{s=1}^{S} N_s \right)^2 + \sum_{s=1}^{t} \frac{\Gamma_s \eta_s \left\| \mathbf{d}_s^{\mathrm{b}} \right\|^2}{\mu} + \frac{\mu \Gamma_s \eta_s \left\| \mathbf{x}_\star - \mathbf{x}_s \right\|^2}{4}.$$
$$(101)$$

Plug (101) back into (100) to get

$$\frac{\Gamma_{t+1}(1 + \mu \eta_{t+1}/2) \left\| \mathbf{x}_\star - \mathbf{x}_{t+1} \right\|^2}{2} + \sum_{s=1}^{t} \Gamma_s \eta_s \left( F(\mathbf{x}_{s+1}) - F_\star \right)$$

$$\leq \frac{R_t^2}{4} + \frac{(1 + \mu \eta_1/2)D^2}{2} + 4 \max_{S \in [t]} \left( \sum_{s=1}^{S} N_s \right)^2 + \sum_{s=1}^{t} \frac{\Gamma_s \eta_s \left\| \mathbf{d}_s^{\mathrm{b}} \right\|^2}{\mu} + \sum_{s=1}^{t} \Gamma_s \eta_s^2 \left\| \mathbf{d}_s^{\mathrm{c}} \right\|^2 + 4G^2 \sum_{s=1}^{t} \Gamma_s \eta_s^2$$

$$\leq \frac{R_t^2}{4} + \frac{(1 + \eta/2)D^2}{2}$$

$$+ \underbrace{4 \max_{S \in [t]} \left( \sum_{s=1}^{S} N_s \right)^2 + 2 \sum_{s=1}^{t} \Gamma_s \eta_s^2 \left\| \mathbf{d}_s^{\mathrm{u}} \right\|^2 + \frac{2\eta + 1}{\mu} \sum_{s=1}^{t} \Gamma_s \eta_s \left\| \mathbf{d}_s^{\mathrm{b}} \right\|^2 + 4G^2 \sum_{s=1}^{t} \Gamma_s \eta_s^2}_{\triangleq I_t^{\mathrm{str}}}, \quad (102)$$

where the last step is by $\eta_1 \leq \eta/\mu$ and

$$\sum_{s=1}^{t} \Gamma_s \eta_s^2 \left\| \mathbf{d}_s^{\mathrm{c}} \right\|^2 = \sum_{s=1}^{t} \Gamma_s \eta_s^2 \left\| \mathbf{d}_s^{\mathrm{u}} + \mathbf{d}_s^{\mathrm{b}} \right\|^2 \leq 2 \sum_{s=1}^{t} \Gamma_s \eta_s^2 \left\| \mathbf{d}_s^{\mathrm{u}} \right\|^2 + 2 \sum_{s=1}^{t} \Gamma_s \eta_s^2 \left\| \mathbf{d}_s^{\mathrm{b}} \right\|^2$$

$$\stackrel{\eta_s \leq \eta/\mu, \forall s \in [T]}{\leq} 2 \sum_{s=1}^{t} \Gamma_s \eta_s^2 \left\| \mathbf{d}_s^{\mathrm{u}} \right\|^2 + \frac{2\eta}{\mu} \sum_{s=1}^{t} \Gamma_s \eta_s \left\| \mathbf{d}_s^{\mathrm{b}} \right\|^2.$$

Now we let $a_t \triangleq \frac{\Gamma_t(1 + \mu \eta_t/2) \left\| \mathbf{x}_\star - \mathbf{x}_t \right\|^2}{2}, \forall t \in [T+1]$, $b_t \triangleq \sum_{s=1}^{t} \Gamma_s \eta_s \left( F(\mathbf{x}_{s+1}) - F_\star \right), \forall t \in [T]$ and $c_t \triangleq \frac{(1 + \eta/2)D^2}{2} + I_{t-1}^{\mathrm{str}}, \forall t \in [T+1]$, where $I_0^{\mathrm{str}} = 0$. Note that $b_t$ is nonnegative, $c_t$ is nondecreasing, and

$$a_1 = \frac{\Gamma_1(1 + \mu \eta_1/2) \left\| \mathbf{x}_\star - \mathbf{x}_1 \right\|^2}{2} \leq \frac{(1 + \eta/2)D^2}{2} \leq (1 + \eta/2)D^2 = 2c_1.$$

Moreover, (102) is saying that

$$a_{t+1} + b_t \leq \frac{\max_{s \in [t]} a_s}{2} + c_{t+1}, \forall t \in [T].$$

Thus, we can invoke Lemma 14 to obtain

$$a_{T+1} + b_T \leq 2c_{T+1},$$

which means

$$\frac{\Gamma_{T+1}(1 + \mu\eta_{T+1}/2)\|\mathbf{x}_\star - \mathbf{x}_{T+1}\|^2}{2} + \sum_{t=1}^{T}\Gamma_t\eta_t\left(F(\mathbf{x}_{t+1}) - F_\star\right) \leq (1 + \eta/2)D^2 + 2I_T^{\mathrm{str}}.$$

Finally, we conclude from $\mu\eta_{T+1} \geq 0$. $\qquad\square$

Equipped with Lemma 7, we prove the following in-expectation convergence result for Clipped SGD under strong convexity.

**Lemma 8.** *Under the same setting in Lemma 7, Clipped SGD (Algorithm 1) guarantees*

$$\frac{\Gamma_{T+1}\mathbb{E}\left[\|\mathbf{x}_\star - \mathbf{x}_{T+1}\|^2\right]}{2} + \sum_{t=1}^{T}\Gamma_t\eta_t\mathbb{E}\left[F(\mathbf{x}_{t+1}) - F_\star\right] \leq (1 + \eta/2)D^2 + 2J_T^{\mathrm{str}},$$

*where*

$$J_T^{\mathrm{str}} \triangleq 18\sum_{t=1}^{T}\Gamma_t\eta_t^2\mathbb{E}\left[\|\mathbf{d}_t^{\mathrm{u}}\|^2\right] + \frac{2\eta + 1}{\mu}\sum_{t=1}^{T}\Gamma_t\eta_t\mathbb{E}\left[\|\mathbf{d}_t^{\mathrm{b}}\|^2\right] + 4G^2\sum_{t=1}^{T}\Gamma_t\eta_t^2.$$

*Proof.* Similar to the proof of Lemma 5, we take expectations on both sides of Lemma 7 and then invoke Lemma 12. The calculations are omitted here to save space. $\qquad\square$

### F.3.2 Bounding Residual Terms

Like previously, we need to upper bound $I_T^{\mathrm{str}}$ and $J_T^{\mathrm{str}}$, which is done in the following lemma.

**Lemma 9.** *Under Assumptions 3, 4 and the following two conditions:*

1. *$\eta_t$ and $\tau_t$ are deterministic for all $t \in [T]$.*

2. *$\tau_t \geq \frac{G}{1-\alpha}$ holds for some constant $\alpha \in (0,1)$ and all $t \in [T]$.*

*We have:*

1. *for any $\delta \in (0,1]$, with probability at least $1 - \delta$, $I_T^{\mathrm{str}} \leq A_T^{\mathrm{str}}$ where $I_T^{\mathrm{str}}$ is defined in Lemma 7 and $A_T^{\mathrm{str}}$ is a constant in the order of*

$$\mathcal{O}\left(\max_{t\in[T]}\Gamma_t\eta_t^2\tau_t^2\ln^2\frac{3}{\delta} + \sum_{t=1}^{T}\frac{\sigma_{\mathfrak{l}}^{\mathfrak{p}}\Gamma_t\eta_t^2}{\tau_t^{\mathfrak{p}-2}} + \sum_{t=1}^{T}\left(\frac{\sigma_{\mathfrak{s}}^{\mathfrak{p}}\Gamma_t\eta_t^2}{\tau_t^{\mathfrak{p}-2}} + \frac{\sigma_{\mathfrak{l}}^{\mathfrak{p}}G^2\Gamma_t\eta_t^2}{\alpha^{\mathfrak{p}-1}\tau_t^{\mathfrak{p}}}\right)\ln\frac{3}{\delta}\right.$$
$$\left. + \sum_{t=1}^{T}\left(\frac{\sigma_{\mathfrak{s}}^2\sigma_{\mathfrak{l}}^{2\mathfrak{p}-2}\Gamma_t\eta_t}{\tau_t^{2\mathfrak{p}-2}} + \frac{\sigma_{\mathfrak{l}}^{2\mathfrak{p}}G^2\Gamma_t\eta_t}{\alpha^{2\mathfrak{p}-2}\tau_t^{2\mathfrak{p}}}\right)\frac{2\eta + 1}{\mu} + \sum_{t=1}^{T}G^2\Gamma_t\eta_t^2\right).$$

2. *$J_T^{\mathrm{str}} \leq B_T^{\mathrm{str}}$ where $J_T^{\mathrm{str}}$ is defined in Lemma 8 and $B_T^{\mathrm{str}}$ is a constant in the order of*

$$\mathcal{O}\left(\sum_{t=1}^{T}\frac{\sigma_{\mathfrak{l}}^{\mathfrak{p}}\Gamma_t\eta_t^2}{\tau_t^{\mathfrak{p}-2}} + \sum_{t=1}^{T}\left(\frac{\sigma_{\mathfrak{s}}^2\sigma_{\mathfrak{l}}^{2\mathfrak{p}-2}\Gamma_t\eta_t}{\tau_t^{2\mathfrak{p}-2}} + \frac{\sigma_{\mathfrak{l}}^{2\mathfrak{p}}G^2\Gamma_t\eta_t}{\alpha^{2\mathfrak{p}-2}\tau_t^{2\mathfrak{p}}}\right)\frac{2\eta + 1}{\mu} + \sum_{t=1}^{T}G^2\Gamma_t\eta_t^2\right).$$

*Proof.* We observe that for any $t \in [T]$, $\tau_t \geq \frac{G}{1-\alpha} \geq \frac{\|\nabla f(\mathbf{x}_t)\|}{1-\alpha}$ holds almost surely due to Condition 2 and Assumption 3, implying that $\chi_t(\alpha)$ in Lemma 2 equals 1 for all $t \in [T]$. Then Lemma 2 and

Assumption 3 together yield the following inequalities holding for any $t \in [T]$:

$$\sqrt{\Gamma_t} \eta_t \left\| \mathbf{d}_t^{\mathrm{u}} \right\| \overset{\text{Inequality 1}}{\leq} 2\sqrt{\Gamma_t} \eta_t \tau_t \leq 2 \max_{t \in [T]} \sqrt{\Gamma_t} \eta_t \tau_t, \tag{103}$$

$$\mathbb{E}\left[ \Gamma_t \eta_t^2 \left\| \mathbf{d}_t^{\mathrm{u}} \right\|^2 \mid \mathcal{F}_{t-1} \right] \overset{\text{Inequality 2}}{\leq} \frac{4\sigma_{\mathfrak{l}}^{\mathfrak{p}} \Gamma_t \eta_t^2}{\tau_t^{\mathfrak{p}-2}}, \tag{104}$$

$$\left\| \mathbb{E}\left[ \Gamma_t \eta_t^2 \mathbf{d}_t^{\mathrm{u}} \left( \mathbf{d}_t^{\mathrm{u}} \right)^{\top} \mid \mathcal{F}_{t-1} \right] \right\| \overset{\text{Inequality 4}}{\leq} \frac{4\sigma_{\mathfrak{s}}^{\mathfrak{p}} \Gamma_t \eta_t^2}{\tau_t^{\mathfrak{p}-2}} + \frac{4\sigma_{\mathfrak{l}}^{\mathfrak{p}} G^2 \Gamma_t \eta_t^2}{\alpha^{\mathfrak{p}-1} \tau_t^{\mathfrak{p}}}, \tag{105}$$

$$\sqrt{\Gamma_t} \eta_t \left\| \mathbf{d}_t^{\mathrm{b}} \right\| \overset{\text{Inequality 6}}{\leq} \frac{\sigma_{\mathfrak{s}} \sigma_{\mathfrak{l}}^{\mathfrak{p}-1} \sqrt{\Gamma_t} \eta_t}{\tau_t^{\mathfrak{p}-1}} + \frac{\sigma_{\mathfrak{l}}^{\mathfrak{p}} G \sqrt{\Gamma_t} \eta_t}{\alpha^{\mathfrak{p}-1} \tau_t^{\mathfrak{p}}}. \tag{106}$$

- Similar to (92), we can prove now with probability at least $1 - 2\delta/3$,

$$\max_{t \in [T]} \left( \sum_{s=1}^{t} N_s \right)^2 \leq \frac{2^5}{9} \max_{t \in [T]} \Gamma_t \eta_t^2 \tau_t^2 \ln^2 \frac{3}{\delta} + 16 \sum_{t=1}^{T} \left( \frac{\sigma_{\mathfrak{s}}^{\mathfrak{p}} \Gamma_t \eta_t^2}{\tau_t^{\mathfrak{p}-2}} + \frac{\sigma_{\mathfrak{l}}^{\mathfrak{p}} G^2 \Gamma_t \eta_t^2}{\alpha^{\mathfrak{p}-1} \tau_t^{\mathfrak{p}}} \right) \ln \frac{3}{\delta}. \tag{107}$$

- Similar to (93), we can prove now with probability at least $1 - \delta/3$,

$$\sum_{t=1}^{T} \Gamma_t \eta_t^2 \left\| \mathbf{d}_t^{\mathrm{u}} \right\|^2 \leq \frac{14}{3} \max_{t \in [T]} \Gamma_t \eta_t^2 \tau_t^2 \ln^2 \frac{3}{\delta} + 8 \sum_{t=1}^{T} \frac{\sigma_{\mathfrak{l}}^{\mathfrak{p}} \Gamma_t \eta_t^2}{\tau_t^{\mathfrak{p}-2}}. \tag{108}$$

- Lastly, there is

$$\sum_{t=1}^{T} \Gamma_t \eta_t \left\| \mathbf{d}_t^{\mathrm{b}} \right\|^2 \overset{(106)}{\leq} \sum_{t=1}^{T} \left( \frac{\sigma_{\mathfrak{s}} \sigma_{\mathfrak{l}}^{\mathfrak{p}-1} \sqrt{\Gamma_t} \eta_t}{\tau_t^{\mathfrak{p}-1}} + \frac{\sigma_{\mathfrak{l}}^{\mathfrak{p}} G \sqrt{\Gamma_t} \eta_t}{\alpha^{\mathfrak{p}-1} \tau_t^{\mathfrak{p}}} \right)^2$$

$$\leq \sum_{t=1}^{T} \left( \frac{2\sigma_{\mathfrak{s}}^2 \sigma_{\mathfrak{l}}^{2\mathfrak{p}-2} \Gamma_t \eta_t}{\tau_t^{2\mathfrak{p}-2}} + \frac{2\sigma_{\mathfrak{l}}^{2\mathfrak{p}} G^2 \Gamma_t \eta_t}{\alpha^{2\mathfrak{p}-2} \tau_t^{2\mathfrak{p}}} \right). \tag{109}$$

We combine (107), (108) and (109) to have with probability at least $1 - \delta$,

$$I_T^{\mathrm{str}} = 4 \max_{t \in [T]} \left( \sum_{s=1}^{t} N_s \right)^2 + 2 \sum_{t=1}^{T} \Gamma_t \eta_t^2 \left\| \mathbf{d}_t^{\mathrm{u}} \right\|^2 + \frac{2\eta + 1}{\mu} \sum_{t=1}^{T} \Gamma_t \eta_t \left\| \mathbf{d}_t^{\mathrm{b}} \right\|^2 + 4G^2 \sum_{t=1}^{T} \Gamma_t \eta_t^2 \leq A_T^{\mathrm{str}},$$

where

$$A_T^{\mathrm{str}} \triangleq \left( \frac{2^7}{9} + \frac{28}{3} \right) \max_{t \in [T]} \Gamma_t \eta_t^2 \tau_t^2 \ln^2 \frac{3}{\delta} + 16 \sum_{t=1}^{T} \frac{\sigma_{\mathfrak{l}}^{\mathfrak{p}} \Gamma_t \eta_t^2}{\tau_t^{\mathfrak{p}-2}} + 64 \sum_{t=1}^{T} \left( \frac{\sigma_{\mathfrak{s}}^{\mathfrak{p}} \Gamma_t \eta_t^2}{\tau_t^{\mathfrak{p}-2}} + \frac{\sigma_{\mathfrak{l}}^{\mathfrak{p}} G^2 \Gamma_t \eta_t^2}{\alpha^{\mathfrak{p}-1} \tau_t^{\mathfrak{p}}} \right) \ln \frac{3}{\delta}$$

$$+ \sum_{t=1}^{T} \left( \frac{2\sigma_{\mathfrak{s}}^2 \sigma_{\mathfrak{l}}^{2\mathfrak{p}-2} \Gamma_t \eta_t}{\tau_t^{2\mathfrak{p}-2}} + \frac{2\sigma_{\mathfrak{l}}^{2\mathfrak{p}} G^2 \Gamma_t \eta_t}{\alpha^{2\mathfrak{p}-2} \tau_t^{2\mathfrak{p}}} \right) \frac{2\eta + 1}{\mu} + 4 \sum_{t=1}^{T} G^2 \Gamma_t \eta_t^2$$

$$= \mathcal{O}\left( \max_{t \in [T]} \Gamma_t \eta_t^2 \tau_t^2 \ln^2 \frac{3}{\delta} + \sum_{t=1}^{T} \frac{\sigma_{\mathfrak{l}}^{\mathfrak{p}} \Gamma_t \eta_t^2}{\tau_t^{\mathfrak{p}-2}} + \sum_{t=1}^{T} \left( \frac{\sigma_{\mathfrak{s}}^{\mathfrak{p}} \Gamma_t \eta_t^2}{\tau_t^{\mathfrak{p}-2}} + \frac{\sigma_{\mathfrak{l}}^{\mathfrak{p}} G^2 \Gamma_t \eta_t^2}{\alpha^{\mathfrak{p}-1} \tau_t^{\mathfrak{p}}} \right) \ln \frac{3}{\delta} \right.$$

$$\left. + \sum_{t=1}^{T} \left( \frac{\sigma_{\mathfrak{s}}^2 \sigma_{\mathfrak{l}}^{2\mathfrak{p}-2} \Gamma_t \eta_t}{\tau_t^{2\mathfrak{p}-2}} + \frac{\sigma_{\mathfrak{l}}^{2\mathfrak{p}} G^2 \Gamma_t \eta_t}{\alpha^{2\mathfrak{p}-2} \tau_t^{2\mathfrak{p}}} \right) \frac{2\eta + 1}{\mu} + \sum_{t=1}^{T} G^2 \Gamma_t \eta_t^2 \right).$$

Now let us bound $J_T^{\mathrm{str}}$. It can be done directly via (104) and (106). Hence, we omit the detail and claim $J_T^{\mathrm{str}} \leq B_T^{\mathrm{str}}$, where $B_T^{\mathrm{str}}$ is a constant in the order of

$$\mathcal{O}\left( \sum_{t=1}^{T} \frac{\sigma_{\mathfrak{l}}^{\mathfrak{p}} \Gamma_t \eta_t^2}{\tau_t^{\mathfrak{p}-2}} + \sum_{t=1}^{T} \left( \frac{\sigma_{\mathfrak{s}}^2 \sigma_{\mathfrak{l}}^{2\mathfrak{p}-2} \Gamma_t \eta_t}{\tau_t^{2\mathfrak{p}-2}} + \frac{\sigma_{\mathfrak{l}}^{2\mathfrak{p}} G^2 \Gamma_t \eta_t}{\alpha^{2\mathfrak{p}-2} \tau_t^{2\mathfrak{p}}} \right) \frac{2\eta + 1}{\mu} + \sum_{t=1}^{T} G^2 \Gamma_t \eta_t^2 \right).$$

$\square$

### F.4 EXISTING TECHNICAL RESULTS

This section contains some technical results existing (or implicitly used) in prior works.

First, Lemma 10 is the famous Freedman's inequality, a useful tool to bound a real-valued MDS.

**Lemma 10** (Freedman's inequality (Freedman, 1975)). *Suppose $X_t \in \mathbb{R}, \forall t \in [T]$ is a real-valued MDS adapted to the filtration $\mathcal{F}_t, \forall t \in \{0\} \cup [T]$ satisfying for any $t \in [T]$, $X_t \leq b$ and $\mathbb{E}\left[X_t^2 \mid \mathcal{F}_{t-1}\right] \leq \sigma_t^2$ almost surely, where $b \geq 0$ and $\sigma_t^2$ are both constant, then for any $\delta \in (0, 1]$, there is*

$$\Pr\left[\sum_{s=1}^{t} X_s \leq \frac{2b}{3} \ln \frac{1}{\delta} + \sqrt{2 \sum_{s=1}^{T} \sigma_s^2 \ln \frac{1}{\delta}}, \forall t \in [T]\right] \geq 1 - \delta.$$

Next, Lemma 11 is another concentration inequality. This is not a new result, and similar ideas were used before in, e.g., Cutkosky & Mehta (2021); Zhang & Cutkosky (2022); Liu & Zhou (2023). We provide a proof here to make the work self-contained.

**Lemma 11.** *Suppose $X_t \in \mathbb{R}, \forall t \in [T]$ is a sequence of random variables adapted to the filtration $\mathcal{F}_t, \forall t \in \{0\} \cup [T]$ satisfying for any $t \in [T]$, $|X_t| \leq b$ and $\mathbb{E}\left[X_t^2 \mid \mathcal{F}_{t-1}\right] \leq \sigma_t^2$ almost surely, where $b \geq 0$ and $\sigma_t^2$ are both constant, then for any $\delta \in (0, 1]$, there is*

$$\Pr\left[\sum_{t=1}^{T} X_t^2 \leq \frac{7b^2}{6} \ln \frac{1}{\delta} + 2 \sum_{t=1}^{T} \sigma_t^2\right] \geq 1 - \delta.$$

*Proof.* Note that we can bound

$$\sum_{t=1}^{T} X_t^2 = \sum_{t=1}^{T} \underbrace{X_t^2 - \mathbb{E}\left[X_t^2 \mid \mathcal{F}_{t-1}\right]}_{\triangleq Y_t} + \sum_{t=1}^{T} \mathbb{E}\left[X_t^2 \mid \mathcal{F}_{t-1}\right] \leq \sum_{t=1}^{T} Y_t + \sum_{t=1}^{T} \sigma_t^2.$$

Observe that $Y_t, \forall t \in [T]$ is a real-valued MDS adapted to the filtration $\mathcal{F}_t, \forall t \in \{0\} \cup [T]$ satisfying

$$Y_t \leq X_t^2 \leq b^2 \quad \text{and} \quad \mathbb{E}\left[Y_t^2 \mid \mathcal{F}_{t-1}\right] \leq \mathbb{E}\left[X_t^4 \mid \mathcal{F}_{t-1}\right] \leq b^2 \sigma_t^2.$$

Then Lemma 10 yields that, for any $\delta \in (0, 1]$, we have with probability at least $1 - \delta$,

$$\sum_{s=1}^{t} Y_s \leq \frac{2b^2}{3} \ln \frac{1}{\delta} + \sqrt{2 \sum_{s=1}^{T} b^2 \sigma_s^2 \ln \frac{1}{\delta}}, \forall t \in [T],$$

which implies

$$\sum_{t=1}^{T} Y_t \leq \frac{2b^2}{3} \ln \frac{1}{\delta} + \sqrt{2 \sum_{t=1}^{T} b^2 \sigma_t^2 \ln \frac{1}{\delta}} \leq \frac{7b^2}{6} \ln \frac{1}{\delta} + \sum_{t=1}^{T} \sigma_t^2$$

where the last step is by $\sqrt{2 \sum_{t=1}^{T} b^2 \sigma_t^2 \ln \frac{1}{\delta}} \leq \frac{b^2}{2} \ln \frac{1}{\delta} + \sum_{t=1}^{T} \sigma_t^2$ due to AM-GM inequality. Hence, it follows that

$$\Pr\left[\sum_{t=1}^{T} X_t^2 \leq \frac{7b^2}{6} \ln \frac{1}{\delta} + 2 \sum_{t=1}^{T} \sigma_t^2\right] \geq 1 - \delta.$$

$\square$

The following Lemma 12 is the famous Doob's $L^2$ maximum inequality. For its proof, see, e.g., Theorem 4.4.4 in Durrett (2019).

**Lemma 12** (Doob's $L^2$ maximum inequality). *Suppose $X_t \in \mathbb{R}, \forall t \in [T]$ is a real-valued MDS, then there is*

$$\mathbb{E}\left[\max_{t \in [T]}\left(\sum_{s=1}^{t} X_s\right)^2\right] \leq 4 \sum_{t=1}^{T} \mathbb{E}\left[X_t^2\right].$$

In addition, we need the following algebraic fact in our analysis.

**Lemma 13** (Lemma C.2 in Ivgi et al. (2023))**.** *Let $a_1, \cdots, a_T$ and $b_1, \cdots, b_T$ be two sequences in $\mathbb{R}$ such that $a_t$ is nonnegative and nondecreasing, then there is*

$$\left| \sum_{s=1}^{t} a_s b_s \right| \leq 2 a_t \max_{S \in [t]} \left| \sum_{s=1}^{S} b_s \right|, \forall t \in [T].$$

Lastly, we introduce another algebraic inequality, the idea behind which can also be found in previous works like Ivgi et al. (2023); Liu & Zhou (2023). For completeness, we produce a proof here.

**Lemma 14.** *Let $a_1, \cdots, a_{T+1}$, $b_1, \cdots, b_T$ and $c_1, \cdots, c_{T+1}$ be three sequences in $\mathbb{R}$ such that $b_t$ is nonnegative and $c_t$ is nondecreasing, if $a_1 \leq 2c_1$ and*

$$a_{t+1} + b_t \leq \frac{\max_{s \in [t]} a_s}{2} + c_{t+1}, \forall t \in [T],$$

*then there is*

$$a_{T+1} + b_T \leq 2c_{T+1}.$$

*Proof.* We first use induction to show

$$a_t \leq 2c_t, \forall t \in [T]. \tag{110}$$

For the base case $t = 1$, we know $a_1 \leq 2c_1$ by the assumption. Suppose (110) holds for all time not greater than $t$ for some $t \in [T-1]$. Then for time $t+1$, we know

$$a_{t+1} \overset{b_t \geq 0}{\leq} a_{t+1} + b_t \leq \frac{\max_{s \in [t]} a_s}{2} + c_{t+1} \overset{(110)}{\leq} \frac{\max_{s \in [t]} 2c_s}{2} + c_{t+1} \leq 2c_{t+1},$$

where the last inequality holds because $c_t$ is nondecreasing. Therefore, (110) is true by induction. Hence, we know

$$a_{T+1} + b_T \leq \frac{\max_{s \in [T]} a_s}{2} + c_{T+1} \overset{(110)}{\leq} \frac{\max_{s \in [T]} 2c_s}{2} + c_{T+1} \leq 2c_{T+1},$$

where the last step is also because $c_t$ is nondecreasing. □

## G    FULL THEOREMS FOR LOWER BOUNDS AND PROOFS

Please refer to Appendix G in the full version of this work.

## H    NUMERICAL SIMULATIONS

In this section, we provide some numerical simulations to support our theory. We limit our attention to the additive noise model, i.e., $\mathbf{g}(\mathbf{x}, \xi) = \nabla f(\mathbf{x}) + \xi$, where all coordinates $\xi_i$ are assumed to be i.i.d. Moreover, we denote by $\sigma \triangleq \left( \mathbb{E}\left[ |\xi_1|^{\mathfrak{p}} \right] \right)^{\frac{1}{\mathfrak{p}}}$.

**Objective.** We pick $\mathbb{X} = \mathbb{R}^d$, $f(\mathbf{x}) = \|\mathbf{x} - \mathbf{y}\|_1$ for some $\mathbf{y} \in \mathbb{R}^d$, and $r(\mathbf{x}) = 0$. Therefore, we know $F = f$, $\operatorname{argmin}_{\mathbf{x} \in \mathbb{R}^d} F(\mathbf{x}) = \mathbf{y}$ and $F_\star = 0$. Moreover, we have $\mu = 0$ and $G = \sqrt{d}$.

**Noise.** We choose $\xi_i \sim \epsilon Z$ i.i.d. for all $i \in [d]$, where $\epsilon$ and $Z$ are independent and satisfy that $\Pr[\epsilon = 2] = \frac{1}{3}$ and $\Pr[\epsilon = -1] = \frac{2}{3}$, and $Z$ follows the Pareto distribution with the scale parameter $\frac{\alpha - 1}{\alpha}$ and the shape parameter $\alpha = \mathfrak{p} + 0.001$, i.e., $\Pr[Z > z] = \left( \frac{\alpha - 1}{\alpha z} \right)^{\alpha} \mathbb{1}\left[ z \geq \frac{\alpha - 1}{\alpha} \right] + \mathbb{1}\left[ z < \frac{\alpha - 1}{\alpha} \right]$. Note that we have $\mathbb{E}[\epsilon Z] = 0$, $\mathbb{E}\left[ |\epsilon|^{\mathfrak{p}} \right] = \frac{2^{\mathfrak{p}} + 2}{3}$ and $\mathbb{E}[Z^{\mathfrak{p}}] = \frac{\alpha}{\alpha - \mathfrak{p}} \left( \frac{\alpha - 1}{\alpha} \right)^{\mathfrak{p}}$, implying that $\mathbb{E}[\xi_i] = 0$ and $\sigma = \left( \mathbb{E}\left[ |\xi_1|^{\mathfrak{p}} \right] \right)^{\frac{1}{\mathfrak{p}}} = \left( \frac{2^{\mathfrak{p}} + 2}{3} \right)^{\frac{1}{\mathfrak{p}}} \left( \frac{\alpha}{\alpha - \mathfrak{p}} \right)^{\frac{1}{\mathfrak{p}}} \frac{\alpha - 1}{\alpha}$.

**Algorithms.** We consider Liu & Zhou (2023) as the baseline, since it is closest to our setting, and choose the stepsize $\eta_t$ and the clipping threshold $\tau_t$ as follows:

- Adopted from Theorem 4 in Liu & Zhou (2023): $\eta_t = \frac{\eta}{\sigma_{\mathfrak{l}} t^{1/\mathfrak{p}}}$ and $\tau_t = \max\left\{2\sqrt{d}, \sigma_{\mathfrak{l}} t^{1/\mathfrak{p}}\right\}$, where $\eta = \|\mathbf{x}_1 - \mathbf{y}\|$ and $\mathbf{x}_1$ is the initial point.

- Adopted from our Theorem 3: $\eta_t = \frac{\eta}{\sigma_{\mathfrak{s}}^{2/\mathfrak{p}-1} \sigma_{\mathfrak{l}}^{2-2/\mathfrak{p}} t^{1/\mathfrak{p}}}$ and $\tau_t = \max\left\{2\sqrt{d}, \frac{\sigma_{\mathfrak{l}}}{d_{\text{eff}}^{1/\mathfrak{p}}} t^{1/\mathfrak{p}}\right\}$, where $\eta = \|\mathbf{x}_1 - \mathbf{y}\|$ and $\mathbf{x}_1$ is the initial point.

*Remark* 12. For both $\eta_t$, we only keep the dominant term in the order of $\mathcal{O}(1/t^{1/\mathfrak{p}})$ for simplicity. We pick $\eta = \|\mathbf{x}_1 - \mathbf{y}\|$ to match the optimal choice in theory. Moreover, $\eta_t$ is set in an anytime fashion, i.e., depending on $t$ instead of $T$. $\sigma_{\mathfrak{l}} = \sqrt{d}\sigma$ and $\sigma_{\mathfrak{s}} = 2^{\frac{2}{\mathfrak{p}}-1} d^{\frac{1}{\mathfrak{p}}-\frac{1}{2}}\sigma$ are set based on their bounds given in (8) and (9), respectively. $d_{\text{eff}}$ is set as its lower bound $\frac{d^{2-\frac{2}{\mathfrak{p}}}}{2^{\frac{4}{\mathfrak{p}}-2}}$ established in (11).

**Parameter values.** In experiments, we fix $d = 50$, set $\mathbf{y}_i = \begin{cases} 2i/d & i \leq d/2 \\ -2i/d & i > d/2 \end{cases}$, initialize $\mathbf{x}_1 = \mathbf{0}$, and let $T = 10000$. For two kinds of $(\eta_t, \tau_t)$, we run 10 trials for each and plot the mean ($\pm$ standard error) of the trajectory $F(\bar{\mathbf{x}}_{t+1}^{\text{cvx}}) - F_\star = F(\bar{\mathbf{x}}_{t+1}^{\text{cvx}})$, as used in the convergence theory, where we recall $\bar{\mathbf{x}}_{t+1}^{\text{cvx}} = \frac{1}{t}\sum_{s=1}^{t} \mathbf{x}_{s+1}$. We test $\mathfrak{p} \in \{1.2, 1.4, 1.6, 1.8\}$ and report the results in Figure 1.

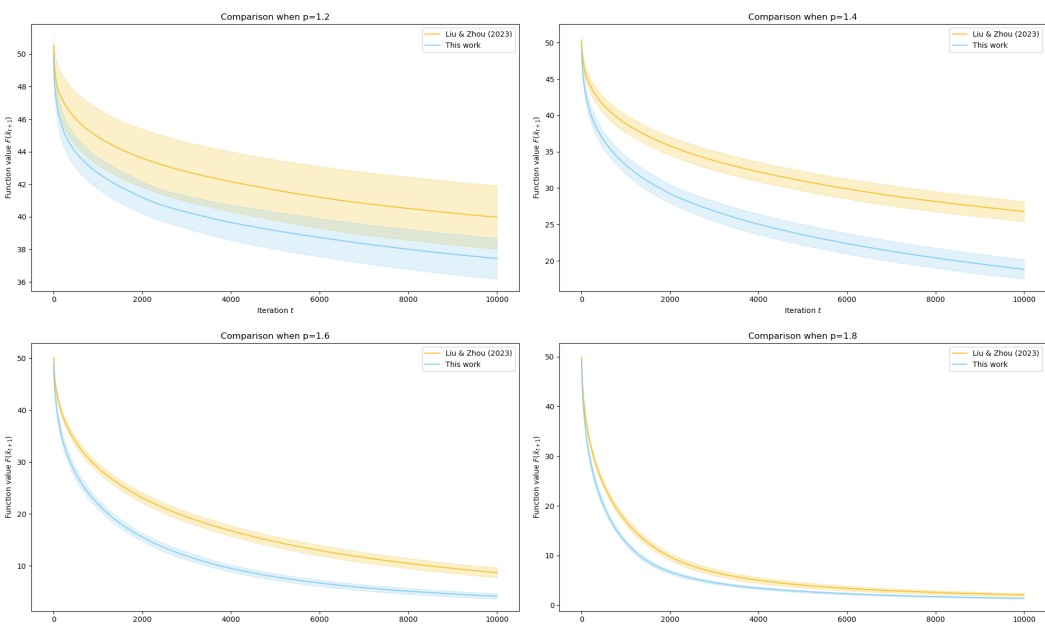

Figure 1: Comparison between Liu & Zhou (2023) and this work when $\mathfrak{p} = 1.2$ (top left), $\mathfrak{p} = 1.4$ (top right), $\mathfrak{p} = 1.6$ (bottom left), $\mathfrak{p} = 1.8$ (bottom right).

**Observation and Conclusion.** In all cases, the $(\eta_t, \tau_t)$ pair chosen based on our work is faster, matching the new theoretical finding when $\sigma_{\mathfrak{s}} \neq \sigma_{\mathfrak{l}}$. As $\mathfrak{p}$ approaches 2, the difference becomes minor, which should be expected, since the improvement predicted by our theory is in the order of $\Theta(1/d_{\text{eff}}^{\frac{2-\mathfrak{p}}{2\mathfrak{p}}})$ (see discussion under Theorem 3), which will vanish if $\mathfrak{p}$ is close to 2.

