# OpenReview forum: "Clipped Gradient Methods for Nonsmooth Convex Optimization under Heavy-Tailed Noise: A Refined Analysis"
_ICLR.cc/2026/Conference — ICLR 2026 Poster_

### Official Review · Reviewer_9vd3 · 2025-10-28

**Soundness:** 3
**Presentation:** 3
**Contribution:** 2
**Rating:** 4
**Confidence:** 4

**Summary:**

The paper presents a refined analysis of the Clipped SGD method under both convex and strongly convex regimes. Through a more careful application of Freedman’s inequality and an enhanced clipping lemma providing tighter bounds on bias and variance, the authors derive improved convergence rates compared to existing results.

**Strengths:**

1. **Extensive theoretical results.** The authors provide a fairly comprehensive analysis of various cases. They also emphasize that the discrepancy between the obtained upper bounds and the existing lower bounds is natural, since the classes of problems do not coincide in terms of the stochastic oracle model.

2. **Improvement of the result in terms of $\ln(T)$.** The authors applied the concentration inequality in a tighter form (to me more precise, Freedman's inequality), which allowed them to eliminate the arising factor $\ln(T)$ compared to Das et al. (2024).

3. **Great clipping lemma.** The authors propose a refined clipping lemma that yields improved bounds for bias and variance. Importantly, this lemma (see Theorem 5) does not rely on fact that the norm of the true gradient should be less than $\frac{\tau}{2}$ (this condition is required in many papers, e.g. [1], [2]), which marks a significant methodological advantage.

---

**References**

[1] High-Probability Bounds for Stochastic Optimization and Variational Inequalities: the Case of Unbounded Variance. Sadiev et al. (2023)

[2] High Probability Convergence of Clipped-SGD Under Heavy-tailed Noise. Nguyen et al. (2023)

**Weaknesses:**

### **Minor weaknesses**

1. **Failure of SGD.** To be fair, the inability of SGD to cope with heavy-tailed noise deserves a clearer explanation. While it is true that, in the absence of a finite second moment for the stochastic gradient error norm, the classical analysis paradigm breaks down, there exists [3] showing that SGD still converges in expectation under condition $p \in (1, 2]$. Nevertheless, it is provable that SGD, and even Adam, fail to adequately cope with this type of stochasticity when high-probability convergence is considered. These results were established in [4] and [5] for SGD and Adam, respectively. Accordingly, work [4] should be cited to substantiate this point.

2. **Clipped SGD or Proximal Clipped SGD.** It would be more appropriate to refer to the method considered by the authors as Proximal Clipped SGD, since the notation introduced in the paper may confuse readers, and this terminology is formally more accurate.

### **Major weaknesses**

1. **Assumptions.** A key limitation concerns the assumptions adopted. In particular, the set $\mathbb{X} \subseteq \mathbb{R}^d$ on which the optimization problem is defined is assumed to be nonempty, closed, and convex. Therefore, $\mathbb{X} = \mathbb{R}^d$ can be considered. At the same time, in the paper, the function $f$ is assumed to be both convex and Lipschitz with constant $G$. Regrettably, this assumption severely restricts the class of admissible functions, as it effectively admits **very limited** examples of $f$ -- linear functions or $|| \cdot ||$ (or other examples of convex functions with unformly bounded subgradient). This automatically leads to the need to impose additional constraints on the set $\mathbb{X}$.

2. **Fighting for $\ln(T)$ elimination.** In fact, one can consider an additional assumption of **boundedness** of the set, that is, taking its closedness into account, the optimization can be considered over a compact set. From the perspective of the class of functions considered, this class is quite broad. However, in this case, striving to eliminate $\ln(T)$ becomes entirely meaningless. In existing analyses, such as in work [6], the factor $\ln\left(\frac{T}{\delta}\right)$ arises from the problem formulation and the assumptions made. The analysis in this work, as well as in other similar studies, is constructed so that the optimization problem is solved over the entire $\mathbb{R}^d$, while the assumptions are made on a compact subset $\mathbb{X}$. Consequently, during the convergence analysis, it is necessary to use induction to ensure that points conducted by optimization method do not leave the domain where assumptions hold. Based on this type of analysis, the probability estimate deteriorates by $\frac{\delta}{T}$ at each step (which is precisely why  $\ln\left(\frac{T}{\delta}\right)$ appears). However, if the problem is formulated on a compact set, no inductive steps are required, and thus the bound with factor $\ln\left(\frac{1}{\delta}\right)$ instead of $\ln\left(\frac{T}{\delta}\right)$ can be achieved.

Emphasizing the above, the main issue lies in the assumptions made, since in the current formulation, without a bounded set, the class of considered functions $f$ is very limited. If boundedness of the set is added, $\ln(T)$ can be eliminated based on existing analyses. Therefore, in my opinion, the main contribution of this work at present is a **more precise treatment of the heavy-tailed stochasticity via Assumption 4**.

Despite the concerns mentioned, I am open to reassess the paper according to further explanations or revisions by authors.

---

**References**

[3] Can SGD Handle Heavy-Tailed Noise? Fatkhullin et al. (2025)

[4]  High-Probability Bounds for Stochastic Optimization and Variational Inequalities: the Case of Unbounded Variance. Sadiev et al. (2023)

[5] Clipping Improves Adam-Norm and AdaGrad-Norm when the Noise Is Heavy-Tailed. Chezhegov et al. (2024)

[6] High Probability Complexity Bounds for Non-Smooth Stochastic Optimization with Heavy-Tailed Noise. Gorbunov et al. (2024)

**Questions:**

1. Can convergence in the strongly convex setting be proved for the last iterate (for the term $f(x) - f^*$)? What are the technical challenges involved in performing such an analysis (if I am not mistaken, the current results are derived for a specially weighted point)?

---

> ### Author Response · Authors · 2025-11-21
>
> We thank the reviewer for the valuable comments and address the reviewer's concerns below.
>
> **Minor weaknesses.** Thanks for the suggestion. We would like to kindly remind the reviewer that both points were discussed in Section 3.
>
> 1. For the first point, see Lines 200-202, where we stated that the clipping operation is the key step for achieving high-probability convergence, by citing Sadiev et al. (2023) as evidence.
>
> 2. For the second point, see Lines 193-198, where we clarified that the algorithm should be called "Proximal Clipped SGD", but we dropped the word “Proximal” for simplicity.
>
> We are happy to make further clarification on these two points.
>
> **Major weaknesses.** We first thank and agree with the reviewer that introducing Assumption 4 is an important contribution. Next, we discuss the other points the reviewer mentioned.
>
> 1. For the first point, we believe the generality of $\mathbb{X}$ doesn't hurt our work.
>
>    - From a theoretical view, we remark that the property of $\mathbb{X}$ and the Lipschitz condition on $f$ should be treated in a decoupled way. We assume $f$ is Lipschitz because this work focused on nonsmooth optimization, and it is the standard condition used in the literature (regardless of convexity). However, as mentioned in Section 7 (Lines 481 to 484), the main concept of this paper (i.e., Assumption 4 plus refined analysis) is not limited to the Lipschitz case; one can extend our central idea to other settings (e.g., the smooth case without bounded gradients) and still obtain an improved rate for Clipped SGD. Therefore, from a high-level view, the boundness of $\mathbb{X}$ (or not) and the Lipschitzness of the gradient (or not) don't really affect each other, since the latter can be replaced if one prefers to, and the former, of course, includes the compact set as a special case due to our generality of $\mathbb{X}$. As such, we believe the generality of $\mathbb{X}$ should be more favorable.
>
>    - From a practical view, many problems in reality are indeed nearly linear and hence satisfy the Lipschitz condition, without imposing compactness on $\mathbb{X}$, e.g., SVM with hinge/huber loss, LAD regression, logistic regression, stochastic bandit. In addition, we highlight that quadratic optimization can also be embedded into $F$ due to the existence of $r$. For example, the mean estimation problem from a heavy-tailed distribution, i.e., $\\\|x-\mathbb{E}[\xi]\\\|^2=f(x)(\triangleq-2\langle x,\mathbb{E}[\xi]\rangle+\\\|\mathbb{E}[\xi]\\\|^2)+r(x)(\triangleq\\\|x\\\|^2)=F(x)$ with stochastic gradient $g(x,\xi)=-2\xi$.
>
>    From both views, the potential unboundness of $\mathbb{X}$ doesn't fundamentally harm our paper.
>
> 1. For the second point, we clarify that the eliminated $\ln(T)$ term in our paper is subtly different from the $\ln(T)$ term mentioned by the reviewer. There are two kinds of elimination in our paper: (a) removing the $\ln(T)$ term in $\ln(\ln(T)/\delta)$ caused by the concentration inequality from [1] when specified to $p=2$ (see lines 236 to 248). (b) removing the $\ln(T)$ term for anytime convergence in the general convex case (see Lines 67 to 69), i.e., Theorems 7 and 9. Certainly, if assuming a compact domain as the reviewer's comment, these two issues could be resolved in an alternative way. However, we prefer to provide a unified solution under a general $\mathbb{X}$ that doesn't rely on any additional assumptions.

---

> ### Author Response · Authors · 2025-11-21
>
> **Questions.** The current convergence for the functional gap is derived for a specially weighted point, as the reviewer said. This is common in the literature of nonsmooth strongly convex optimization, even for the deterministic case (e.g., see [2]).
>
> As for last-iterate convergence, we believe it is possible, but it may require more careful, complex calculations. A potential way could be to employ the average-to-last iterate method introduced in the seminal work [3]. However, proving last-iterate convergence is known to be harder than (weighted) average-iterate convergence. In particular, under heavy-tailed noise and for the clipping-based method, we are only aware of one work [4] that provides a last-iterate guarantee on the functional gap for the general convex case, using the average-to-last iterate method from [4] we mentioned before. Therefore, we think this topic is far beyond the current scope of the paper and would rather leave it as a future direction.
>
> **References**
>
> [1] Das, A., Nagaraj, D., Pal, S., Suggala, A., & Varshney, P. (2024). Near-optimal streaming heavy-tailed statistical estimation with clipped SGD. Advances in Neural Information Processing Systems, 37, 8834-8900.
>
> [2] Lacoste-Julien, S., Schmidt, M., & Bach, F. (2012). A simpler approach to obtaining an O (1/t) convergence rate for the projected stochastic subgradient method. arXiv preprint arXiv:1212.2002.
>
> [3] Shamir, O., & Zhang, T. (2013, February). Stochastic gradient descent for non-smooth optimization: Convergence results and optimal averaging schemes. In International conference on machine learning (pp. 71-79). PMLR.
>
> [4] Parletta, D. A., Paudice, A., & Salzo, S. (2024). An Improved Analysis of the Clipped Stochastic subGradient Method under Heavy-Tailed Noise. arXiv preprint arXiv:2410.00573.

---

> ### Comment · Reviewer_9vd3 · 2025-11-22
> **Response to Authors**
>
> Dear Authors,
>
> Thank you for the detailed feedback regarding my concerns.
>
> ### **Minor weaknesses:**
>
> 1. This remark did not arise because you did not mention work [1] as the one showing that clipping is a key step for obtaining high-probability convergence guarantees; it arose because of the statement in Line 43: SGD still converges in expectation [2] in case $p <2$. However, it is precisely work [1] that shows that SGD has poor high-probability convergence, i.e., required number of iterations $T$ to reach $\varepsilon$-accuracy with failure probability $\delta$ can be expressed as $T = \Omega\Big(\frac{1}{\sqrt{\varepsilon \delta}}\Big)$ in the worst case. Therefore, it is not entirely correct to cite work [3] with the statement that “SGD has been proved to fail in convergence” (see Line 43) — that work shows that if $p < 2$, then $\mathbb{E}[\|\|\nabla f(x_k)\|\|^2] = +\infty$, but this *is not a guarantee that convergence of SGD cannot be proved*, since work [2] exists. Thus, I would recommend clarifying what exactly is meant in that line — it seems that the current phrasing is not entirely accurate.
>
> 2. This is fine, but the name "Clip-SGD" implies clipping the stochastic gradient of the entire objective function (in your notation, it is $F$). In your work, the stochastic gradient is taken specifically for $f$, i.e., for a part of the composite problem. Nevertheless, this comment is merely a suggestion to improve the readability of the paper.
>
> Regarding **Major weaknesses** concerns, I would like to thank you for almost fully addressing my concerns. Based on the authors’ response, I can highlight the main key advantages of this work:
>
> 1) Improved treatment of heavy-tailed stochasticity: Assumption 4 and the strengthened clipping lemma.
> 2) Elimination of the additional factor $\ln(T)$ in case $\mathbb{X} = \mathbb{R}^d$ (as confirmed by the authors in their response) compared to Das et al.
>
> Moreover, I am grateful for the clarifications regarding the questions that arose. Based on the above, I have raised my score.
>
> ---
>
> **References**
>
> [1] High-Probability Bounds for Stochastic Optimization and Variational Inequalities: the Case of Unbounded Variance. Sadiev et al. (2023)
>
> [2] Can SGD Handle Heavy-Tailed Noise? Fatkhullin et al. (2025)
>
> [3] Why are Adaptive Methods Good for Attention Models? Zhang et al. (2020)

---

> > ### Author Response · Authors · 2025-11-25
> >
> > We greatly appreciate the reviewer's further feedback. We have slightly modified the corresponding sentence to eliminate the ambiguity. We will continue polishing the paper to improve its quality.

---

### Official Review · Reviewer_WNtT · 2025-10-29

**Soundness:** 4
**Presentation:** 3
**Contribution:** 3
**Rating:** 6
**Confidence:** 3

**Summary:**

The paper studies the clipped stochastic gradient descent under tighter assumptions on heavy-tailed noise (Assumption 4 in the manuscript). They analyze the proximal version, assuming that the objective function is convex and $G$-Lipschitz. By better utilizing the Freedman's inequality, they obtain better convergence rates for high-probability convergence, replacing the $\log\frac{\log T}{\delta}$ term with $\log \frac{1}{\delta}$. For in-expectation convergence, utilizing Assumption 4 results in an improvement by a factor of $\Theta\left(d_{\text{eff}}^\frac{p-2}{2p}\right)$.

**Strengths:**

1)Improving convergence rates in both high-probability and in-expectation analyses.

2)Considering tighter assumption on heavy-tailed noise and introducing a generalized effective dimension $d_{\text{eff}}$.

3)Conducting a tighter analysis using martingale inequalities.

**Weaknesses:**

1)Bounded gradients are quite a restrictive assumption for Clipped SGD.

2)Following the previous point, the lower bound on iteration depends on $G$. Therefore, with exploding gradients we have to perform significantly more iterations.

3)The paper lacks a performance comparison of Clipped SGD with $\sigma_s\neq\sigma_l$, as well as their evaluation on the different datasets.

**Questions:**

1)What were the main difficulties in adapting the previous analysis in [1]?

2)How can the analysis be extended to more realistic assumptions -- $L$-smooth or $(L_0,L_1)$-smooth?

3)Can the existing analysis be extended for nonconvex case?

[1] Das, A., Nagaraj, D., Pal, S., Suggala, A., & Varshney, P. (2024). Near-optimal streaming heavy-tailed statistical estimation with clipped SGD. Advances in Neural Information Processing Systems, 37, 8834-8900.

**Details Of Ethics Concerns:**

No additional ethical concerns.

---

> ### Author Response · Authors · 2025-11-21
>
> We thank the reviewer for the valuable comments and address the reviewer's concerns below.
>
> **W1.** We would like to clarify two points here:
>
> 1. For nonsmooth optimization (regardless of convexity), the bounded gradients assumption is commonly adopted in the literature [1,2] (even for deterministic optimization). Therefore, this is a normal assumption.
>
> 1. More importantly, as discussed in Section 7 (Lines 481 to 484), the main concept of this paper (i.e., Assumption 4 plus refined analysis) is not limited to the Lipschitz case; one can extend our central idea to other settings (e.g., the smooth case without bounded gradients) and still obtain an improved rate for Clipped SGD.
>
> In summary, the bounded gradients assumption is not essential to us. We chose it following the standard literature.
>
> **W2.** A dependence on $G$ is standard in nonsmooth optimization theory and is known to be inevitable even in deterministic optimization [1]. In analogy, for smooth optimization with a smooth parameter $L$, the convergence rate will also depend on $L$ [2]. Therefore, we recognize this as normal.
>
> **W3.** Thanks for the suggestion. We have added numerical simulations in Appendix G, which successfully justify the improvement predicted by our theory when $\sigma_\mathfrak{l}\neq \sigma_\frak{s}$.
>
> **Q1.** There are two points we cannot directly adopt from [3].
>
> 1. The first is the clipping error analysis. Since [3] studied the case $p=2$, the clipping error analysis given in [3] is specified for $p=2$ as well. For example, their error bound depends on $\mathbb{E}[\\\|g(x,\xi)-\nabla f(x)\\\|^2]$, which can be infinite and hence vacuous under our Assumption 4. Therefore, we need to analyze the clipping error under our Assumption 4 from scratch.
>
> 1. Strictly speaking, equipped with our new clipping error bounds, one could employ the concentration inequality introduced by [3] (their Theorem 5) to give a high-probability analysis. However, this will result in a weaker rate, as supported by the comparison between [3] and our Theorem 1 (for $p=2$) (see Lines 236 to 248). Moreover, we believe our better utilization of Freedman's inequality is simpler and hence more reader-friendly than [3].
>
> **Q2&Q3.** The analysis can be extended to other settings (e.g., the (generalized) smooth, nonconvex case), under our newly proposed Assumption 4. Simply speaking, with Assumption 4, the core ideas can be summarized to better utilize Freedman's inequality and apply refined clipping error bounds in the analysis. For the first point, it doesn't relate to the problem itself. For the second point, our new clipping bounds also don't rely on the objective (see the general version, Theorem 5).
>
> Therefore, for the case asked by the reviewer, one can first start from any of the existing analyses, and then modify the proof based on our ideas whenever the following two situations are met: (a). The concentration step (b). A clipping error bound is needed. Consequently, a better upper bound can be shown.
>
> **References**
>
> [1] Bubeck, S. (2015). Convex optimization: Algorithms and complexity. Foundations and Trends® in Machine Learning, 8(3-4), 231-357.
>
> [2] Nesterov, Y. (2018). Lectures on convex optimization (Vol. 137). Berlin: Springer International Publishing.
>
> [3] Das, A., Nagaraj, D., Pal, S., Suggala, A., & Varshney, P. (2024). Near-optimal streaming heavy-tailed statistical estimation with clipped SGD. Advances in Neural Information Processing Systems, 37, 8834-8900.

---

### Official Review · Reviewer_BWN2 · 2025-10-31

**Soundness:** 3
**Presentation:** 3
**Contribution:** 2
**Rating:** 6
**Confidence:** 3

**Summary:**

The authors consider the problem of heavy-tailed composite optimization with a convex or strongly convex regularizer. They provide refined high-probability rates of convergence for proximal clipped stochastic gradient descent in both convex and strongly convex scenarios. The improvements are based on two main ingredients. The first one is a tighter bound on the variance term in Freedman's inequality. The second one is a better bound on clipping error following from Lemma 1 (in particular, from the refined bounds on $\|\| \mathbb E_{t - 1} [d_t^u (d_t^u)^\top]\|\|$ and $\|\|d_t^b\|\|$). The authors also prove in-expectation bounds (Theorems 3 and 4) with essentially the same rates of convergence.

**Strengths:**

The rates of convergence in Theorems 1 and 2 are tighter compared to [Das et al., NeurIPS 2024], [Gorbunov et al., JOTA 2024], [Liu, Zhou, arXiv:2303.12277, 2023]. The present paper also removes some artefacts. For instance, in [Das et al., NeurIPS 2024] the upper bound blows up when $\sigma_s$ tends to zero.

**Weaknesses:**

1. The upper bound on $\mathbb E_{t - 1} X_t^2 = \mathbb E_{t - 1} \langle d_t^u, y_t\rangle^2$ is quite standard. The fact that $\mathbb E_{t - 1} X_t^2 \leq \|\| \mathbb E_{t - 1} [ d_t^u (d_t^u)^\top ] \|\|$ has been used in statistics. I am a bit surprised that it was overlooked in optimization.

2. According to Section 5, the main novelty of the paper is based on the refined bounds on $\|\| \mathbb E_{t - 1} [d_t^u (d_t^u)^\top]\|\|$ and $\|\|d_t^b\|\|$ stated in Lemma 1. Their proofs take a few lines each. For this reason, the contribution looks somewhat incremental.

**Questions:**

Is it possible to choose the clipping threshold independently of $\delta$? If you could do so, then the results of Theorem 3 and Theorem 4 would easily follow from the high-probability upper bounds.

---

> ### Author Response · Authors · 2025-11-21
>
> We thank the reviewer for the valuable comments and address the reviewer's concerns below.
>
> **W1.** We are also surprised that this fact was overlooked in optimization. We hope our work can offer fresh perspectives to the optimization community.
>
> **W2.** We would like to clarify that, among the total 6 provided bounds for clipping error (see Theorem 5/Lemma 2), except for the first $\\\|d_t^u\\\|\leq 2\tau_t$, the remaining 5 results are either new or improve upon the existing literature. In addition, we highlight that the appearance of these 5 bounds is highly affected by the new Assumption 4. Therefore, from a modeling perspective, we recognize introducing Assumption 4 to the optimization community as another important contribution. In the following, we elaborate more on this point.
>
> 1. For the two bounds mentioned by the reviewer (corresponding to bounds 4 and 6 in Theorem 5/Lemma 2). The bound on $\\\|\mathbb{E}_{t-1}[d_t^u(d_t^u)^\top]\\\|$ is new, as far as we know. The bound on $\\\|d_t^b\\\|$ is a refined result over the existing literature. However, we emphasize that refinement is possible only after introducing Assumption 4. Therefore, Assumption 4 is another key in this work.
>
> 1. For bounds 2, 3, and 6. Bound 2 is a refined result, and both bounds 3 and 6 are new. More crucially, it is noteworthy that these three bounds drop the usual condition $\tau_t\geq 2\\\|\nabla f(x_t)\\\|$ used in prior works. This fact seems simple, but it is indeed deep, as this is the first time a clipping error analysis is presented without requiring this restrictive assumption. This point is also supported by the third strength in another Reviewer 9vd3's comment. Consequently, within these three bounds, it is possible to prove the convergence rate of Clipped SGD without knowledge of problem-dependent parameters, which is arguably the more realistic case, since even estimating them may be hard in practice. However, we didn't discuss this point much yet (only a few words in Remark 7) to keep the work focused on one topic. But we are happy to explicitly mention this point in the revision if the reviewer thinks this could clarify our contributions and strengthen our work.
>
> **Questions.** Thanks for this insightful question. In short, the reviewer's suggestion is possible (see 1. below). However, our derivation of in-expectation rates remains important, since the reviewer's mentioned approach could yield a weaker in-expectation rate (see 2. below).
>
> 1. We can obtain a high-probability bound without employing $\delta$ in the clipping threshold and stepsize. Using the convex case as an example, one can first combine Eqs. (34) and (35), then pick the clipping threshold $\tau_t$ and stepsize $\eta_t$ independent from $\delta$ (a simple choice is to replace every $\ln(1/\delta)$ in $\eta_t$ and $\tau_t$ with $1$). Under this change, we still have a high-probability bound, but with a worse dependence on $\ln(1/\delta)$ as a tradeoff.
>
> 1. Even if the above approach is doable, the in-expectation rate obtained by integrating on the tail bound could be weaker. Still consider the convex case as an example and, for simplicity, use the constant stepsize $\eta_t=\eta$ and the constant clipping threshold $\tau_t=\tau$ (for convenience, assume $\tau\geq 2G$). Moreover, let $Z\triangleq F(x_{\rm{output}})-F_\star$ and $[x]_+=\max\\\{0,x\\\}$.
>
>    - For our current in-expectation bound, one can find that $\mathbb{E}[Z]=O(M)$ holds for some $M$ that depends on $\eta$, $\tau$, and other parameters (see the proof of Theorem 8).
>
>    - By the approach mentioned in 1., we have $\mathbb{P}[Z\geq M+N\ln^2(3/\delta)]\leq\delta$ after combining Eqs. (34) and (36), where $M$ is the same as the one in the above in-expectation bound and $N\triangleq\Theta(\eta\tau^2/T)$. Next, let us convert it into an in-expectation rate. Since for $t>0$, $\mathbb{P}[[Z-M]\_+/N\geq t]= \mathbb{P}[Z\geq M+Nt]\leq 3\exp(-\sqrt{t})$, we thus have $\mathbb{E}[[Z-M]\_+/N]=\int_{t\geq 0}\mathbb{P}[[Z-M]\_+/N\geq t]\text{d}t\leq \int_{t\geq 0}3\exp(-\sqrt{t})\text{d}t=O(1)$. Therefore, we conclude $\mathbb{E}[Z]\leq \mathbb{E}[M+[Z-M]_+]=O(M+N)$.
>
>    As one can see, the newly obtained bound $O(M+N)$ is weaker than our current bound $O(M)$, leading to a worse rate in the end.

---

> > ### Comment · Reviewer_BWN2 · 2025-11-27
> >
> > The authors have adequately addressed my remarks from the section "Weaknesses". I have no further questions.
> >
> > Thank you for giving an insight why the $\delta$-dependent choice of threshold leads to worse bounds. It seems that the question of tighter large deviation bounds for $Z$ goes beyond the scope of the present paper.

---

### Official Review · Reviewer_j1s2 · 2025-11-11

**Soundness:** 3
**Presentation:** 4
**Contribution:** 3
**Rating:** 4
**Confidence:** 3

**Summary:**

This paper studies (strongly) convex non-smooth stochastic optimization under heavy-tailed gradient noise. Clipped-SGD is analyzed, and two refined rates are derived in both high-probability and in-expectation sense.

**Strengths:**

1. This paper provides a careful analysis that better utilizes Freedman’s inequality that leads to refined high-probability and in-expectation bounds.
2. This work generalizes the effective dimension into heavy-tailed robust optimization.

**Weaknesses:**

1. The boundedness of gradients is a strong assumption given that the motivation of heavy-tailed noise is typically training neural networks.
2. The improvements of the bound is mostly in terms of constant, and it is not clear whether this is a significant improvement.
3. No experiments provided. Some simple numerical examples to validate theory can also be helpful.

**Questions:**

1. The improvement depends on the magnitude of $d_{eff}$. It would help to discuss how the noise structure change $d_{eff}$.
2. The breaking of lower bounds seem to be dependent on the restrictions on noise family. If that's the case, can you discuss more on this point why lower bounds can be improved?
3. Can this refined analysis be applied on other nonlinearities such as normalization and sign?

---

> ### Author Response · Authors · 2025-11-21
>
> We thank the reviewer for the valuable comments and address the reviewer's concerns below.
>
> **W1.** We clarify that the bounded gradients assumption is not essential to us. We chose it following the literature.
>
> 1. For nonsmooth optimization considered in our work (regardless of convexity), the bounded gradients assumption is commonly adopted in the literature (e.g., [1,2]), even for deterministic optimization.
>
> 1. We agree and understand that many people may know heavy-tailed noise from neural network training. However, in fact, heavy-tailed distributions/noise/data are common in different fields (e.g., stocks; see [3] for numerous examples), not limited to neural networks. Under heavy-tailed data, many convex optimization problems satisfy our assumptions (e.g., LAD regression).
>
> 1. Moreover, as discussed in Section 7 (Lines 481 to 484), the main concept of this paper (i.e., Assumption 4 plus refined analysis) is not limited to the Lipschitz case; one can extend our central idea to other settings (e.g., the smooth case without bounded gradients) and still obtain an improved rate for Clipped SGD.
>
> **W2&Q1.** As the reviewer also mentioned, the improvement is in the order of $\text{poly}(1/d_\text{eff})$, which should, however, not be viewed as a constant, since $d_\rm{eff}$ can be in the same order as the true dimension $d$. A concrete example was provided in Appendix A.
>
> We briefly discuss how the noise structure changes $d_\rm{eff}$ below. For simplicity, we consider $g(x,\xi)=\nabla f(x)+\xi$, i.e., the additive noise model. The case $d_\rm{eff}=1$ is certainly achievable (e.g., all but the first coordinates of $\xi$ are $0$ almost surely). So, we only need to understand when $d_\rm{eff}$ can be as large as $\Omega(d)$. (For the case between $\Omega(d)$ and $1$, one can make a reduction by setting some coordinates of $\xi$ to $0$.)
>
> 1. For $p=2$, this can be analytically computed. Let $\Sigma\triangleq\mathbb{E}[\xi\xi^\top]$ be the covariance matrix of the noise, then $\sigma^2_\mathfrak{l}=\mathrm{Tr}(\Sigma)=\sum_{i\leq d}\lambda_i$ and $\sigma^2_\frak{s}=\\\|\Sigma\\\|\_\rm{op}=\lambda_1$, for $\lambda_1\geq\dots\geq\lambda_d\geq0$ being the eigenvalues of $\Sigma$. In this case $d_\mathrm{eff}=\sum_{i\leq d}\lambda_i/\lambda_1$. This suggests that if $\lambda_i$ are all in the same order, then $d_\rm{eff}$ can be as large as $\Omega(d)$ (this can be easily achieved when $\xi$ has i.i.d. coordinates, implying $d_\mathrm{eff}=d$).
>
> 1. For general $p\in(1,2)$, it is not easy to understand $d_\rm{eff}$ immediately. But from our example in Appendix A, we know an $\Omega(d)$ bound is still achievable.
>
> **W3.** Thanks for the suggestion. We have added numerical simulations in Appendix G, which successfully justify the improvement predicted by our theory.
>
> **Q2.** Simply speaking, breaking the lower bound is achieved by restricting the problem instance to a smaller, more realistic set that excludes some pathological cases not expected in high-dimensional optimization problems. We elaborate more on this point below.
>
> 1. The existing lower bound is proved for the family of unbiased Stochastic First-Order (SFO) oracles $g(x,\xi)$ satisfying $\mathbb{E}[\\\|d(x,\xi)\\\|^p]= \sigma^p_\frak{l}$, without any restriction on $\sup_{e\in\mathbb{S}^{d-1}}\mathbb{E}[|\langle e,d(x,\xi)\rangle|^p]$, where $d(x,\xi)\triangleq g(x,\xi)-\nabla f(x)$. Consequently, some pathological SFO oracles are included, e.g., $d(x,\xi)=0$ for all but the first coordinate, which essentially reduces the problem to 1-dimensional stochastic optimization. But such an SFO oracle doesn't always happen in practice (except the problem itself is really on $\mathbb{R}$). This means the existing lower bound may be too pessimistic.
>
> 1. To exclude the above pathological SFO oracle for high-dimensional problems, let us take $p=2$ as an example and to simplify the discussion, assume $d(x,\xi)=\xi$ where each coordinate $\xi_i$ is independent (not necessarily identically distributed), a simple strategy is to limit the contribution of $\xi_i$ to the quantity $\mathbb{E}[\\\|d(x,\xi)\\\|^2]=\sum_{i\leq d}\mathbb{E}[\xi_i^2]$. In other words, we can impose another condition $\max_{i\in[d]}\mathbb{E}[\xi_i^2]=\sup_{e\in\mathbb{S}^{d-1}}\mathbb{E}[(\langle e,d(x,\xi)\rangle)^2]\leq \sigma^2_\frak{s}$. Once $\sigma_\frak{s}< \sigma_l$, we eliminate the possible pathological SFO oracles and could possibly break the existing pessimistic lower bound. The high-level idea also applies to general $p\in(1,2)$.

---

> > ### Author Response · Authors · 2025-11-21
> >
> > **Q3.** This is an interesting question.
> >
> > 1. The currently presented clipping error analysis (i.e., Lemmas 1/2 and Theorem 5) is specified for clipped methods. We don't think it generally fits other nonlinear methods.
> >
> > 1. However, the principle of better utilizing Freedman's inequality in general holds, and one can always consider Assumption 4. Therefore, at a high level, the central ideas presented in our work are possibly extended to other nonlinearities, but this may be done by a different analysis style, which is out of scope for our paper, and we leave it for future exploration.
> >
> > **References**
> >
> > [1] Bubeck, S. (2015). Convex optimization: Algorithms and complexity. Foundations and Trends® in Machine Learning, 8(3-4), 231-357.
> >
> > [2] Nesterov, Y. (2018). Lectures on convex optimization (Vol. 137). Berlin: Springer International Publishing.
> >
> > [3] Nair, J., Wierman, A., & Zwart, B. (2022). The fundamentals of heavy tails: Properties, emergence, and estimation (Vol. 53). Cambridge University Press.

---

### Author Response · Authors · 2025-11-21
**Revision History**

We thank all the reviewers for their valuable comments. We summarize the revision history below:

- **Nov 20.** Some typos have been fixed. Numerical simulations are added to Appendix G to validate our theory.

- **Nov 24.** More typos have been fixed, and some polishing has been made.

---

### Meta-Review · Area_Chair_BFUc · 2026-01-17

**Summary:**

This paper focuses on the nonsmooth composite optimization template $\min_{x\in X} f(x) + r(x)$ where $f, r$ are convex, $f$ is Lipschitz and $r$ is $\mu$-strongly convex for some $\mu\geq 0$ (the authors handle both case $\mu=0$ and $\mu>0$) where the noise is heavy-tailed. The authors improve the rate of convergence for clipped SGD in both convex and strongly convex cases by a factor of poly$(1/d_{eff})$ where $d_{eff}$ is the effective dimension, which can have the same order as the dimension $d$ as the authors show an example in Appendix A. The authors also introduce a finer assumption compared to the existing works which help them also improve the rates for the rates in expectation. All the reviewers found the results and the techniques to be novel and interesting. I also agree with this.

It is also worth noting that the paper is quite well-written both in the main text and the supplementary material, which shows that the authors spend significant time and energy for polishing their work. This is a very positive aspect especially given that the submission is quite technical. I also liked how the authors have well-written discussions such as the one right before Section 7 about existing lower bounds and how their results fit into the picture.

**Reviewer Concerns:**

Important concerns of the reviewers were relating to the order of the improvement which depends on $d_{eff}$ and the authors addressed this manner in a convincing way to show how this can be at the order of the dimension. Next, Reviewer 9vd3 argued that the improvement in the log was easy to obtain when the domain is bounded, who was later convinced that the submission can handle the unbounded domain case.

**Reviewer Scores:**

I think that Reviewer 9vd3 would have increased their score since their main concern about the assumptions and the log improvement are addressed by the authors. Moreover, this reviewer already had many positive things to say about the submission such as "Great clipping lemma. The authors propose a refined clipping lemma that yields improved bounds for bias and variance. Importantly, this lemma (see Theorem 5) does not rely on fact that the norm of the true gradient should be less than $\tau/2$ (this condition is required in many papers, e.g. [1], [2]), which marks a significant methodological advantage."

I think that Reviewer j1s2 had similar concerns to Reviewer 9vd3 and that the authors addressed these concerns well and that this reviewer would too increase their score.

I think the other two reviewers would keep their already positive scores.

---

### Decision · Program_Chairs · 2026-01-26

Accept (Poster)